# Scaling Laws in Linear Regression: Compute, Parameters, and Data

**Licong Lin**
UC Berkeley
liconglin@berkeley.edu

**Jingfeng Wu**
UC Berkeley
uuujf@berkeley.edu

**Sham M. Kakade**
Harvard University
sham@seas.harvard.edu

**Peter L. Bartlett**
UC Berkeley and Google DeepMind
peter@berkeley.edu

**Jason D. Lee**
Princeton University
jasonlee@princeton.edu

## Abstract

Empirically, large-scale deep learning models often satisfy a neural scaling law: the test error of the trained model improves polynomially as the model size and data size grow. However, conventional wisdom suggests the test error consists of approximation, bias, and variance errors, where the variance error increases with model size. This disagrees with the general form of neural scaling laws, which predict that increasing model size monotonically improves performance.

We study the theory of scaling laws in an infinite dimensional linear regression setup. Specifically, we consider a model with $M$ parameters as a linear function of sketched covariates. The model is trained by one-pass stochastic gradient descent (SGD) using $N$ data. Assuming the optimal parameter satisfies a Gaussian prior and the data covariance matrix has a power-law spectrum of degree $a > 1$, we show that the reducible part of the test error is $\Theta(M^{-(a-1)} + N^{-(a-1)/a})$. The variance error, which increases with $M$, is dominated by the other errors due to the implicit regularization of SGD, thus disappearing from the bound. Our theory is consistent with the empirical neural scaling laws and verified by numerical simulation.

## 1 Introduction

Deep learning models, particularly those on a large scale, are pivotal in advancing the state-of-the-art across various fields. Recent empirical studies have shed light on the so-called *neural scaling laws* [see 26, 21, for example], which suggest that the generalization performance of these models improves polynomially as both model size, denoted by $M$, and data size, denoted by $N$, increase. The neural scaling law quantitatively describes the population risk as:

$$\mathcal{R}(M, N) \approx \mathcal{R}^* + \frac{c_1}{M^{a_1}} + \frac{c_2}{N^{a_2}}, \tag{1}$$

where $\mathcal{R}^*$ is a positive irreducible risk and $c_1, c_2, a_1, a_2$ are positive constants independent of $M$ and $N$. For instance, by fitting the above formula with empirical measurements in standard large-scale language benchmarks, Hoffmann et al. [21] estimated $a_1 \approx 0.34$ and $a_2 \approx 0.28$, while Besiroglu et al. [7] estimated that $a_1 \approx 0.35$ and $a_2 \approx 0.37$. Though the exact exponents depend on the tasks, neural scaling laws in (1) are observed consistently in practice and are used as principled guidance to build state-of-the-art models, especially under a compute budget [21].

From the perspective of statistical learning theory, (1) is rather intriguing. Standard statistical learning bounds [see 30, 41, for example] often decompose the population risk into the sum of irreducible

38th Conference on Neural Information Processing Systems (NeurIPS 2024).

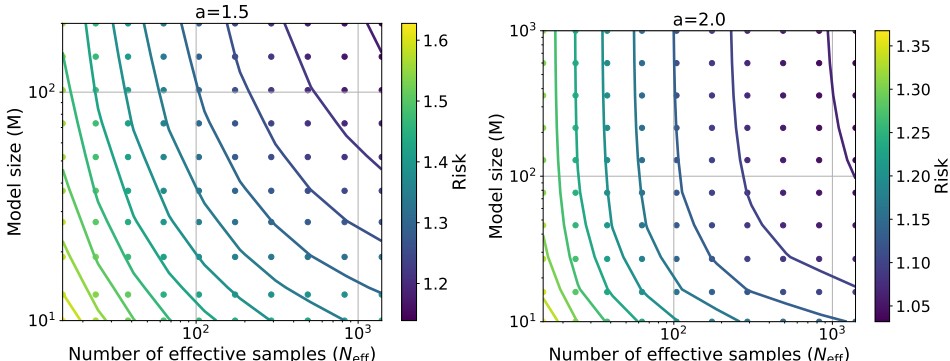

Figure 1: The expected risk (Risk) of the last iterate of (SGD) versus the effective sample size $N_{\texttt{eff}}$ and the model size $M$ for different power-law degrees $a$. The expected risk is computed by averaging over 1000 independent samples of $(\mathbf{w}^*, \mathbf{S})$. We fit the expected risk using the formula Risk $\sim \sigma^2 + c_1/M^{a_1} + c_2/N^{a_2}$ via minimizing the Huber loss as in [21]. Parameters: $\sigma = 1, \gamma = 0.1$. Left: For $a = 1.5$, $d = 20000$, the fitted exponents are $(a_1, a_2) = (0.54, 0.34) \approx (0.5, 0.33)$. Right: For $a = 2$, $d = 2000$, the fitted exponents are $(a_1, a_2) = (1.07, 0.49) \approx (1.0, 0.5)$. Note that the values of $(a_1, a_2)$ are close to our theoretical predictions $(a-1, 1-1/a)$ in both cases, verifying the sharpness of our risk bounds. More details can be found in Sections 4 and 5.

error, approximation error, bias error, and variance error (some theory replaces bias and variance errors by optimization and generalization errors, respectively) as in the form of

$$\mathcal{R}(M, N) = \mathcal{R}^* + \underbrace{\mathcal{O}\left(\frac{1}{M^{a_1}}\right)}_{\text{approximation}} + \underbrace{\mathcal{O}\left(\frac{1}{N^{a_2}}\right)}_{\text{bias}} + \underbrace{\mathcal{O}\left(\frac{c(M)}{N^{a_3}}\right)}_{\text{variance}}, \tag{2}$$

where $a_1, a_2, a_3$ are positive constants and $c(M)$ is a measure of *model complexity* that typically increases with the model size $M$. In (2), the approximation error is induced by the mismatch of the best-in-class predictor and the best possible predictor, hence decreasing with the model size $M$. The bias error is induced by the mismatch of the expected algorithm output and the best-in-class predictor, hence decreasing with the data size $N$. The variance error measures the uncertainty of the algorithm output, which decreases with the data size $N$ but increases with the model size $M$ (since the model complexity $c(M)$ increases).

**A mystery.** The empirical neural scaling law (1) is incompatible with the typical statistical learning theory bound (2). While the two error terms in the neural scaling law (1) can be explained by the approximation and bias errors in the theoretical bound (2) respectively, it is not clear why *the variance error is unobservable when fitting the neural scaling law empirically*. This difference must be reconciled, otherwise, the statistical learning theory and the empirical scaling law make conflict predictions: as the model size $M$ increases, the theoretical bound (2) predicts an increase of variance error that eventually causes an increase of the population risk, but the neural scaling law (1) predicts a decrease of the population risk. In other words, it remains unclear when to follow the prediction of the empirical scaling law (1) and when to follow that of the statistical learning bound (2).

Certain prior works provided risk upper bounds that do not grow with model size [see for example 36, 12]. Still, their results are insufficient for studying scaling law as those bounds require a large model size such that the approximation error is ignorable. Moreover, they do not provide instance-wise matching lower bounds to verify the tightness of the upper bounds. See a detailed discussion in Section 2.

**Our explanation.** We investigate this issue in an infinite dimensional linear regression setup. We only assume access to $M$-dimensional sketched covariates given by a fixed Gaussian sketch and their responses. We consider a linear predictor with $M$ trainable parameters, which is trained by one-pass *stochastic gradient descent* (SGD) with geometrically decaying stepsizes using $N$ sketched data. Assuming that the spectrum of the data covariance matrix satisfies a power-law of degree $a > 1$ and that the optimal model parameters satisfy a Gaussian prior, we derive matching upper and lower bounds on the population risk achieved by the SGD output (see Theorem 4.1). Specifically, we show

that

$$\mathcal{R}(M,N) = \mathcal{R}^* + \underbrace{\Theta\left(\frac{1}{M^{a-1}}\right) + \tilde{\Theta}\left(\frac{1}{(N\gamma)^{(a-1)/a}}\right)}_{\text{leading order given by the sum of Approx and Bias}}, \quad \mathrm{Var} = \underbrace{\tilde{\Theta}\left(\frac{\min\{M,(N\gamma)^{1/a}\}}{N}\right)}_{\text{higher order, thus unobservable}},$$

where $\gamma = \mathcal{O}(1)$ is the initial stepsize used in SGD and $\tilde{\Theta}(\cdot)$ hides $\log(N)$ factors. In our bound, the sum of the approximation and bias errors determines the order of the excess risk, while the variance error is of a strictly higher order and is therefore nearly unobservable when fitting $\mathcal{R}(M,N)$ as a function of $M$ and $N$ empirically. In addition, our analysis reveals that the small variance error is due to the implicit regularization effect of one-pass SGD [47]. Our theory suggests that the empirical neural scaling law (1) is a simplification of the statistical learning bound (2) in a special regime when strong regularization (either implicit or explicit) is employed.

Moreover, we generalize the above scaling law to (1) constant stepsize SGD with iterate average (see Theorem F.6), (2) cases where the optimal model parameter satisfies an anisotropic prior (see Theorem 4.2), and (3) where the spectrum of the data covariance matrix satisfies a logarithmic power law (see Theorem 4.3).

**Emprical evidence.**    Based on our theoretical results, we conjecture that the clean neural scaling law (1) observed in practice is due to the disappearance of variance error caused by strong regularization. Two pieces of empirical evidence to support our understanding. First, large language models that follow the scaling law (1) are often *underfitted*, as the models are trained over a single pass or a few passes over the data [27, 31, 9, 39]. When models are underfitted, the variance error tends to be smaller. Second, when language models are trained with multiple passes (up to 7 passes), Muennighoff et al. [31] found that the clean scaling law in (1) no longer holds and they proposed a more sophisticated scaling law to explain their data. This can be explained by a relatively large variance error caused by multiple passes.

**Notation.**    For two positive-valued functions $f(x)$ and $g(x)$, we write $f(x) \lesssim g(x)$ (and $f(x) = \mathcal{O}(g(x))$) or $f(x) \gtrsim g(x)$ (and $f(x) = \Omega(g(x))$) if $f(x) \leq cg(x)$ or $f(x) \geq cg(x)$ holds for some absolute (if not otherwise specified) constant $c > 0$ respectively. We write $f(x) \approx g(x)$ (and $f(x) = \Theta(g(x))$) if $f(x) \lesssim g(x) \lesssim f(x)$. For two vectors $\mathbf{u}$ and $\mathbf{v}$ in a Hilbert space, we denote their inner product by $\langle \mathbf{u}, \mathbf{v} \rangle$ or $\mathbf{u}^\top \mathbf{v}$. For two matrices $\mathbf{A}$ and $\mathbf{B}$ of appropriate dimensions, we define their inner product by $\langle \mathbf{A}, \mathbf{B} \rangle := \mathrm{tr}(\mathbf{A}^\top \mathbf{B})$. We use $\| \cdot \|$ to denote the operator norm for matrices and $\ell_2$-norm for vectors. For a positive semi-definite (PSD) matrix $\mathbf{A}$ and a vector $\mathbf{v}$ of appropriate dimension, we write $\|\mathbf{v}\|_{\mathbf{A}}^2 := \mathbf{v}^\top \mathbf{A} \mathbf{v}$. For a symmetric matrix $\mathbf{A}$, we use $\mu_j(\mathbf{A})$ to refer to the $j$-th eigenvalue of $\mathbf{A}$ and $r(\mathbf{A})$ to refer to its rank. Finally, $\log(\cdot)$ refers to logarithm base 2.

## 2   Related work

**Empirical scaling laws.**    In recent years, the scaling laws of deep neural networks in compute, sample size, and model size have been widely studied across different models and domains [20, 35, 26, 19, 21, 46, 31]. The early work by Kaplan et al. [26] first proposed the neural scaling laws of transformer-based models. They observed that the test loss exhibits a power-law decay in quantities including the amount of compute, sample size, and model size, and provided joint formulas in these quantities to predict the test loss. The proposed formulas were later generalized and refined in subsequent works [19, 21, 1, 10, 31]. Notably, Hoffmann et al. [21] proposed the Chinchilla law, that is, (1) with $a_1 \approx 0.34$ and $a_2 \approx 0.28$. The empirical observation guided them to allocate data and model size under a given compute budget. The Chinchilla law is further revised by Besiroglu et al. [7]. Motivated by the Chinchilla law, Muennighoff et al. [31] considered the effect of multiple passes over training data and empirically fitted a more sophisticated scaling law that takes account of the effect of data reusing.

**Theory of scaling laws.**    Although neural scaling laws have been empirically observed over a broad spectrum of problems, there is a relatively limited literature on understanding these scaling laws from a theoretical perspective [37, 4, 28, 22, 42, 29, 23, 8, 2, 32, 17]. Among these works, [37] showed that the test loss scales as $N^{4/d}$ for regression on data with intrinsic dimension $d$. Hutter [22] studied a toy problem under which a non-trivial power of $N$ arises in the test loss. Jain et al. [23]

considered scaling laws in data selection. Bahri et al. [4] considered a linear teacher-student model under a power-law spectrum assumption on the covariates, and they showed that the test loss of the ordinary least square estimator decreases following a power law in sample size $N$ (resp. model size $M$) when the model size $M$ (resp. sample size $N$) is infinite. Bordelon et al. [8] considered a linear random feature model and analyzed the test loss of the solution found by (batch) gradient flow. They focused on the bottleneck regimes where two of the quantities $N, M, T$ (training steps) are infinite and showed that the risk has a power-law decay in the remaining quantity. The problem in Bahri et al. [4], Bordelon et al. [8] can be viewed as a sketched linear regression model similar to ours. It should be noted that both Bahri et al. [4] and Bordelon et al. [8] only derived the dependence of population risk on one of the data size, model size, or training steps in the asymptotic regime where the remaining quantities go to infinity, and their derivations are based on statistical physics heuristics. In comparison, we prove matching (ignoring constant factors) upper and lower risk bounds jointly depending on the finite model size $M$ and data size $N$.

**Implicit regularization of SGD.** One-pass SGD in linear regression has been extensively studied in both the classical finite-dimensional setting [34, 3, 14, 16, 25, 24, 18] and the modern high-dimensional setting [15, 6, 48, 47, 44, 45, 40]. In particular, Zou et al. [47] showed that SGD induces an implicit regularization effect that is comparable to, and in certain cases even more preferable than, the explicit regularization effect induced by ridge regression. This is one of the key motivations of our scaling law interpretation. From a technical perspective, we utilize the sharp finite-sample and dimension-free analysis of SGD developed by Zou et al. [48], Wu et al. [44, 45]. Different from them, we consider a sequence of linear regression models with an increasing number of trainable parameters given by data sketch. Our main technical innovation is to sharply control the effect of data sketch. Some of our intermediate results, for example, tight bounds on the spectrum of the sketched data covariance under the power law (see Lemma 6.2), might be of independent interest.

Prior works investigated linear regression with random features [36, 12], which can be viewed as a kind of sketched features via random coordinate selection. They mainly focused on the small approximation error regime, where the model size (or the number of features) is much larger than the data size. In comparison, we treat both model size and data size as free variables. Moreover, we provide matching upper and lower bounds while prior works mainly focused on upper bounds. These two innovations are crucial for studying scaling laws that predict test error as a function of both model size and data size. Finally, in the comparable regimes with small or zero approximation error, our excess risk bounds recover the bounds in prior works [36, 12, 33, 15, 13].

## 3 Setup

We use $\mathbf{x} \in \mathbb{H}$ to denote a feature vector, where $\mathbb{H}$ is a finite $d$-dimensional or countably infinite dimensional Hilbert space, and $y \in \mathbb{R}$ to denote its label. In linear regression, we measure the population risk of a parameter $\mathbf{w} \in \mathbb{H}$ by the mean squared error,

$$\mathcal{R}(\mathbf{w}) := \mathbb{E}\big(\langle \mathbf{x}, \mathbf{w} \rangle - y\big)^2, \quad \mathbf{w} \in \mathbb{H},$$

where the expectation is over $(\mathbf{x}, y) \sim P$ for some distribution $P$ on $\mathbb{H} \times \mathbb{R}$.

**Definition 1** (Data covariance and optimal parameter). Let $\mathbf{H} := \mathbb{E}[\mathbf{x}\mathbf{x}^\top]$ be the data covariance. Assume that $\mathrm{tr}(\mathbf{H})$ and all entries of $\mathbf{H}$ are finite. Let $(\lambda_i)_{i \geq 0}$ be the eigenvalues of $\mathbf{H}$ sorted in non-increasing order. Let $\mathbf{w}^* \in \arg\min_{\mathbf{w}} \mathcal{R}(\mathbf{w})$ be the optimal model parameter[1]. Assume that $\|\mathbf{w}^*\|_{\mathbf{H}}^2 := (\mathbf{w}^*)^\top \mathbf{H} \mathbf{w}^*$ is finite.

We only assume access to $M$-dimensional sketched covariates and their responses, that is, $(\mathbf{S}\mathbf{x}, y)$, where $\mathbf{S} \in \mathbb{R}^M \times \mathbb{H}$ is a fixed *sketch* matrix. We focus on the Gaussian sketch matrix[2], that is, entries of $\mathbf{S}$ are independently sampled from $\mathcal{N}(0, 1/M)$. We then consider linear predictors with $M$ trainable parameters given by

$$f_{\mathbf{v}} : \mathbb{H} \to \mathbb{R}, \quad \mathbf{x} \mapsto \langle \mathbf{v}, \mathbf{S}\mathbf{x} \rangle,$$

where $\mathbf{v} \in \mathbb{R}^M$ are the trainable parameters. Varying $M$ should be viewed as a linear analog of varying the neural network model size. Our sketched linear regression setting is comparable to the teacher-student setting considered by Bahri et al. [4], Bordelon et al. [8].

---

[1] If $\arg\min \mathcal{R}(\cdot)$ is not unique, we choose $\mathbf{w}^*$ to be the minimizer with minimal $\mathbf{H}$-norm.
[2] Our results can be extended to other sketching methods [see 43, for example].

We consider the training of $f_{\mathbf{v}}$ via one-pass *stochastic gradient descent* (SGD), that is,

$$
\begin{aligned}
\mathbf{v}_t &:= \mathbf{v}_{t-1} - \gamma_t \big( f_{\mathbf{v}_{t-1}}(\mathbf{x}_t) - y_t \big) \nabla_{\mathbf{v}} f_{\mathbf{v}_{t-1}}(\mathbf{x}_t) \\
&:= \mathbf{v}_{t-1} - \gamma_t \big( \mathbf{x}_t^\top \mathbf{S}^\top \mathbf{v}_{t-1} - y_t \big) \mathbf{S} \mathbf{x}_t, \qquad t = 1, \ldots, N,
\end{aligned}
\tag{SGD}
$$

where $(\mathbf{x}_t, y_t)_{t=1}^N$ are independent samples from $P$ and $(\gamma_t)_{t=1}^N$ are the stepsizes. We consider a popular geometric decaying stepsize scheduler [18, 44],

$$
\text{for } t = 1, \ldots, N, \ \gamma_t := \gamma/2^\ell, \ \text{where } \ell = \lfloor t/(N/\log(N)) \rfloor.
\tag{3}
$$

Here, the initial stepsize $\gamma$ is a hyperparameter for the SGD algorithm. Without loss of generality, we assume the initial parameter is $\mathbf{v}_0 = 0$. The output of the SGD algorithm is the last iterate $\mathbf{v}_N$. Our proof techniques apply to other stepsize schedulers (e.g., polynomial decay) as well, but we focus on geometric decay as it is known to achieve near minimax-optimal excess risk for the last iterate of SGD [18].

Conditioning on a sketch matrix $\mathbf{S} \in \mathbb{R}^M \times \mathbb{H}$, each parameter $\mathbf{v} \in \mathbb{R}^M$ induces a sketched predictor through $\mathbf{x} \mapsto \langle \mathbf{S}^\top \mathbf{v}, \mathbf{x} \rangle$, and we denote its risk by

$$
\mathcal{R}_M(\mathbf{v}) := \mathcal{R}(\mathbf{S}^\top \mathbf{v}) = \mathbb{E} \big( \langle \mathbf{S} \mathbf{x}, \mathbf{v} \rangle - y \big)^2, \quad \mathbf{v} \in \mathbb{R}^M.
$$

By increasing $M$ and $N$, we have a sequence of datasets and trainable parameters of increasing sizes, respectively. This prepares us to study the scaling law (1) in the sketched linear regression problem, that is, to understand $\mathcal{R}_M(\mathbf{v}_N)$ as a function of both $M$ and $N$.

**Risk decomposition.** In a standard way, we decompose the risk achieved by $\mathbf{v}_N$, the last iterate of (SGD), to the sum of *irreducible risk*, *approximization error*, and *excess risk* as follows,

$$
\mathcal{R}_M(\mathbf{v}_N) = \underbrace{\min \mathcal{R}(\cdot)}_{\text{Irreducible}} + \underbrace{\min \mathcal{R}_M(\cdot) - \min \mathcal{R}(\cdot)}_{\text{Approx}} + \underbrace{\mathcal{R}_M(\mathbf{v}_N) - \min \mathcal{R}_M(\cdot)}_{\text{Excess}}.
\tag{4}
$$

We emphasize that the irreducible risk is independent of $M$ and $N$ and thus can be viewed as a constant; the approximation error is determined by the sketch matrix $\mathbf{S}$, thus depends on $M$ but is independent of $N$; the excess risk depends on both $M$ and $N$ as it is determined by the algorithm.

## 4 Scaling laws

We first demonstrate a scaling-law behavior when the data spectrum satisfies a power law.

**Assumption 1** (Distributional conditions). *Assume the following about the data distribution.*

A. **Gaussian design.** *Assume that $\mathbf{x} \sim \mathcal{N}(0, \mathbf{H})$.*

B. **Well-specified model.** *Assume that $\mathbb{E}[y|\mathbf{x}] = \mathbf{x}^\top \mathbf{w}^*$. Define $\sigma^2 := \mathbb{E}(y - \mathbf{x}^\top \mathbf{w}^*)^2$.*

C. **Parameter prior.** *Assume that $\mathbf{w}^*$ satisfies a prior such that $\mathbb{E}(\mathbf{w}^*)^{\otimes 2} = \mathbf{I}$.*

**Assumption 2** (Power-law spectrum). *There exists $a > 1$ such that the eigenvalues of $\mathbf{H}$ satisfy $\lambda_i \eqsim i^{-a}$, $i > 0$.*

**Theorem 4.1** (Scaling law). *Suppose that Assumptions 1 and 2 hold. Consider an $M$-dimensional sketched predictor trained by (SGD) with $N$ samples. Let $N_{\texttt{eff}} := N/\log(N)$ and recall the risk decomposition in (4). Then there exists some $a$-dependent constant $c > 0$ such that when the initial stepsize $\gamma \le c$, with probability at least $1 - e^{-\Omega(M)}$ over the randomness of the sketch matrix $\mathbf{S}$, we have*

1. Irreducible $:= \mathcal{R}(\mathbf{w}^*) = \sigma^2$.

2. $\mathbb{E}_{\mathbf{w}^*} \mathsf{Approx} \eqsim M^{1-a}$.

3. *Suppose in addition $\sigma^2 \gtrsim 1$. The expected excess risk (Excess) can be decomposed into a bias error (Bias) and a variance error (Var), namely,*

$$
\mathbb{E} \mathsf{Excess} \eqsim \mathsf{Bias} + \sigma^2 \mathsf{Var},
$$

*where the expectation is over the randomness of $\mathbf{w}^*$ and $(\mathbf{x}_i, y_i)_{i=1}^N$. Moreover,* Bias *and* Var *satisfy*

$$\text{Bias} \lesssim \max\left\{ M^{1-a}, \ (N_{\mathtt{eff}}\gamma)^{1/a-1} \right\},$$

$$\text{Bias} \gtrsim (N_{\mathtt{eff}}\gamma)^{1/a-1} \ \textit{when } (N_{\mathtt{eff}}\gamma)^{1/a} \leq M/c \quad \textit{for some constant } c > 0,$$

$$\text{Var} \asymp \min\left\{ M, \ (N_{\mathtt{eff}}\gamma)^{1/a} \right\}/N_{\mathtt{eff}}.$$

*In all results, the hidden constants only depend on the power-law degree $a$. As a direct consequence, when $\sigma^2 \asymp 1$, it holds with probability at least $1 - e^{-\Omega(M)}$ over the randomness of the sketch matrix $\mathbf{S}$ that*

$$\mathbb{E}\mathcal{R}_M(\mathbf{v}_N) = \sigma^2 + \Theta\left(\frac{1}{M^{a-1}}\right) + \Theta\left(\frac{1}{(N_{\mathtt{eff}}\gamma)^{(a-1)/a}}\right),$$

*where the expectation is over the randomness of $\mathbf{w}^*$ and $(\mathbf{x}_i, y_i)_{i=1}^N$.*

Theorem 4.1 shows a sharp (up to constant factors) scaling law risk bound under an isotroptic prior assumption and the power-law spectrum assumption. We emphasize that the scaling law bound in Theorem 4.1 holds for every $M, N \geq 1$. We also remark that the sum of approximization and bias errors dominates $\mathbb{E}\mathcal{R}_M(\mathbf{v}_N) - \sigma^2$, whereas the variance error is of strict higher order in terms of both $M$ and $N$, and is thus disappeared in the population risk bound.

**Optimal stepsize.** Based on the tight scaling law in Theorem 4.1, we can calculate the optimal stepsize that minimizes the risk. Specifically, the optimal stepsize is $\gamma \asymp 1$ when $N_{\mathtt{eff}} \lesssim M^a$ and can be anything such that $M^a/N_{\mathtt{eff}} \lesssim \gamma \lesssim 1$ when $N_{\mathtt{eff}} \gtrsim M^a$. In both cases, choosing $\gamma \asymp 1$ is optimal. When the sample size is large such that $N_{\mathtt{eff}} \gtrsim M^a$, the optimal stepsize is relatively robust and can be chosen from a range.

**Allocation of data and model sizes.** Following Hoffmann et al. [21], we measure the compute complexity by $MN$ as (SGD) queries $M$-dimensional gradients for $N$ times. Given a total compute budget of $MN = C$, from Theorem 6.1 and $N_{\mathtt{eff}} := N/\log(N)$, we see that the best population risk is achieved by setting $\gamma = \Theta(1)$, $M = \tilde{\Theta}(C^{1/(a+1)})$, and $N = \tilde{\Theta}(C^{a/(a+1)})$. Our theory suggests setting a data size slightly larger than the model size when the compute budget is the bottleneck.

**Comparison with [8].** The work by Bordelon et al. [8] considered the scaling law of batch gradient descent (or gradient flow) on a teacher-student model (see their equation (14)). Their teacher-student model can be viewed as our sketched linear regression model. However, we consider one-pass SGD, therefore in our setting the number of gradient steps is equivalent to the data size. When we equalize the number of gradient steps and the data size in their equation (14) and set the parameter prior as Assumption 1C, their prediction is consistent with ours. However, our analysis shows the computational advantage of SGD over batch GD since each iteration requires only $1/N$ the compute. Bordelon et al. [8] obtained the limit of the population risk as two out of the data size, model size, and the number of gradient steps go to infinity based on statistical physics heuristics. In comparison, we obtain upper and lower risk bounds that hold for any finite $M$ and $N$ and match ignoring a constant factor depending only on the spectrum power-law degree $a$.

**Average of the SGD iterates** Results similar to Theorem 4.1 can also be established for the average of the iterates of online SGD with constant stepsize [34, 16, 25, 24, 48]. All results will be the same once replacing the effective sample size $N_{\mathtt{eff}}$ in Theorem 4.1 to the sample size $N$. For more details see Theorem F.6 in Appendix F.

## 4.1 Scaling law under source condition

The isotropic parameter prior condition (Assumption 1C) in Theorem 4.1 can be generalized to the following anisotropic version [11].

**Assumption 3** (Source condition). *Let $(\lambda_i, \mathbf{v}_i)_{i>0}$ be the eigenvalues and eigenvectors of $\mathbf{H}$ with $(\lambda_i)_{i>0}$ in non-increasing order. Assume $\mathbf{w}^*$ satisfies a prior such that*

$$\textit{for } i \neq j, \ \ \mathbb{E}\langle\mathbf{v}_i, \mathbf{w}^*\rangle\langle\mathbf{v}_j, \mathbf{w}^*\rangle = 0; \ \textit{and for } i > 0, \ \ \mathbb{E}\lambda_i\langle\mathbf{v}_i, \mathbf{w}^*\rangle^2 \asymp i^{-b}, \ \textit{for some } b > 1.$$

A larger exponent $b$ implies a faster decay of signal $\mathbf{w}^*$ and thus corresponds to a simpler task [11]. Note that Assumption 1C satisfies Assumption 3 with $b = a$.

**Theorem 4.2** (Scaling law under source condition). *In Theorem 4.1, suppose Assumption 1C is replaced by Assumption 3 with $1 < b < a + 1$. Then there exists some $a$-dependent constant $c > 0$ such that when $\gamma \leq c$, with probability at least $1 - e^{-\Omega(M)}$ over the randomness of the sketch matrix $\mathbf{S}$, we have*

$$\mathbb{E}\mathcal{R}_M(\mathbf{v}_N) = \sigma^2 + \underbrace{\Theta\left(\frac{1}{M^{b-1}}\right) + \Theta\left(\frac{1}{(N_{\texttt{eff}}\gamma)^{(b-1)/a}}\right)}_{\text{Approx+Bias}} + \underbrace{\Theta\left(\frac{\min\left\{M,\ (N_{\texttt{eff}}\gamma)^{1/a}\right\}}{N_{\texttt{eff}}}\right)}_{\text{Var}}.$$

*where the expectation is over the randomness of $\mathbf{w}^*$ and $(\mathbf{x}_i, y_i)_{i=1}^N$, and $\Theta(\cdot)$ hides constants that may depend on $(a, b)$.*

When $1 < b \leq a$, the tasks are relatively hard (compared to when $b = a$), and the variance error is dominated by the sum of approximation and bias errors for all choices of $M$, $N$, and $\gamma \lesssim 1$. In this case, Theorem 4.2 gives the same prediction about optimal stepsize and optimal allocation of data and model sizes under compute budget as Theorem 4.1.

When $a < b < a + 1$, the tasks are relatively easy (compared to when $b = a$), and variance remains dominated by the sum of approximation and bias error if the stepsize is optimally tuned. Recall that $\gamma \lesssim 1$, thus we can rewrite the risk bound in Theorem 4.2 as

$$\mathbb{E}\mathcal{R}_M(\mathbf{v}_N) - \sigma^2 \approx \frac{1}{\min\left\{M,\ (N_{\texttt{eff}}\gamma)^{1/a}\right\}^{b-1}} + \frac{\min\left\{M,\ (N_{\texttt{eff}}\gamma)^{1/a}\right\}}{N_{\texttt{eff}}}$$

$$\approx \begin{cases} \min\left\{M,\ (N_{\texttt{eff}}\gamma)^{1/a}\right\}/N_{\texttt{eff}} & M \gtrsim N_{\texttt{eff}}^{1/b} \text{ and } N_{\texttt{eff}}^{a/b-1} \lesssim \gamma \lesssim 1, \\ \min\left\{M,\ (N_{\texttt{eff}}\gamma)^{1/a}\right\}^{1-b} & M \lesssim N_{\texttt{eff}}^{1/b} \text{ or } \gamma \lesssim N_{\texttt{eff}}^{a/b-1}. \end{cases}$$

Therefore the optimal stepsize and the risk under the optimal stepsize is

$$\gamma \approx N_{\texttt{eff}}^{a/b-1} \text{ if } M \gtrsim N_{\texttt{eff}}^{1/b}, \quad \text{and } M^a/N_{\texttt{eff}} \lesssim \gamma \lesssim 1 \text{ if } M \lesssim N_{\texttt{eff}}^{1/b}.$$

Under the *optimally tuned* stepsize, the population risk is in the form of

$$\min_\gamma \mathbb{E}\mathcal{R}_M(\mathbf{v}_N) = \sigma^2 + \Theta(N_{\texttt{eff}}^{(1-b)/b}) + \Theta(M^{1-b}),$$

which is again in the scaling law form (1). This is expected since an optimally tuned stepsize controls the variance error by adjusting the strength of the implicit bias of SGD. Under a fixed compute budget $C = MN$, our theory suggests to assign $M = \tilde{\Theta}(C^{1/(b+1)})$ and $N = \tilde{\Theta}(C^{b/(b+1)})$, and set the stepsize to $\gamma \approx \tilde{\Theta}(C^{(a-b)/(b+1)})$.

When $b \geq a + 1$, the tasks are even simpler. We provide upper and lower bounds in Appendix D.3. However, there exists a gap between the bounds, fixing which is left for future work.

Moreover, we note that in the comparable regimes where $M$ is large, the results in Theorem 4.2 match existing bounds on the risk of SGD iterates and ridge estimators [33, 36].

### 4.2 Scaling law under logarithmic power law

We also derive the risk formula when the data covariance has a logarithmic power-law spectrum [5].

**Assumption 4** (Logarithmic power-law spectrum). *There exists $a > 1$ such that the eigenvalues of $\mathbf{H}$ satisfy $\lambda_i \approx i^{-1}\log^{-a}(i + 1)$, $i > 0$.*

**Theorem 4.3** (Scaling law under logarithmic power spectrum). *In Theorem 4.1, suppose Assumption 2 is replaced by Assumption 4. Then with probability at least $1 - e^{-\Omega(M)}$ over the randomness of the sketch matrix $\mathbf{S}$, we have*

$$\mathbb{E}\mathcal{R}_M(\mathbf{v}_N) = \sigma^2 + \Theta\left(\frac{1}{\log^{a-1}(M)}\right) + \Theta\left(\frac{1}{\log^{a-1}(N_{\texttt{eff}}\gamma)}\right), \quad \mathsf{Var} \approx \frac{\min\left\{M,\ \frac{N_{\texttt{eff}}\gamma}{\log^a(N_{\texttt{eff}}\gamma)}\right\}}{N_{\texttt{eff}}},$$

*where the expectation is over the randomness of $\mathbf{w}^*$ and $(\mathbf{x}_i, y_i)_{i=1}^N$.*

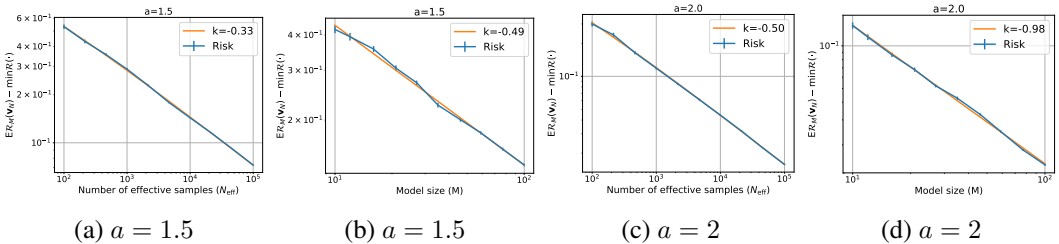

(a) $a = 1.5$  (b) $a = 1.5$  (c) $a = 2$  (d) $a = 2$

Figure 2: The expected risk of the last iterate of (SGD) minus the irreducible risk versus the effective sample size and model size. Parameters $\sigma = 1, \gamma = 0.1$. (a), (b): $a = 1.5, d = 10000$; (c), (d): $a = 2, d = 1000$. The error bars denote the $\pm 1$ standard deviation of estimating the expected risk using 100 independent samples of $(\mathbf{w}^*, \mathbf{S})$. We use linear functions to fit the expected risk under the log-log scale and report the slope of the fitted lines (denoted by $k$).

Theorem 4.3 provides a scaling law under the logarithmic power-law spectrum. Similar to Theorem 4.1, the variance error is dominated by the approximation and bias errors for all choices of $M$, $N$, and $\gamma$, and thus disappeared from the risk bound. Different from Theorem 4.1, here the population risk is a polynomial of $\log(M)$ and $\log(N_{\mathtt{eff}}\gamma)$.

## 5 Experiments

In this section, we examine the relation between the expected risk of the (SGD) output, the data size $N$, and the model size $M$ when the covariates satisfy a power-law covariance spectrum. Although our results in Section 4 hold with high probability over $\mathbf{S}$, for simplicity, we assume the expectation of the risk is taken over both $\mathbf{w}^*$ and $\mathbf{S}$ in our simulations. We adopt the model in Section 3 and train it using one-pass (SGD) with geometric decaying stepsize (3). We choose the dimension $d$ sufficiently large to approximate the infinite-dimensional case, and the data are generated so that Assumption 1 is satisfied. Moreover, we choose the covariance $\mathbf{H} \in \mathbb{R}^{d \times d}$ to be diagonal with $\mathbf{H}_{ii} \propto i^a$ and $\mathrm{tr}(\mathbf{H}) = 1$ for some $a > 1$. From Figure 1, we observe that the risk indeed follows a power-law formula jointly in the number of samples and the number of parameters. In addition, the fitted exponents are aligned with our theoretical predictions $(a - 1, 1 - 1/a)$ in Theorem 4.1. Figure 2 shows the scaling of the expected risk in data size (or model size) when the model size (or data size) is relatively large. We see that the expected risk also satisfies a power-law decay with exponents matching our predictions. It is noteworthy that our simulations demonstrate stronger observations than the theoretical results in Theorem 4.1, which only establishes matching upper and lower bounds up to a constant factor. Additional simulation results on the risk of the average of (SGD) iterates can be found in Appendix F.

## 6 Risk bounds under a general spectrum

In this section, we present some general results on the upper and lower bounds of the risk of the output of (SGD). Due to the rotational invariance of the sketched matrix $\mathbf{S}$, without loss of generality, we assume the covariance $\mathbf{H}$ is diagonal with non-increasing diagonal entries. Our main results in Section 4 are directly built on the general bounds introduced here.

**Assumption 5** (General distributional conditions). *Assume the following about the data distribution.*

A. **Hypercontractivity.** *There exists $\alpha \geq 1$ such that for every PSD matrix $\mathbf{A}$ it holds that*

$$\mathbb{E}\mathbf{x}\mathbf{x}^\top \mathbf{A}\mathbf{x}\mathbf{x}^\top \preceq \alpha \, \mathrm{tr}(\mathbf{H}\mathbf{A})\mathbf{H}.$$

B. **Misspecified model.** *There exists $\sigma^2 > 0$ such that $\mathbb{E}(y - \mathbf{x}^\top\mathbf{w}^*)^2\mathbf{x}\mathbf{x}^\top \preceq \sigma^2\mathbf{H}$.*

It is clear that Assumption 1 implies Assumption 5 with $\alpha = 3$.

**Excess risk decomposition.**  Conditioning on the sketch matrix $\mathbf{S}$, the training of the sketched linear predictor can be viewed as an $M$-dimensional linear regression problem. We can therefore

invoke existing SGD analysis [44, 45] to sharply control the excess risk by controlling the bias and variance errors. Specifically, let us define the ($\mathbf{w}^*$-dependent) bias error as

$$\mathsf{Bias}(\mathbf{w}^*) := \left\| \prod_{t=1}^{N} \left(\mathbf{I} - \gamma_t \mathbf{SHS}^\top\right)\mathbf{v}^* \right\|_{\mathbf{SHS}^\top}^2, \quad \text{where } \mathbf{v}^* := (\mathbf{SHS}^\top)^{-1}\mathbf{SHw}^*, \tag{5}$$

and the variance error as

$$\mathsf{Var} := \frac{\#\{\tilde{\lambda}_j \geq 1/(N_{\mathtt{eff}}\gamma)\} + (N_{\mathtt{eff}}\gamma)^2 \sum_{\tilde{\lambda}_j < 1/(N_{\mathtt{eff}}\gamma)} \tilde{\lambda}_j^2}{N_{\mathtt{eff}}}, \quad N_{\mathtt{eff}} := N/\log(N), \tag{6}$$

where $(\tilde{\lambda}_j)_{j=1}^{M}$ are eigenvalues of $\mathbf{SHS}^\top$. We also let $\mathsf{Bias} := \mathbb{E}\mathsf{Bias}(\mathbf{w}^*)$, where the expectation is over the prior of $\mathbf{w}^*$. Using the existing results on the output of (SGD) in Wu et al. [44, 45], we show that the excess risk in (4) can be exactly decomposed as the sum of bias and variance errors under weak conditions.

**Theorem 6.1** (Excess risk decomposition). *Conditioning on the sketch matrix $\mathbf{S}$, consider the excess risk in (4) induced by the output of (SGD). Assume $\mathbf{v}_0 = 0$. Then for any $\mathbf{w}^* \in \mathbb{H}$,*

1. *Under Assumptions 5A and 5B and suppose $\gamma \leq 1/\left(c\alpha \operatorname{tr}(\mathbf{SHS}^\top)\right)$ for some constant $c > 0$, we have*
$$\mathbb{E}\mathsf{Excess} \lesssim \mathsf{Bias}(\mathbf{w}^*) + \left(\alpha\|\mathbf{w}^*\|_{\mathbf{H}}^2 + \sigma^2\right)\mathsf{Var}.$$

2. *Under the stronger Assumptions 1A and 1B and suppose $\gamma \leq 1/\left(c\alpha \operatorname{tr}(\mathbf{SHS}^\top)\right)$ for some constant $c > 0$, we have*
$$\mathbb{E}\mathsf{Excess} \gtrsim \mathsf{Bias}(\mathbf{w}^*) + \sigma^2 \mathsf{Var}.$$

*In both results, the expectations of $\mathsf{Excess}$ are taken over $(\mathbf{x}_t, y_t)_{t=1}^{N}$.*

Assuming that the signal-to-noise ratio is upper bounded, that is, $\|\mathbf{w}^*\|_{\mathbf{H}}^2/\sigma^2 \lesssim 1$, then the bias-variance decomposition of the excess risk is sharp up to constant factors.

The variance error is in a nice form and can be computed using the following important lemma on the spectrum of $\mathbf{SHS}^\top$. Similar results for logarithmic power-law are also established in Lemma G.6 in Appendix G.

**Lemma 6.2** (Power law). *Under Assumption 2, it holds with probability at least $1 - e^{-\Omega(M)}$ that*
$$\mu_j(\mathbf{SHS}^\top) \asymp \mu_j(\mathbf{H}) \asymp j^{-a}, \quad j = 1, \dots, M.$$

For any $0 \leq k^* \leq k^\dagger \leq \infty$, let $\mathbf{S}_{k^*:k^\dagger} \in \mathbb{R}^{M \times (k^\dagger - k^*)}$ denote the matrix formed by the $k^* + 1 - k^\dagger$-th columns of $\mathbf{S}$. We also abuse the notation $k^\dagger : \infty$ for $k^\dagger : d$ when $d$ is finite. We let $\mathbf{H}_{k^*:k^\dagger} \in \mathbb{R}^{(k^\dagger - k^*) \times (k^\dagger - k^*)}$ be the submatrix of $\mathbf{H}$ formed by the $k^* + 1 - k^\dagger$-th eigenvalues. For the approximation and bias error, we use the following upper and lower bounds to compute their values.

**Theorem 6.3** (A general upper bound). *Suppose Assumption 5 holds. Assume $\mathbf{v}_0 = 0$, $r(\mathbf{H}) \geq 2M$ and the initial stepsize satisfies $\gamma < 1/(c\alpha \operatorname{tr}(\mathbf{SHS}^\top))$ for some constant $c > 0$. Then for any $k_1, k_2 \leq M/3$, with probability at least $1 - e^{-\Omega(M)}$*

$$\mathsf{Approx} \lesssim \|\mathbf{w}_{k_1:\infty}^*\|_{\mathbf{H}_{k_1:\infty}}^2 + \left(\frac{\sum_{i>k_1}\lambda_i}{M} + \lambda_{k_1+1} + \sqrt{\frac{\sum_{i>k_1}\lambda_i^2}{M}}\right)\|\mathbf{w}_{0:k_1}^*\|^2,$$

$$\mathsf{Bias}(\mathbf{w}^*) \lesssim \frac{\|\mathbf{w}_{0:k_2}^*\|_2^2}{N_{\mathtt{eff}}\gamma} \cdot \left[\frac{\mu_{M/2}(\mathbf{S}_{k_2:\infty}\mathbf{H}_{k_2:\infty}\mathbf{S}_{k_2:\infty}^\top)}{\mu_M(\mathbf{S}_{k_2:\infty}\mathbf{H}_{k_2:\infty}\mathbf{S}_{k_2:\infty}^\top)}\right]^2 + \|\mathbf{w}_{k_2:\infty}^*\|_{\mathbf{H}_{k_2:\infty}}^2.$$

**Theorem 6.4** (A general lower bound). *Suppose Assumption 1 holds. Assume $\mathbf{v}_0 = 0$, $r(\mathbf{H}) \geq M$ and the initial stepsize $\gamma < 1/\left(c \operatorname{tr}(\mathbf{SHS}^\top)\right)$ for some constant $c > 0$. Then*

$$\mathbb{E}_{\mathbf{w}^*}\mathsf{Approx} \gtrsim \sum_{i=M}^{d}\lambda_i, \quad \mathbb{E}_{\mathbf{w}^*}\mathsf{Bias}(\mathbf{w}^*) \gtrsim \sum_{i:\tilde{\lambda}_i < 1/(N_{\mathtt{eff}}\gamma)}\frac{\mu_i(\mathbf{SH}^2\mathbf{S}^\top)}{\mu_i(\mathbf{SHS}^\top)}$$

*almost surely, where $(\lambda_i)_{i=1}^{d}$ are eigenvalues of $\mathbf{H}$ in non-increasing order, $(\tilde{\lambda}_i)_{i=1}^{d}$ are eigenvalues of $\mathbf{SHS}^\top$ in non-increasing order.*

# 7 Conclusion

We analyze neural scaling laws in infinite-dimensional linear regression. We consider a linear predictor with $M$ trainable parameters on the sketched covariates, which is trained by one-pass stochastic gradient descent with $N$ data. Under a Gaussian prior assumption on the optimal model parameter and a power law (of degree $a > 1$) assumption on the spectrum of the data covariance, we derive matching upper and lower bounds on the population risk minus the irreducible error, that is, $\Theta(M^{-(a-1)} + N^{-(a-1)/a})$. In particular, we show that the variance error, which increases with $M$, is of strictly higher order compared to the other errors, thus disappearing from the risk bound. We attribute the nice empirical formula of the neural scaling law to the non-domination of the variance error, which ultimately is an effect of the implicit regularization of SGD.

Many directions remain open for future study. First, our work is limited to the linear model; it would be interesting to see whether similar scaling laws can be derived in more complex models, such as random feature models or two-layer networks. Second, we focus on one-pass SGD training, and it is unclear if similar results hold for other optimization methods like accelerated SGD or Adam. Additionally, from a technical perspective, many results in our work depend on the Gaussian assumption and the source condition of the data. Investigating how these assumptions can be relaxed would also be valuable.

## Acknowledgements

We gratefully acknowledge the support of the NSF for FODSI through grant DMS-2023505, of the NSF and the Simons Foundation for the Collaboration on the Theoretical Foundations of Deep Learning through awards DMS-2031883 and #814639, and of the ONR through MURI award N000142112431. JDL acknowledges support of the NSF CCF 2002272, NSF IIS 2107304, and NSF CAREER Award 2144994. SMK acknowledges a gift from the Chan Zuckerberg Initiative Foundation to establish the Kempner Institute for the Study of Natural and Artificial Intelligence; support from ONR under award N000142212377, and NSF under award IIS 2229881.

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

# Appendix

## Table of Contents

## A  Preliminary

In this section, we provide some preliminary discussions and a proof of Theorem 6.1. Concretely, in Section A.1 we discuss our data assumptions and introduce additional notations. In Section A.2, A.3 we derive intermediate results that contribute to the proof of Theorem 6.1. Finally, a complete proof of Theorem 6.1 is contained in Section A.4.

### A.1  Additional notations and comments on data assumptions

**Tensors.**  For matrices $\mathbf{A}$, $\mathbf{B}$, $\mathbf{C}$, $\mathbf{D}$, and $\mathbf{X}$ of appropriate shape, it holds that

$$(\mathbf{B}^\top \otimes \mathbf{A}) \circ \mathbf{X} = \mathbf{A}\mathbf{X}\mathbf{B},$$

and that

$$(\mathbf{D}^\top \otimes \mathbf{C}) \circ (\mathbf{B}^\top \otimes \mathbf{A}) \circ \mathbf{X} = \left((\mathbf{D}^\top\mathbf{B}^\top) \otimes (\mathbf{C}\mathbf{A})\right) \circ \mathbf{X}$$

$$= \mathbf{CAXBD}.$$

For simplicity, we denote

$$\mathbf{A}^{\otimes 2} := \mathbf{A} \otimes \mathbf{A}.$$

**Comments on Assumption 2, 3 and 4** Due to the rotational invariance of the Gaussian sketched matrix $\mathbf{S}$, throughout the appendix, we assume w.l.o.g. that the covariance of the input covariates $\mathbf{H}$ is diagonal with the $(i, i)$-th entry being the $i$-th eigenvalue. Specifically, Assumption 3 can be rewritten as

**Assumption 6** (Source condition). *Assume* $\mathbf{H} = (h_{ij})_{i,j \geq 1}$ *is a diagonal matrix with diagonal entries in non-increasing order, and* $\mathbf{w}^*$ *satisfies a prior such that*

$$\text{for } i \neq j, \ \mathbb{E}\mathbf{w}_i^* \mathbf{w}_j^* = 0; \ \text{ and for } i > 0, \ \mathbb{E}\lambda_i \mathbf{w}_i^{*2} \asymp i^{-b}, \ \text{ for some } b > 1.$$

Now that we assume $\mathbf{H}$ is diagonal. We make the following notations. Define

$$\mathbf{H}_{k^*:k^\dagger} := \text{diag}(\lambda_{k^*+1}, \dots, \lambda_{k^\dagger}) \in \mathbb{R}^{(k^\dagger - k^*)^2},$$

where $0 \leq k^* \leq k^\dagger$ are two integers, and we allow $k^\dagger = \infty$. For example,

$$\mathbf{H}_{0:k} = \text{diag}(\lambda_1, \dots, \lambda_i), \quad \mathbf{H}_{k:\infty} = \text{diag}(\lambda_{k+1}, \dots).$$

Similarly, for a vector $\mathbf{w} \in \mathbb{H}$, we have

$$\mathbf{w}_{k^*:k^\dagger} := (\mathbf{w}_{k^*+1}, \dots, \mathbf{w}_{k^\dagger}^*)^\top \in \mathbb{R}^{k^\dagger - k^*}.$$

## A.2 Approximation error

Recall the risk decomposition in (4),

$$\mathcal{R}_M(\mathbf{v}_N) = \underbrace{\min \mathcal{R}(\cdot)}_{\text{Irreducible}} + \underbrace{\min \mathcal{R}_M(\cdot) - \min \mathcal{R}(\cdot)}_{\text{Approx}} + \underbrace{\mathcal{R}_M(\mathbf{v}_N) - \min \mathcal{R}_M(\cdot)}_{\text{Excess}}.$$

**Lemma A.1** (Approximization error). *Conditional on the sketch matrix* $\mathbf{S}$*, the minimizer of* $\mathcal{R}_M(\mathbf{v})$ *is given by*

$$\mathbf{v}^* := (\mathbf{SHS}^\top)^{-1}\mathbf{SHw}^*,$$

*and the approximation error in (4) is*

$$\textsf{Approx} := \min \mathcal{R}_M(\cdot) - \min \mathcal{R}(\cdot)$$
$$= \left\| \left( \mathbf{I} - \mathbf{H}^{\frac{1}{2}}\mathbf{S}^\top \left( \mathbf{SHS}^\top \right)^{-1} \mathbf{SH}^{\frac{1}{2}} \right) \mathbf{H}^{\frac{1}{2}} \mathbf{w}^* \right\|^2. \tag{7}$$

*Moreover,* $\textsf{Approx} \leq \|\mathbf{w}^*\|_{\mathbf{H}}^2$ *almost surely over the randomness of* $\mathbf{S}$*.*

*Proof of Lemma A.1.* Recall that the risk

$$\mathcal{R}(\mathbf{w}) := \mathbb{E}(\langle \mathbf{x}, \mathbf{w} \rangle - y)^2$$

is a quadratic function and that $\mathbf{w}^*$ is the minimizer of $\mathcal{R}(\cdot)$, so we have

$$\left( \mathbb{E}\mathbf{x}^{\otimes 2} \right)\mathbf{w}^* = \mathbb{E}\mathbf{x}y \Leftrightarrow \mathbf{Hw}^* = \mathbb{E}\mathbf{x}y,$$

and

$$\mathcal{R}(\mathbf{w}) = \mathbb{E}(\langle \mathbf{x}, \mathbf{w} \rangle - \langle \mathbf{x}, \mathbf{w}^* \rangle)^2 + \mathcal{R}(\mathbf{w}^*)$$
$$= \|\mathbf{H}^{\frac{1}{2}}(\mathbf{w} - \mathbf{w}^*)\|^2 + \mathcal{R}(\mathbf{w}^*).$$

Recall that the risk in a restricted subspace

$$\mathcal{R}_M(\mathbf{v}) := \mathcal{R}(\mathbf{S}^\top \mathbf{v}) = \mathbb{E}(\langle \mathbf{Sx}, \mathbf{v} \rangle - y)^2$$

is also a quadratic function, so its minimizer is given by

$$\mathbf{v}^* = \left( \mathbb{E}(\mathbf{Sx})^{\otimes 2} \right)^{-1} \mathbb{E}\mathbf{Sx}y$$

$$= (\mathbf{SHS}^\top)^{-1} \mathbf{SHw}^*.$$

Therefore, the approximation error is

$$\begin{aligned}
\mathsf{Approx} &:= \mathcal{R}_M(\mathbf{v}^*) - \mathcal{R}(\mathbf{w}^*)\\
&= \mathcal{R}(\mathbf{S}^\top \mathbf{v}^*) - \mathcal{R}(\mathbf{w}^*)\\
&= \|\mathbf{H}^{\frac{1}{2}}(\mathbf{S}^\top \mathbf{v}^* - \mathbf{w}^*)\|^2\\
&= \|\mathbf{H}^{\frac{1}{2}}(\mathbf{S}^\top (\mathbf{SHS}^\top)^{-1} \mathbf{SHw}^* - \mathbf{w}^*)\|^2\\
&= \left\|\left(\mathbf{I} - \mathbf{H}^{\frac{1}{2}}\mathbf{S}^\top (\mathbf{SHS}^\top)^{-1} \mathbf{SH}^{\frac{1}{2}}\right)\mathbf{H}^{\frac{1}{2}}\mathbf{w}^*\right\|^2.
\end{aligned}$$

Finally, since

$$\left(\mathbf{I} - \mathbf{H}^{\frac{1}{2}}\mathbf{S}^\top (\mathbf{SHS}^\top)^{-1} \mathbf{SH}^{\frac{1}{2}}\right)^2 = \mathbf{I} - \mathbf{H}^{\frac{1}{2}}\mathbf{S}^\top (\mathbf{SHS}^\top)^{-1} \mathbf{SH}^{\frac{1}{2}} \preceq \mathbf{I},$$

it follows that $\mathsf{Approx} \le \|\mathbf{w}^*\|_{\mathbf{H}}^2$. □

### A.3  Bias-variance decomposition

The excess risk in (4) can be viewed as the SGD excess risk in an $M$-dimensional (misspecified) linear regression problem. We will utilize Corollary 3.4 in [45] to get a bias-variance decomposition of the excess risk. The following two lemmas check the related assumptions for Corollary 3.4 in [45] in our setup.

**Lemma A.2** (Hypercontractivity and the misspecified noise under sketched feature). *Suppose that Assumptions 5A and 5B hold. Conditioning on the sketch matrix $\mathbf{S}$, for every PSD matrix $\mathbf{A} \in \mathbb{R}^{M \times M}$, we have*

$$\mathbb{E}(\mathbf{Sx})^{\otimes 2}\mathbf{A}(\mathbf{Sx})^{\otimes 2} \preceq \alpha \operatorname{tr}(\mathbf{SHS}^\top \mathbf{A})\mathbf{SHS}^\top.$$

*Moreover, for the minimizer of $\mathcal{R}_M(\mathbf{v})$, that is, $\mathbf{v}^*$ defined in Lemma A.1, we have*

$$\mathbb{E}(y - \langle \mathbf{v}^*, \mathbf{Sx}\rangle)^2 (\mathbf{Sx})^{\otimes 2} \preceq 2(\sigma^2 + \alpha\|\mathbf{w}^*\|_{\mathbf{H}}^2)\mathbf{SHS}^\top.$$

*The expectation in the above is over $(\mathbf{x}, y)$.*

*Proof of Lemma A.2.*  The first part is a direct application of Assumption 5A:

$$\begin{aligned}
\mathbb{E}(\mathbf{Sx})^{\otimes 2}\mathbf{A}(\mathbf{Sx})^{\otimes 2} &= \mathbf{S}(\mathbb{E}\mathbf{xx}^\top (\mathbf{S}^\top \mathbf{AS})\mathbf{xx}^\top)\mathbf{S}^\top\\
&\preceq \mathbf{S}(\alpha \operatorname{tr}(\mathbf{HS}^\top \mathbf{AS})\mathbf{H})\mathbf{S}^\top\\
&= \alpha \operatorname{tr}(\mathbf{SHS}^\top \mathbf{A})\mathbf{SHS}^\top.
\end{aligned}$$

For the second part, we first show that

$$\begin{aligned}
\mathbb{E}(y - \langle \mathbf{v}^*, \mathbf{Sx}\rangle)^2 \mathbf{x}^{\otimes 2} &\preceq 2\mathbb{E}(y - \langle \mathbf{w}^*, \mathbf{x}\rangle)^2 \mathbf{x}^{\otimes 2} + 2\mathbb{E}\langle \mathbf{w}^* - \mathbf{S}^\top \mathbf{v}^*, \mathbf{x}\rangle^2 \mathbf{x}^{\otimes 2}\\
&\preceq 2\sigma^2 \mathbf{H} + 2\alpha\langle \mathbf{H}, (\mathbf{w}^* - \mathbf{S}^\top \mathbf{v}^*)^{\otimes 2}\rangle \mathbf{H},
\end{aligned}$$

where the last inequality is by Assumptions 5A and 5B. From the proof of Lemma A.1, we know that

$$\langle \mathbf{H}, (\mathbf{w}^* - \mathbf{S}^\top \mathbf{v}^*)^{\otimes 2}\rangle = \mathsf{Approx} \le \|\mathbf{w}^*\|_{\mathbf{H}}^2, \text{ almost surely.}$$

So we have

$$\mathbb{E}(y - \langle \mathbf{v}^*, \mathbf{Sx}\rangle)^2 \mathbf{x}^{\otimes 2} \preceq 2(\sigma^2 + \alpha\|\mathbf{w}^*\|_{\mathbf{H}}^2)\mathbf{H}.$$

Left and right multiplying both sides with $\mathbf{S}$ and $\mathbf{S}^\top$, we obtain the second claim. □

**Lemma A.3** (Gaussianity and well-specified noise under sketched features). *Suppose that Assumptions 1A and 1B hold. Conditional on the sketch matrix $\mathbf{S}$, we have*

$$\mathbf{Sx} \sim \mathcal{N}(0, \mathbf{SHS}^\top).$$

*Moreover, for the minimizer of $\mathcal{R}_M(\mathbf{v})$, that is, $\mathbf{v}^*$ defined in Lemma A.1, we have*

$$\mathbb{E}[y|\mathbf{Sx}] = \langle \mathbf{Sx}, \mathbf{v}^*\rangle, \quad \mathbb{E}(y - \langle \mathbf{Sx}, \mathbf{v}^*\rangle)^2 = \sigma^2 + \mathsf{Approx} \ge \sigma^2.$$

*Proof of Lemma A.3.* The first claim is a direct consequence of Assumption 1A.

For the second claim, by Assumption 1A and Lemma A.1, we have

$$
\begin{aligned}
\mathbb{E}[y|\mathbf{x}] &= \langle \mathbf{x}, \mathbf{w}^* \rangle \\
&= \langle \mathbf{x}, \mathbf{S}^\top \mathbf{v}^* \rangle + \langle \mathbf{x}, \mathbf{w}^* - \mathbf{S}^\top \mathbf{v}^* \rangle \\
&= \langle \mathbf{x}, \mathbf{S}^\top \mathbf{v}^* \rangle + \langle \mathbf{x}, [\mathbf{I} - (\mathbf{SHS}^\top)^{-1}\mathbf{SH}]\mathbf{w}^* \rangle \\
&= \langle \mathbf{H}^{-\frac{1}{2}}\mathbf{x}, \mathbf{H}^{\frac{1}{2}}\mathbf{S}^\top \mathbf{v}^* \rangle + \langle \mathbf{H}^{-\frac{1}{2}}\mathbf{x}, [\mathbf{I} - \mathbf{H}^{\frac{1}{2}}\mathbf{S}^\top(\mathbf{SHS}^\top)^{-1}\mathbf{SH}^{\frac{1}{2}}]\mathbf{H}^{\frac{1}{2}}\mathbf{w}^* \rangle \\
&= \langle \mathbf{SH}^{\frac{1}{2}}\mathbf{H}^{-\frac{1}{2}}\mathbf{x}, \mathbf{v}^* \rangle + \langle [\mathbf{I} - \mathbf{H}^{\frac{1}{2}}\mathbf{S}^\top(\mathbf{SHS}^\top)^{-1}\mathbf{SH}^{\frac{1}{2}}]\mathbf{H}^{-\frac{1}{2}}\mathbf{x}, \mathbf{H}^{\frac{1}{2}}\mathbf{w}^* \rangle. \quad (8)
\end{aligned}
$$

Notice that

$$
\mathbf{H}^{-\frac{1}{2}}\mathbf{x} \sim \mathcal{N}(0, \mathbf{I}),
$$

by Assumption 1A and that

$$
\mathbf{SH}^{\frac{1}{2}}\big[\mathbf{I} - \mathbf{H}^{\frac{1}{2}}\mathbf{S}^\top(\mathbf{SHS}^\top)^{-1}\mathbf{SH}^{\frac{1}{2}}\big] = 0,
$$

therefore

$$
\mathbf{Sx} = \mathbf{SH}^{\frac{1}{2}}\mathbf{H}^{-\frac{1}{2}}\mathbf{x} \text{ is independent of } \big[\mathbf{I} - \mathbf{H}^{\frac{1}{2}}\mathbf{S}^\top(\mathbf{SHS}^\top)^{-1}\mathbf{SH}^{\frac{1}{2}}\big]\mathbf{H}^{-\frac{1}{2}}\mathbf{x}.
$$

Taking expectation over the second random vector in (8), we find

$$
\mathbb{E}[y|\mathbf{Sx}] = \mathbb{E}\mathbb{E}[y|\mathbf{x}] = \langle \mathbf{SH}^{\frac{1}{2}}\mathbf{H}^{-\frac{1}{2}}\mathbf{x}, \mathbf{v}^* \rangle = \langle \mathbf{Sx}, \mathbf{v}^* \rangle.
$$

It remains to show

$$
\mathbb{E}(y - \langle \mathbf{Sx}, \mathbf{v}^* \rangle)^2 = \sigma^2 + \mathsf{Approx}.
$$

This follows from the proof of Lemma A.1. Specifically,

$$
\begin{aligned}
\mathbb{E}(y - \langle \mathbf{Sx}, \mathbf{v}^* \rangle)^2 &= \mathcal{R}(\mathbf{S}^\top \mathbf{v}^*) \\
&= \mathsf{Approx} + \mathcal{R}(\mathbf{w}^*) \\
&= \mathsf{Approx} + \sigma^2 \\
&\geq \sigma^2,
\end{aligned}
$$

where the second equality is by the definition of Approx and the third equality is by Assumption 1B. We have completed the proof. □

## A.4 Proof of Theorem 6.1

We now use the results in [44, 45] for SGD to obtain the following bias-variance decomposition on the excess risk.

**Theorem A.4** (Excess risk bounds)**.** *Consider the excess risk in (4) induced by the output of (SGD). Let*

$$
N_{\mathtt{eff}} := N/\log(N), \quad \mathtt{SNR} := (\|\mathbf{w}^*\|_{\mathbf{H}}^2 + \|\mathbf{v}_0\|_{\mathbf{SHS}^\top}^2)/\sigma^2.
$$

*Then conditioning on the sketch matrix $\mathbf{S}$, for any $\mathbf{w}^* \in \mathbb{H}$*

1. *Under Assumptions 5A and 5B, we have*

$$
\mathbb{E}\mathsf{Excess} \lesssim \left\| \prod_{t=1}^{N} (\mathbf{I} - \gamma_t \mathbf{SHS}^\top)(\mathbf{v}_0 - \mathbf{v}^*) \right\|_{\mathbf{SHS}^\top}^2 + (1 + \alpha\mathtt{SNR})\sigma^2 \cdot \frac{D_{\mathtt{eff}}}{N_{\mathtt{eff}}}
$$

*when $\gamma \lesssim \frac{1}{c\alpha\,\mathrm{tr}(\mathbf{SHS}^\top)}$ for some constant $c > 0$.*

2. *Under Assumptions 1A and 1B, we have*

$$
\mathbb{E}\mathsf{Excess} \gtrsim \left\| \prod_{t=1}^{N} (\mathbf{I} - \gamma_t \mathbf{SHS}^\top)(\mathbf{v}_0 - \mathbf{v}^*) \right\|_{\mathbf{SHS}^\top}^2 + \sigma^2 \cdot \frac{D_{\mathtt{eff}}}{N_{\mathtt{eff}}}
$$

*when $\gamma \lesssim \frac{1}{c\,\mathrm{tr}(\mathbf{SHS}^\top)}$ for some constant $c > 0$.*

*In both results, the expectation is over $(\mathbf{x}_t, y_t)_{t=1}^N$, and*

$$D_{\texttt{eff}} := \#\{\tilde{\lambda}_j \geq 1/(N_{\texttt{eff}}\gamma)\} + (N_{\texttt{eff}}\gamma)^2 \sum_{\tilde{\lambda}_j < 1/(N_{\texttt{eff}}\gamma)} \tilde{\lambda}_j^2,$$

*where $(\tilde{\lambda}_j)_{j=1}^M$ are eigenvalue of $\mathbf{SHS}^\top$.*

Theorem 6.1 follows immediately by Lemma A.1 and by setting $\mathbf{v}_0 = 0$ and plugging the definition of Bias($\mathbf{w}^*$) and Var into Theorem A.4.

*Proof of Theorem A.4.* This follows from Corollary 3.4 in [45] for a linear regression problem with population data given by $(\mathbf{Sx}, y)$. Note that the data covariance becomes $\mathbf{SHS}^\top$ and the optimal model parameter becomes $\mathbf{v}^*$.

For the upper bound, Lemma A.2 verifies Assumptions 1A and 2 in [45], with the noise level being
$$\tilde{\sigma}^2 = 2(\sigma^2 + \alpha\|\mathbf{w}^*\|_{\mathbf{H}}^2).$$
Then we can apply the upper bound in Corollary 3.4 in [45] (setting their index set $\mathbb{K} = \emptyset$) to get

$$\mathbb{E}\text{Excess} \lesssim \left\| \prod_{t=1}^N (\mathbf{I} - \gamma_t \mathbf{SHS}^\top)(\mathbf{v}_0 - \mathbf{v}^*) \right\|_{\mathbf{SHS}^\top}^2 + (\|\mathbf{v}^* - \mathbf{v}_0\|_{\mathbf{SHS}^\top}^2 + \tilde{\sigma}^2)\frac{D_{\texttt{eff}}}{N_{\texttt{eff}}}.$$

We verify that

$$\begin{aligned}
\|\mathbf{v}^* - \mathbf{v}_0\|_{\mathbf{SHS}^\top}^2 &\leq 2\|\mathbf{H}^{\frac{1}{2}}\mathbf{S}^\top \mathbf{v}^*\|^2 + 2\|\mathbf{v}_0\|_{\mathbf{SHS}^\top}^2 \\
&= 2\|\mathbf{H}^{\frac{1}{2}}\mathbf{S}^\top(\mathbf{SHS}^\top)^{-1}\mathbf{SHw}^*\|^2 + 2\|\mathbf{v}_0\|_{\mathbf{SHS}^\top}^2 \\
&\leq 2\|\mathbf{H}^{\frac{1}{2}}\mathbf{w}^*\|^2 + 2\|\mathbf{v}_0\|_{\mathbf{SHS}^\top}^2 \\
&= 2\|\mathbf{w}^*\|_{\mathbf{H}}^2 + 2\|\mathbf{v}_0\|_{\mathbf{SHS}^\top}^2,
\end{aligned}$$

which implies that

$$\begin{aligned}
(\|\mathbf{v}^* - \mathbf{v}_0\|_{\mathbf{SHS}^\top}^2 + \tilde{\sigma}^2) &\leq 2\|\mathbf{w}^*\|_{\mathbf{H}}^2 + 2\|\mathbf{v}_0\|_{\mathbf{SHS}^\top}^2 + 2(\sigma^2 + \alpha\|\mathbf{w}^*\|_{\mathbf{H}}^2) \\
&\lesssim (1 + \alpha\text{SNR})\sigma^2.
\end{aligned}$$

Substituting, we get the upper bound.

For the lower bound, Lemma A.3 shows $\mathbf{Sx}$ is Gaussian, therefore it satisfies Assumption 1B in Wu et al. [45] with $\beta = 1$. Besides, Lemma A.3 shows that the linear regression problem is well-specified, with the noise level being
$$\tilde{\sigma}^2 = \sigma^2 + \text{Approx} \geq \sigma^2.$$
Although the lower bound in Corollary 3.4 in Wu et al. [45] is stated for Gaussian additive noise (see their Assumption 2'), it is easy to check that the lower bound holds for any well-specified noise as described by Lemma A.3. Using the lower bound in Corollary 3.4 in Wu et al. [45], we obtain

$$\mathbb{E}\text{Excess} \gtrsim \left\| \prod_{t=1}^N (\mathbf{I} - \gamma_t \mathbf{SHS}^\top)(\mathbf{v}_0 - \mathbf{v}^*) \right\|_{\mathbf{SHS}^\top}^2 + \tilde{\sigma}^2 \frac{D_{\texttt{eff}}}{N_{\texttt{eff}}}.$$

Plugging in $\tilde{\sigma}^2 \geq \sigma^2$ gives the desired lower bound. □

### A.5 Proofs of Lemma 6.2, Theorem 6.3 and 6.4

Lemma 6.2 is proved in Lemma G.4. Theorem 6.3 follows from Lemma C.1 and D.1. Theorem 6.4 follows from Lemma C.2 and D.2.

## B Proofs in Section 4

### B.1 Proof of Theorem 4.1

**Proof of part 1.** By Assumption 1B and the definition of $\mathcal{R}(\cdot)$, we have

$$\begin{aligned}
\mathcal{R}(\mathbf{w}) = \mathbb{E}(\langle \mathbf{x}, \mathbf{w} \rangle - y)^2 &= \mathbb{E}(\langle \mathbf{x}, \mathbf{w} \rangle - \mathbb{E}[y \mid \mathbf{x}])^2 + \mathbb{E}(y - \mathbb{E}[y \mid \mathbf{x}])^2 \\
&= \mathbb{E}(\langle \mathbf{x}, \mathbf{w} \rangle - \langle \mathbf{x}, \mathbf{w}^* \rangle)^2 + \sigma^2 \geq \sigma^2.
\end{aligned}$$

Note that the equality holds if and only if $\mathbf{w} = \mathbf{w}^*$. Therefore we have $\min \mathcal{R}(\cdot) = \mathcal{R}(\mathbf{w}^*) = \sigma^2$.

**Proof of part 2.** Part 2 of Theorem 4.1 follows immediately from Lemma C.2.

**Proof of part 3.** We choose $\mathsf{Bias}(\mathbf{w}^*), \mathsf{Var}$ as defined in Eq. (5) and (6) and let $\mathsf{Bias} := \mathbb{E}_{\mathbf{w}^*}\mathsf{Bias}(\mathbf{w}^*)$. Part 3 of Theorem 4.1 follows directly from the decomposition of the excess risk in Theorem 6.1 (note that $\mathbb{E}\|\mathbf{w}^*\|_{\mathbf{H}}^2/\sigma^2 \lesssim 1$), and the matching bounds in Lemma D.3 and E.1.

It remains to verify the stepsize assumption required in Lemma D.3. Since we have from Lemma G.4 that

$$\frac{1}{\mathrm{tr}(\mathbf{SHS}^\top)} = \frac{1}{\sum_{i=1}^M \tilde{\lambda}_i} \geq \frac{c_1}{\sum_{i=1}^M \lambda_i} \geq \frac{c_2}{\sum_{i=1}^M i^{-a}} \geq c_3$$

for some $a$-dependent constants $c_1, c_2, c_3 > 0$ with probability at least $1 - e^{-\Omega(M)}$, it follows that for any constant $c > 1$, we can choose $\gamma \leq c_0$ for some $a$-dependent $c_0$ such that $\gamma \leq \frac{1}{c\,\mathrm{tr}(\mathbf{SHS}^\top)}$. Therefore, we have verified the stepsize assumption.

Finally, the last claim in Theorem 4.1 follows directly from combining the previous three parts and Theorem 6.1, noting $\sigma^2 \lesssim 1$. and

$$\mathsf{Var} \asymp \frac{\min\{M,\ (N_{\mathrm{eff}}\gamma)^{1/a}\}}{N_{\mathrm{eff}}} \lesssim \frac{(N_{\mathrm{eff}}\gamma)^{1/a}}{N_{\mathrm{eff}}} \lesssim (N_{\mathrm{eff}}\gamma)^{1/a-1} \lesssim \mathsf{Bias} + \mathsf{Approx}$$

under the stepsize assumption $\gamma \lesssim 1$. Here the hidden constants may depend on $a$.

## B.2 Proof of Theorem 4.2

Similar to the proof of Theorem 4.1, we have $\min \mathcal{R}(\cdot) = \sigma^2$ under Assumption 1B. Moreover, by Lemma C.5, D.4 and E.1, we have with probability at least $1 - e^{-\Omega(M)}$ that

$$\mathbb{E}_{\mathbf{w}^*}\mathsf{Approx} \asymp M^{1-b},$$
$$\mathsf{Bias} \lesssim \max\{M^{1-b},\ (N_{\mathrm{eff}}\gamma)^{(1-b)/a}\},$$
$$\mathsf{Bias} \gtrsim (N_{\mathrm{eff}}\gamma)^{(1-b)/a} \text{ when } (N_{\mathrm{eff}}\gamma)^{1/a} \leq M/3,$$
$$\mathsf{Var} \asymp \min\{M,\ (N_{\mathrm{eff}}\gamma)^{1/a}\}/N_{\mathrm{eff}},$$

when the stepsize $\gamma \leq c$ for some $a$-dependent constant $c > 0$. Here the hidden constants in the bounds may depend only on $(a, b)$. Combining the bounds on $\mathsf{Approx}, \mathsf{Bias}, \mathsf{Var}$ and noting

$$\mathsf{Var} \asymp \frac{\min\{M,\ (N_{\mathrm{eff}}\gamma)^{1/a}\}}{N_{\mathrm{eff}}} \lesssim \frac{(N_{\mathrm{eff}}\gamma)^{1/a}}{N_{\mathrm{eff}}} \lesssim (N_{\mathrm{eff}}\gamma)^{(1-b)/a} \lesssim \mathsf{Bias} + \mathsf{Approx}$$

yields Theorem 4.2. Here in the second inequality, we use the assumption $b \leq a$.

## B.3 Proof of Theorem 4.3

Similar to the proof of Theorem 4.1, we have $\min \mathcal{R}(\cdot) = \sigma^2$ under Assumption 1B. Notice that we have $\gamma \lesssim 1$ implies $\gamma \lesssim 1/(\mathrm{tr}(\mathbf{SHS}^\top))$ with probability at least $1 - e^{-\Omega(M)}$ by Lemma G.6. It follows from Lemma C.6, D.5 and E.2 that

$$\mathbb{E}_{\mathbf{w}^*}\mathsf{Approx} \asymp \log^{1-a} M,$$
$$\mathsf{Bias} \lesssim \max\{\log^{1-a} M,\ \log^{1-a}(N_{\mathrm{eff}}\gamma)\},$$
$$\mathsf{Bias} \gtrsim \log^{1-a}(N_{\mathrm{eff}}\gamma) \text{ when } (N_{\mathrm{eff}}\gamma)^{1/a} \leq M^c \text{ for some small constant } c > 0,$$
$$\mathsf{Var} \asymp \frac{\min\{M,\ (N_{\mathrm{eff}}\gamma)/\log^a(N_{\mathrm{eff}}\gamma)\}}{N_{\mathrm{eff}}} \lesssim \frac{(N_{\mathrm{eff}}\gamma)/\log^a(N_{\mathrm{eff}}\gamma)}{N_{\mathrm{eff}}\gamma} = \log^{-a}(N_{\mathrm{eff}}\gamma)$$

with probability at least $1 - e^{-\Omega(M)}$ when the stepsize $\gamma \leq c$ for some $a$-dependent constant $c > 0$. Since $\mathsf{Var} \lesssim \mathsf{Bias}$ and $\log^{1-a}(N_{\mathrm{eff}}\gamma) \lesssim \log^{1-a} M$ when $(N_{\mathrm{eff}}\gamma)^{1/a} \gtrsim M^c$, putting the bounds together gives Theorem 4.3.

# C Approximation error

In this section, we derive upper and lower bounds for the approximation error in (4) (and 7). We will also show that the upper and lower bounds match up to constant factors in several examples.

## C.1 An upper bound

**Lemma C.1** (An upper bound on the approximation error). *Given any $k \leq d$ such that $r(\mathbf{H}) \geq k+M$, the approximation error in (4) (and 7) satisfies*

$$\mathsf{Approx} \lesssim \|\mathbf{w}^*_{k:\infty}\|^2_{\mathbf{H}_{k:\infty}} + \langle [\mathbf{H}^{-1}_{0:k} + \mathbf{S}^\top_{0:k}\mathbf{A}^{-1}_k \mathbf{S}_{0:k}]^{-1}, \mathbf{w}^*_{0:k}\mathbf{w}^{*}_{0:k}{}^\top \rangle$$

*almost surely, where $\mathbf{A}_k := \mathbf{S}_{k:\infty}\mathbf{H}_{k:\infty}\mathbf{S}^\top_{k:\infty}$. If in addition $k \leq M/2$, then with probability $1 - e^{-\Omega(M)}$*

$$\mathsf{Approx} \lesssim \|\mathbf{w}^*_{k:\infty}\|^2_{\mathbf{H}_{k:\infty}} + \Big( \frac{\sum_{i>k}\lambda_i}{M} + \lambda_{k+1} + \sqrt{\frac{\sum_{i>k}\lambda^2_i}{M}} \Big) \|\mathbf{w}^*_{0:k}\|^2,$$

*where $(\lambda_i)^p_{i=1}$ are eigenvalues of $\mathbf{H}$ in non-increasing order.*

*Proof of Lemma C.1.* Write the singular value decomposition $\mathbf{H} = \mathbf{U}\mathbf{\Lambda}\mathbf{U}^\top$, where $\mathbf{\Lambda} := \mathrm{diag}\{\lambda_1, \lambda_2, \ldots\}$ with $\lambda_1 \geq \lambda_2 \geq \ldots \geq 0$ and $\mathbf{U}\mathbf{U}^\top = \mathbf{I}$. Define $\tilde{\mathbf{S}} := \mathbf{S}\mathbf{U}$, $\tilde{\mathbf{w}}^* := \mathbf{U}^\top \mathbf{w}^*$. Then by Lemma A.1 the approximation error $\mathsf{Approx} = \mathsf{Approx}(\mathbf{S}, \mathbf{H}, \mathbf{w}^*)$ satisfies

$$\mathsf{Approx}(\mathbf{S}, \mathbf{H}, \mathbf{w}^*) = \left\| \left( \mathbf{I} - \mathbf{H}^{\frac{1}{2}}\mathbf{S}^\top (\mathbf{S}\mathbf{H}\mathbf{S}^\top)^{-1}\mathbf{S}\mathbf{H}^{\frac{1}{2}} \right) \mathbf{H}^{\frac{1}{2}}\mathbf{w}^* \right\|^2$$

$$= \left\| \left( \mathbf{I} - \mathbf{U}\mathbf{\Lambda}^{\frac{1}{2}}\tilde{\mathbf{S}}^\top (\tilde{\mathbf{S}}\mathbf{\Lambda}\tilde{\mathbf{S}}^\top)^{-1}\mathbf{S}\mathbf{\Lambda}^{\frac{1}{2}}\mathbf{U}^\top \right) \mathbf{U}\mathbf{\Lambda}^{\frac{1}{2}}\mathbf{U}^\top\mathbf{w}^* \right\|^2$$

$$= \left\| \mathbf{U} \left( \mathbf{I} - \mathbf{\Lambda}^{\frac{1}{2}}\tilde{\mathbf{S}}^\top (\tilde{\mathbf{S}}\mathbf{\Lambda}\tilde{\mathbf{S}}^\top)^{-1}\mathbf{S}\mathbf{\Lambda}^{\frac{1}{2}} \right) \mathbf{\Lambda}^{\frac{1}{2}}\mathbf{U}^\top\mathbf{w}^* \right\|^2$$

$$= \left\| \left( \mathbf{I} - \mathbf{\Lambda}^{\frac{1}{2}}\tilde{\mathbf{S}}^\top (\tilde{\mathbf{S}}\mathbf{\Lambda}\tilde{\mathbf{S}}^\top)^{-1}\mathbf{S}\mathbf{\Lambda}^{\frac{1}{2}} \right) \mathbf{\Lambda}^{\frac{1}{2}}\tilde{\mathbf{w}}^* \right\|^2 = \mathsf{Approx}(\tilde{\mathbf{S}}, \mathbf{\Lambda}, \tilde{\mathbf{w}}^*).$$

Since $\tilde{\mathbf{S}} \overset{d}{=} \mathbf{S}$ by rotational invariance of standard gaussian variables, it suffices to analyze the case where $\mathbf{H} = \mathbf{\Lambda}$ is a diagonal matrix, as the results may transfer to general $\mathbf{H}$ by replacing $\tilde{\mathbf{w}}^*$ with $\mathbf{w}^*$.

Therefore, from now on we assume w.l.o.g. that $\mathbf{H}$ is a diagonal matrix with non-increasing diagonal entries. Define $\mathbf{A} := \mathbf{S}\mathbf{H}\mathbf{S}^\top$.

By definition of $\mathsf{Approx}$, we have

$$\mathsf{Approx} = \left\| \left( \mathbf{I} - \mathbf{H}^{\frac{1}{2}}\mathbf{S}^\top (\mathbf{S}\mathbf{H}\mathbf{S}^\top)^{-1}\mathbf{S}\mathbf{H}^{\frac{1}{2}} \right) \mathbf{H}^{\frac{1}{2}}\mathbf{w}^* \right\|^2$$

$$= \langle [\mathbf{H}^{1/2}\mathbf{S}^\top\mathbf{A}^{-1}\mathbf{S}\mathbf{H}^{1/2} - \mathbf{I}_p]^{\otimes 2}, \mathbf{H}^{1/2}\mathbf{w}^*\mathbf{w}^{*\top}\mathbf{H}^{1/2} \rangle.$$

Moreover, for any $k \in [p]$

$$\mathbf{H}^{1/2}\mathbf{S}^\top\mathbf{A}^{-1}\mathbf{S}\mathbf{H}^{1/2} - \mathbf{I}_p = \begin{pmatrix} \mathbf{H}^{1/2}_{0:k}\mathbf{S}^\top_{0:k} \\ \mathbf{H}^{1/2}_{k:\infty}\mathbf{S}^\top_{k:\infty} \end{pmatrix} \mathbf{A}^{-1} \begin{pmatrix} \mathbf{S}_{k:\infty}\mathbf{H}^{1/2}_{k:\infty} & \mathbf{S}_{k:\infty}\mathbf{H}^{1/2}_{k:\infty} \end{pmatrix} - \mathbf{I}_p$$

$$= \begin{pmatrix} \mathbf{H}^{1/2}_{0:k}\mathbf{S}^\top_{0:k}\mathbf{A}^{-1}\mathbf{S}_{0:k}\mathbf{H}^{1/2}_{0:k} - \mathbf{I}_k & \mathbf{H}^{1/2}_{0:k}\mathbf{S}^\top_{0:k}\mathbf{A}^{-1}\mathbf{S}_{k:\infty}\mathbf{H}^{1/2}_{k:\infty} \\ \mathbf{H}^{1/2}_{k:\infty}\mathbf{S}^\top_{k:\infty}\mathbf{A}^{-1}\mathbf{S}_{0:k}\mathbf{H}^{1/2}_{0:k} & \mathbf{H}^{1/2}_{k:\infty}\mathbf{S}^\top_{k:\infty}\mathbf{A}^{-1}\mathbf{S}_{k:\infty}\mathbf{H}^{1/2}_{k:\infty} - \mathbf{I}_{d-k} \end{pmatrix}$$

$$=: \begin{pmatrix} \mathsf{U} & \mathsf{V} \\ \mathsf{V}^\top & \mathsf{W} \end{pmatrix} \tag{9}$$

Therefore

$$[\mathbf{H}^{1/2}\mathbf{S}^\top\mathbf{A}^{-1}\mathbf{S}\mathbf{H}^{1/2} - \mathbf{I}_p]^{\otimes 2} = \begin{pmatrix} \mathsf{U}^2 + \mathsf{V}\mathsf{V}^\top & \mathsf{U}\mathsf{V} + \mathsf{V}\mathsf{W} \\ \mathsf{V}^\top\mathsf{U} + \mathsf{W}\mathsf{V}^\top & \mathsf{W}^2 + \mathsf{V}^\top\mathsf{V} \end{pmatrix} \preceq 2 \begin{pmatrix} \mathsf{U}^2 + \mathsf{V}\mathsf{V}^\top & \mathbf{0} \\ \mathbf{0} & \mathsf{W}^2 + \mathsf{V}^\top\mathsf{V} \end{pmatrix},$$

and hence

$$\mathsf{Approx} \leq 2 \left\langle \begin{pmatrix} \mathsf{U}^2 + \mathsf{V}\mathsf{V}^\top & \mathbf{0} \\ \mathbf{0} & \mathsf{W}^2 + \mathsf{V}^\top\mathsf{V} \end{pmatrix}, \mathbf{H}^{1/2}\mathbf{w}^*\mathbf{w}^{*\top}\mathbf{H}^{1/2} \right\rangle$$

$$= 2\langle \mathsf{U}^2 + \mathsf{V}\mathsf{V}^\top, \mathbf{H}_{0:k}^{1/2}\mathbf{w}_{*,0:k}\mathbf{w}_{*,0:k}^\top\mathbf{H}_{0:k}^{1/2}\rangle + 2\langle \mathsf{W}^2 + \mathsf{V}^\top\mathsf{V}, \mathbf{H}_{k:\infty}^{1/2}\mathbf{w}_{*,k:\infty}\mathbf{w}_{*,k:\infty}^\top\mathbf{H}_{k:\infty}^{1/2}\rangle.$$

We claim the following results which we will prove at the end of the proof.

$$\langle \mathsf{W}^2 + \mathsf{V}^\top\mathsf{V}, \mathbf{H}_{k:\infty}^{1/2}\mathbf{w}_{*,k:\infty}\mathbf{w}_{*,k:\infty}^\top\mathbf{H}_{k:\infty}^{1/2}\rangle \le \|\mathbf{w}_{k:\infty}^*\|_{\mathbf{H}_{k:\infty}}^2, \tag{10}$$

$$\langle \mathsf{U}^2 + \mathsf{V}\mathsf{V}^\top, \mathbf{H}_{0:k}^{1/2}\mathbf{w}_{*,0:k}\mathbf{w}_{*,0:k}^\top\mathbf{H}_{0:k}^{1/2}\rangle = \langle [\mathbf{H}_{0:k}^{-1} + \mathbf{S}_{0:k}^\top\mathbf{A}_k^{-1}\mathbf{S}_{0:k}]^{-1}, \mathbf{w}_{0:k}^*\mathbf{w}_{0:k}^{*\top}\rangle. \tag{11}$$

Note that in claim (11) the inverse $\mathbf{A}_k^{-1}$ exists almost surely since $r(\mathbf{H}_{k:\infty}) \ge r(\mathbf{H}) - k \ge M$ by our assumption and $\mathbf{S}_{k:\infty} \in \mathbb{R}^{M\times(d-k)}$ is a random gaussian projection onto $\mathbb{R}^M$. First part of Lemma C.1 follows immediately from combining claim (10) and (11).

To prove the second part of Lemma C.1, first note that with probability $1 - e^{-\Omega(M)}$ we have

$$\mu_{\min}(\mathbf{A}_k^{-1}) = \|\mathbf{A}_k\|^{-1} \ge c/\left(\frac{\sum_{i>k}\lambda_i}{M} + \lambda_{k+1} + \sqrt{\frac{\sum_{i>k}\lambda_i^2}{M}}\right)$$

forc some constant $c > 0$ by Lemma G.2. Moreover, by the concentration of the Gaussian variance matrix (see e.g., Theorem 6.1 in [41]), we have $\mathbf{S}_{0:k}^\top\mathbf{S}_{0:k} \succeq \mathbf{I}_k/5$ with probability $1 - e^{-\Omega(M)}$ when $M/k \ge 2$. Combining the last two arguments, we obtain

$$\mathbf{S}_{0:k}^\top\mathbf{A}_k^{-1}\mathbf{S}_{0:k} \succeq c\mathbf{S}_{0:k}^\top\mathbf{S}_{0:k}/\left(\frac{\sum_{i>k}\lambda_i}{M} + \lambda_{k+1} + \sqrt{\frac{\sum_{i>k}\lambda_i^2}{M}}\right)$$

$$\gtrsim \mathbf{I}_k/\left(\frac{\sum_{i>k}\lambda_i}{M} + \lambda_{k+1} + \sqrt{\frac{\sum_{i>k}\lambda_i^2}{M}}\right),$$

and therefore

$$\langle [\mathbf{H}_{0:k}^{-1} + \mathbf{S}_{0:k}^\top\mathbf{A}_k^{-1}\mathbf{S}_{0:k}]^{-1}, \mathbf{w}_{0:k}^*\mathbf{w}_{0:k}^{*\top}\rangle \le \langle [\mathbf{S}_{0:k}^\top\mathbf{A}_k^{-1}\mathbf{S}_{0:k}]^{-1}, \mathbf{w}_{0:k}^*\mathbf{w}_{0:k}^{*\top}\rangle$$

$$\le \|[\mathbf{H}_{0:k}^{-1} + \mathbf{S}_{0:k}^\top\mathbf{A}_k^{-1}\mathbf{S}_{0:k}]^{-1}\|\|\mathbf{w}_{0:k}^*\|^2$$

$$\lesssim \left(\frac{\sum_{i>k}\lambda_i}{M} + \lambda_{k+1} + \sqrt{\frac{\sum_{i>k}\lambda_i^2}{M}}\right)\|\mathbf{w}_{0:k}^*\|^2 \tag{12}$$

with probability $1 - e^{-\Omega(M)}$. Combining Eq. (12) with the first part of Lemma C.1 completes the proof.

**Proof of claim** (10)   Note that

$$-\mathbf{I}_{d-k} \preceq \mathsf{W} = \mathbf{H}_{k:\infty}^{1/2}\mathbf{S}_{k:\infty}^\top\mathbf{A}^{-1}\mathbf{S}_{k:\infty}\mathbf{H}_{k:\infty}^{1/2} - \mathbf{I}_{d-k}$$

$$= \mathbf{H}_{k:\infty}^{1/2}\mathbf{S}_{k:\infty}^\top(\mathbf{S}_{0:k}\mathbf{H}_{0:k}\mathbf{S}_{0:k}^\top + \mathbf{S}_{k:\infty}\mathbf{H}_{k:\infty}\mathbf{S}_{k:\infty}^\top)^{-1}\mathbf{S}_{k:\infty}\mathbf{H}_{k:\infty}^{1/2} - \mathbf{I}_{d-k}$$

$$\preceq \mathbf{H}_{k:\infty}^{1/2}\mathbf{S}_{k:\infty}^\top(\mathbf{S}_{k:\infty}\mathbf{H}_{k:\infty}\mathbf{S}_{k:\infty}^\top)^{-1}\mathbf{S}_{k:\infty}\mathbf{H}_{k:\infty}^{1/2} - \mathbf{I}_{d-k} \preceq \mathbf{0}_{d-k},$$

where the last inequality uses the fact that the norm of projection matrices is no greater than one. Therefore, we have $\|\mathsf{W}\|_2 \le 1$. Now, it remains to show

$$\mathsf{W}^2 + \mathsf{V}^\top\mathsf{V} = -\mathsf{W}, \tag{13}$$

as claim (10) is a direct consequence of Eq. (13) and the fact that $\|\mathsf{W}\| \le 1$.

By definition of $\mathsf{W}$ in Eq. (9), we have

$$\mathsf{W}^2 = (\mathbf{H}_{k:\infty}^{1/2}\mathbf{S}_{k:\infty}^\top\mathbf{A}^{-1}\mathbf{S}_{k:\infty}\mathbf{H}_{k:\infty}^{1/2} - \mathbf{I}_{d-k})^2$$

$$= \mathbf{I}_{d-k} - 2\mathbf{H}_{k:\infty}^{1/2}\mathbf{S}_{k:\infty}^\top\mathbf{A}^{-1}\mathbf{S}_{k:\infty}\mathbf{H}_{k:\infty}^{1/2} + \mathbf{H}_{k:\infty}^{1/2}\mathbf{S}_{k:\infty}^\top\mathbf{A}^{-1}\mathbf{S}_{k:\infty}\mathbf{H}_{k:\infty}\mathbf{S}_{k:\infty}^\top\mathbf{A}^{-1}\mathbf{S}_{k:\infty}\mathbf{H}_{k:\infty}^{1/2}$$

$$= \mathbf{I}_{d-k} - 2\mathbf{H}_{k:\infty}^{1/2}\mathbf{S}_{k:\infty}^\top\mathbf{A}^{-1}\mathbf{S}_{k:\infty}\mathbf{H}_{k:\infty}^{1/2} + \mathbf{H}_{k:\infty}^{1/2}\mathbf{S}_{k:\infty}^\top\mathbf{A}^{-1}\mathbf{A}_k\mathbf{A}^{-1}\mathbf{S}_{k:\infty}\mathbf{H}_{k:\infty}^{1/2}.$$

By definition of $\mathsf{V}$ in Eq. (9), we have

$$\mathsf{V}^\top\mathsf{V} = \mathbf{H}_{k:\infty}^{1/2}\mathbf{S}_{k:\infty}^\top\mathbf{A}^{-1}(\mathbf{S}_{0:k}\mathbf{H}_{0:k}\mathbf{S}_{0:k}^\top)\mathbf{A}^{-1}\mathbf{S}_{k:\infty}\mathbf{H}_{k:\infty}^{1/2}.$$

Since $\mathbf{S}_{0:k}\mathbf{H}_{0:k}\mathbf{S}_{0:k}^\top + \mathbf{A}_k = \mathbf{A}$, it follows that

$$\mathsf{W}^2 + \mathsf{V}^\top\mathsf{V} = \mathbf{I}_{d-k} - 2\mathbf{H}_{k:\infty}^{1/2}\mathbf{S}_{k:\infty}^\top\mathbf{A}^{-1}\mathbf{S}_{k:\infty}\mathbf{H}_{k:\infty}^{1/2} + \mathbf{H}_{k:\infty}^{1/2}\mathbf{S}_{k:\infty}^\top\mathbf{A}^{-1}\mathbf{A}\mathbf{A}^{-1}\mathbf{S}_{k:\infty}\mathbf{H}_{k:\infty}^{1/2}$$

$$= \mathbf{I}_{d-k} - \mathbf{H}_{k:\infty}^{1/2}\mathbf{S}_{k:\infty}^\top\mathbf{A}^{-1}\mathbf{S}_{k:\infty}\mathbf{H}_{k:\infty}^{1/2} = -\mathsf{W}.$$

**Proof of claim** (11)    It suffices to show $\mathsf{U}^2 + \mathsf{V}^\top \mathsf{V} = [\mathbf{H}_{0:k}^{-1} + \mathbf{S}_{0:k}^\top \mathbf{A}_k^{-1} \mathbf{S}_{0:k}]^{-1}$. Using the definition of $\mathsf{U}$ in Eq. (9), we obtain

$$
\begin{aligned}
\mathsf{U} &= \mathbf{H}_{0:k}^{1/2} \mathbf{S}_{0:k}^\top \mathbf{A}^{-1} \mathbf{S}_{0:k} \mathbf{H}_{0:k}^{1/2} - \mathbf{I}_k \\
&= \mathbf{H}_{0:k}^{1/2} \mathbf{S}_{0:k}^\top \mathbf{A}_k^{-1} \mathbf{S}_{0:k} \mathbf{H}_{0:k}^{1/2} - \mathbf{H}_{0:k}^{1/2} \mathbf{S}_{0:k}^\top \mathbf{A}_k^{-1} \mathbf{S}_{0:k} [\mathbf{H}_{0:k}^{-1} + \mathbf{S}_{0:k}^\top \mathbf{A}_k^{-1} \mathbf{S}_{0:k}]^{-1} \mathbf{S}_{0:k}^\top \mathbf{A}_k^{-1} \mathbf{S}_{0:k} \mathbf{H}_{0:k}^{1/2} - \mathbf{I}_k \\
&= \mathbf{H}_{0:k}^{1/2} \mathbf{S}_{0:k}^\top \mathbf{A}_k^{-1} \mathbf{S}_{0:k} [\mathbf{H}_{0:k}^{-1} + \mathbf{S}_{0:k}^\top \mathbf{A}_k^{-1} \mathbf{S}_{0:k}]^{-1} \mathbf{H}_{0:k}^{-1} \mathbf{H}_{0:k}^{1/2} - \mathbf{I}_k,
\end{aligned}
$$

where the second line uses Woodbury's matrix identity, namely

$$
\mathbf{A}^{-1} = [\mathbf{S}_{0:k} \mathbf{H}_{0:k} \mathbf{S}_{0:k}^\top + \mathbf{A}_k]^{-1} = \mathbf{A}_k^{-1} - \mathbf{A}_k^{-1} \mathbf{S}_{0:k} [\mathbf{H}_{0:k}^{-1} + \mathbf{S}_{0:k}^\top \mathbf{A}_k^{-1} \mathbf{S}_{0:k}]^{-1} \mathbf{S}_{0:k}^\top \mathbf{A}_k^{-1}.
$$

Continuing the calculation of $\mathsf{U}$, we have

$$
\begin{aligned}
\mathsf{U} &= \mathbf{H}_{0:k}^{1/2} \mathbf{S}_{0:k}^\top \mathbf{A}_k^{-1} \mathbf{S}_{0:k} [\mathbf{H}_{0:k}^{-1} + \mathbf{S}_{0:k}^\top \mathbf{A}_k^{-1} \mathbf{S}_{0:k}]^{-1} \mathbf{H}_{0:k}^{-1/2} - \mathbf{I}_k \\
&= \mathbf{H}_{0:k}^{1/2} (\mathbf{S}_{0:k}^\top \mathbf{A}_k^{-1} \mathbf{S}_{0:k} [\mathbf{H}_{0:k}^{-1} + \mathbf{S}_{0:k}^\top \mathbf{A}_k^{-1} \mathbf{S}_{0:k}]^{-1} - \mathbf{I}_k) \mathbf{H}_{0:k}^{-1/2} \\
&= -\mathbf{H}_{0:k}^{-1/2} [\mathbf{H}_{0:k}^{-1} + \mathbf{S}_{0:k}^\top \mathbf{A}_k^{-1} \mathbf{S}_{0:k}]^{-1} \mathbf{H}_{0:k}^{-1/2}.
\end{aligned}
$$

Therefore,

$$
\mathsf{U}^2 = \mathbf{H}_{0:k}^{-1/2} [\mathbf{H}_{0:k}^{-1} + \mathbf{S}_{0:k}^\top \mathbf{A}_k^{-1} \mathbf{S}_{0:k}]^{-1} \mathbf{H}_{0:k}^{-1} [\mathbf{H}_{0:k}^{-1} + \mathbf{S}_{0:k}^\top \mathbf{A}_k^{-1} \mathbf{S}_{0:k}]^{-1} \mathbf{H}_{0:k}^{-1/2}. \tag{14}
$$

Since

$$
\begin{aligned}
\mathbf{H}_{0:k}^{1/2} \mathbf{S}_{0:k}^\top \mathbf{A}^{-1} &= \mathbf{H}_{0:k}^{1/2} \mathbf{S}_{0:k}^\top \mathbf{A}_k^{-1} - \mathbf{H}_{0:k}^{1/2} \mathbf{S}_{0:k}^\top \mathbf{A}_k^{-1} \mathbf{S}_{0:k} [\mathbf{H}_{0:k}^{-1} + \mathbf{S}_{0:k}^\top \mathbf{A}_k^{-1} \mathbf{S}_{0:k}]^{-1} \mathbf{S}_{0:k}^\top \mathbf{A}_k^{-1} \\
&= \mathbf{H}_{0:k}^{-1/2} [\mathbf{H}_{0:k}^{-1} + \mathbf{S}_{0:k}^\top \mathbf{A}_k^{-1} \mathbf{S}_{0:k}]^{-1} \mathbf{S}_{0:k}^\top \mathbf{A}_k^{-1}
\end{aligned}
$$

by Woodbury's matrix indentity, it follows from the definition of $\mathsf{V}$ in Eq. (9) that

$$
\begin{aligned}
\mathsf{V}\mathsf{V}^\top &= \mathbf{H}_{0:k}^{1/2} \mathbf{S}_{0:k}^\top \mathbf{A}^{-1} \mathbf{S}_{k:\infty} \mathbf{H}_{k:\infty} \mathbf{S}_{k:\infty}^\top \mathbf{A}^{-1} \mathbf{S}_{0:k} \mathbf{H}_{0:k}^{1/2} \\
&= \mathbf{H}_{0:k}^{-1/2} [\mathbf{H}_{0:k}^{-1} + \mathbf{S}_{0:k}^\top \mathbf{A}_k^{-1} \mathbf{S}_{0:k}]^{-1} \mathbf{S}_{0:k}^\top \mathbf{A}_k^{-1} (\mathbf{S}_{k:\infty} \mathbf{H}_{k:\infty} \mathbf{S}_{k:\infty}^\top) \mathbf{A}_k^{-1} \mathbf{S}_{0:k} [\mathbf{H}_{0:k}^{-1} + \mathbf{S}_{0:k}^\top \mathbf{A}_k^{-1} \mathbf{S}_{0:k}]^{-1} \mathbf{H}_{0:k}^{-1/2} \\
&= \mathbf{H}_{0:k}^{-1/2} [\mathbf{H}_{0:k}^{-1} + \mathbf{S}_{0:k}^\top \mathbf{A}_k^{-1} \mathbf{S}_{0:k}]^{-1} \mathbf{S}_{0:k}^\top \mathbf{A}_k^{-1} \mathbf{S}_{0:k} [\mathbf{H}_{0:k}^{-1} + \mathbf{S}_{0:k}^\top \mathbf{A}_k^{-1} \mathbf{S}_{0:k}]^{-1} \mathbf{H}_{0:k}^{-1/2}. \tag{15}
\end{aligned}
$$

Combining Eq. (14) and (15) yields

$$
\mathsf{U}^2 + \mathsf{V}\mathsf{V}^\top = \mathbf{H}_{0:k}^{-1/2} [\mathbf{H}_{0:k}^{-1} + \mathbf{S}_{0:k}^\top \mathbf{A}_k^{-1} \mathbf{S}_{0:k}]^{-1} \mathbf{H}_{0:k}^{-1/2}, \tag{16}
$$

and therefore

$$
\left\langle \mathsf{U}^2 + \mathsf{V}\mathsf{V}^\top, \mathbf{H}_{0:k}^{1/2} \mathbf{w}_{*,0:k} \mathbf{w}_{*,0:k}^\top \mathbf{H}_{0:k}^{1/2} \right\rangle = \left\langle [\mathbf{H}_{0:k}^{-1} + \mathbf{S}_{0:k}^\top \mathbf{A}_k^{-1} \mathbf{S}_{0:k}]^{-1}, \mathbf{w}_{0:k}^* \mathbf{w}_{0:k}^{* \top} \right\rangle.
$$

$\square$

### C.2    A lower bound

For the approximation error Approx, we have the following result.

**Lemma C.2** (Lower bound on the approximation error). *When $r(\mathbf{H}) \geq M$, under Assumption 1C, the approximation error in (4) (and 7) satisfies*

$$
\mathbb{E}_{\mathbf{w}^*} \mathsf{Approx} \gtrsim \sum_{i=M}^{d} \lambda_i,
$$

*where $(\lambda_i)_{i=1}^d$ are eigenvalues of $\mathbf{H}$ in non-increasing order.*

*Proof of Lemma C.2.* For any $k \leq d$, following the proof of Lemma C.1, we have

$$
\mathsf{Approx} = \left\langle [\mathbf{H}^{1/2} \mathbf{S}^\top \mathbf{A}^{-1} \mathbf{S} \mathbf{H}^{1/2} - \mathbf{I}_d]^{\otimes 2}, \mathbf{H}^{1/2} \mathbf{w}^* (\mathbf{w}^*)^\top \mathbf{H}^{1/2} \right\rangle
$$

and

$$
\mathbf{H}^{1/2} \mathbf{S}^\top \mathbf{A}^{-1} \mathbf{S} \mathbf{H}^{1/2} - \mathbf{I}_d = \begin{pmatrix} \mathbf{H}_{0:k}^{1/2} \mathbf{S}_{0:k}^\top \mathbf{A}^{-1} \mathbf{S}_{0:k} \mathbf{H}_{0:k}^{1/2} - \mathbf{I}_k & \mathbf{H}_{0:k}^{1/2} \mathbf{S}_{0:k}^\top \mathbf{A}^{-1} \mathbf{S}_{k:\infty} \mathbf{H}_{k:\infty}^{1/2} \\ \mathbf{H}_{k:\infty}^{1/2} \mathbf{S}_{k:\infty}^\top \mathbf{A}^{-1} \mathbf{S}_{0:k} \mathbf{H}_{0:k}^{1/2} & \mathbf{H}_{k:\infty}^{1/2} \mathbf{S}_{k:\infty}^\top \mathbf{A}^{-1} \mathbf{S}_{k:\infty} \mathbf{H}_{k:\infty}^{1/2} - \mathbf{I}_{d-k} \end{pmatrix}
$$

$$=: \begin{pmatrix} \mathsf{U} & \mathsf{V} \\ \mathsf{V}^\top & \mathsf{W} \end{pmatrix}. \tag{17}$$

Therefore

$$[\mathbf{H}^{1/2}\mathbf{S}^\top \mathbf{A}^{-1}\mathbf{S}\mathbf{H}^{1/2} - \mathbf{I}_d]^{\otimes 2} = \begin{pmatrix} \mathsf{U}^2 + \mathsf{V}\mathsf{V}^\top & \mathsf{U}\mathsf{V} + \mathsf{V}\mathsf{W} \\ \mathsf{V}^\top \mathsf{U} + \mathsf{W}\mathsf{V}^\top & \mathsf{W}^2 + \mathsf{V}^\top \mathsf{V} \end{pmatrix}$$

and

$$\begin{aligned}
\mathbb{E}_{\mathbf{w}^*}\mathsf{Approx} &= \mathbb{E}_{\mathbf{w}^*}\big\langle \mathsf{U}^2 + \mathsf{V}\mathsf{V}^\top, \mathbf{H}_{0:k}^{1/2}\mathbf{w}_{0:k}^*\mathbf{w}_{0:k}^{*}\mathbf{H}_{0:k}^{1/2}\big\rangle + \mathbb{E}_{\mathbf{w}^*}\big\langle \mathsf{W}^2 + \mathsf{V}^\top \mathsf{V}, \mathbf{H}_{0:k}^{1/2}\mathbf{w}_{k:\infty}^*\mathbf{w}_{k:\infty}^{*}\mathbf{H}_{k:\infty}^{1/2}\big\rangle \\
&\quad + 2\mathbb{E}_{\mathbf{w}^*}\big\langle \mathsf{U}\mathsf{V} + \mathsf{V}\mathsf{W}, \mathbf{H}_{0:k}^{1/2}\mathbf{w}_{0:k}^*\mathbf{w}_{k:\infty}^{*}\mathbf{H}_{k:\infty}^{1/2}\big\rangle \\
&= \operatorname{tr}((\mathsf{U}^2 + \mathsf{V}\mathsf{V}^\top)\mathbf{H}_{0:k}) + \operatorname{tr}((\mathsf{W}^2 + \mathsf{V}^\top \mathsf{V})\mathbf{H}_{k:\infty}),
\end{aligned}$$

where the last line uses the fact that $\mathbb{E}_{\mathbf{w}^*}(\mathbf{w}^*)^{\otimes 2} = \mathbf{I}_d$. Using Eq. (13) and (16) in the proof of Lemma C.1, we further obtain

$$\begin{aligned}
\mathbb{E}_{\mathbf{w}^*}\mathsf{Approx} &= \operatorname{tr}(\mathbf{H}_{0:k}^{-1/2}[\mathbf{H}_{0:k}^{-1} + \mathbf{S}_{0:k}^\top \mathbf{A}_k^{-1}\mathbf{S}_{0:k}]^{-1}\mathbf{H}_{0:k}^{-1/2}\mathbf{H}_{0:k}) - \operatorname{tr}(\mathsf{W}\mathbf{H}_{k:\infty}) \\
&= \operatorname{tr}([\mathbf{H}_{0:k}^{-1} + \mathbf{S}_{0:k}^\top \mathbf{A}_k^{-1}\mathbf{S}_{0:k}]^{-1}) - \operatorname{tr}(\mathsf{W}\mathbf{H}_{k:\infty}) \\
&\geq -\operatorname{tr}(\mathsf{W}\mathbf{H}_{k:\infty}) =: T_3.
\end{aligned}$$

where $\mathbf{A}_k := \mathbf{S}_{k:\infty}\mathbf{H}_{k:\infty}\mathbf{S}_{k:\infty}^\top$. For $T_3$, we further have

$$\begin{aligned}
T_3 &= \operatorname{tr}(\mathbf{H}_{k:\infty}^{1/2}[\mathbf{I}_{d-k} - \mathbf{H}_{k:\infty}^{1/2}\mathbf{S}_{k:\infty}^\top \mathbf{A}^{-1}\mathbf{S}_{k:\infty}\mathbf{H}_{k:\infty}^{1/2}]\mathbf{H}_{k:\infty}^{1/2}) \\
&\geq \operatorname{tr}(\mathbf{H}_{k:\infty}^{1/2}[\mathbf{I}_{d-k} - \mathbf{H}_{k:\infty}^{1/2}\mathbf{S}_{k:\infty}^\top \mathbf{A}_k^{-1}\mathbf{S}_{k:\infty}\mathbf{H}_{k:\infty}^{1/2}]\mathbf{H}_{k:\infty}^{1/2}) \\
&\geq \sum_{i=1}^{d-k} \mu_i(\mathbf{I}_{d-k} - \mathbf{H}_{k:\infty}^{1/2}\mathbf{S}_{k:\infty}^\top \mathbf{A}_k^{-1}\mathbf{S}_{k:\infty}\mathbf{H}_{k:\infty}^{1/2}) \cdot \mu_{d+1-k-i}(\mathbf{H}_{k:\infty}),
\end{aligned}$$

where the second line is due to $\mathbf{A} \succeq \mathbf{A}_k$ (and hence $-\mathbf{A}^{-1} \succeq -\mathbf{A}_k^{-1}$), the third line follows from Von-Neuman's inequality. Since $\mathbf{M} := \mathbf{I}_{d-k} - \mathbf{H}_{k:\infty}^{1/2}\mathbf{S}_{k:\infty}^\top \mathbf{A}_k^{-1}\mathbf{S}_{k:\infty}\mathbf{H}_{k:\infty}^{1/2}$ is a projection matrix such that $\mathbf{M}^2 = \mathbf{M}$ and $\operatorname{tr}(\mathbf{I}_{d-k} - \mathbf{M}) = M$, it follows that $\mathbf{M}$ has $M$ eigenvalues 0 and $d - k - M$ eigenvalues 1. Therefore, we further have

$$T_3 \geq \sum_{i=1}^{d-k} \mu_i(\mathbf{M}) \cdot \mu_{d+1-k-i}(\mathbf{H}_{k:\infty}) \geq \sum_{i=k+M}^{d} \lambda_i$$

for any $k \leq d$. Letting $k = 0$ maximizes the lower bound and concludes the proof. $\qquad\square$

## C.3 A lower bound under Assumption 3

**Lemma C.3** (Lower bound on the approximation error under Assumption 3). *Under Assumption 3, the approximation error in (4) (and 7) satisfies*

$$\mathbb{E}_{\mathbf{w}^*}\mathsf{Approx} \gtrsim \sum_{i=M}^{d} \lambda_i i^{a-b},$$

*where $(\lambda_i)_{i=1}^{d}$ are eigenvalues of $\mathbf{H}$ in non-increasing order and the inequality hides some $(a, b)$-dependent constant.*

*Proof of Lemma C.3.* The proof is essentially the same as the proof of Lemma C.2 but we include it here for completeness. Let $\mathbf{H}^{\mathbf{w}} := \mathbb{E}[\mathbf{w}^*\mathbf{w}^{*\top}]$ be the covariance of the prior on $\mathbf{w}^*$. Following the proof of Lemma C.2, we have

$$\begin{aligned}
\mathbb{E}_{\mathbf{w}^*}\mathsf{Approx} &= \mathbb{E}_{\mathbf{w}^*}\big\langle \mathsf{U}^2 + \mathsf{V}\mathsf{V}^\top, \mathbf{H}_{0:k}^{1/2}\mathbf{w}_{0:k}^*\mathbf{w}_{0:k}^{*}\mathbf{H}_{0:k}^{1/2}\big\rangle + \mathbb{E}_{\mathbf{w}^*}\big\langle \mathsf{W}^2 + \mathsf{V}^\top \mathsf{V}, \mathbf{H}_{0:k}^{1/2}\mathbf{w}_{k:\infty}^*\mathbf{w}_{k:\infty}^{*}\mathbf{H}_{k:\infty}^{1/2}\big\rangle \\
&\quad + 2\mathbb{E}_{\mathbf{w}^*}\big\langle \mathsf{U}\mathsf{V} + \mathsf{V}\mathsf{W}, \mathbf{H}_{0:k}^{1/2}\mathbf{w}_{0:k}^*\mathbf{w}_{k:\infty}^{*}\mathbf{H}_{k:\infty}^{1/2}\big\rangle \\
&= \operatorname{tr}((\mathsf{U}^2 + \mathsf{V}\mathsf{V}^\top)\mathbf{H}_{0:k}\mathbf{H}_{0:k}^{\mathbf{w}}) + \operatorname{tr}((\mathsf{W}^2 + \mathsf{V}^\top \mathsf{V})\mathbf{H}_{k:\infty}\mathbf{H}_{k:\infty}^{\mathbf{w}}),
\end{aligned}$$

where the last line uses Assumption 3 and notice that $\mathbf{H}, \mathbf{H}^{\mathbf{w}}$ are both diagonal. Next, similar to the proof of Lemma C.2, using Eq. (13) and (16), we derive

$$\mathbb{E}_{\mathbf{w}^*}\mathsf{Approx} = \mathrm{tr}(\mathbf{H}_{0:k}^{-1/2}[\mathbf{H}_{0:k}^{-1} + \mathbf{S}_{0:k}^{\top}\mathbf{A}_k^{-1}\mathbf{S}_{0:k}]^{-1}\mathbf{H}_{0:k}^{-1/2}\mathbf{H}_{0:k}\mathbf{H}_{0:k}^{\mathbf{w}}) - \mathrm{tr}(\mathbf{W}\mathbf{H}_{k:\infty}\mathbf{H}_{k:\infty}^{\mathbf{w}})$$
$$= \mathrm{tr}([\mathbf{H}_{0:k}^{-1} + \mathbf{S}_{0:k}^{\top}\mathbf{A}_k^{-1}\mathbf{S}_{0:k}]^{-1}\mathbf{H}_{0:k}^{\mathbf{w}}) - \mathrm{tr}(\mathbf{W}\mathbf{H}_{k:\infty}\mathbf{H}_{k:\infty}^{\mathbf{w}})$$
$$\geq -\mathrm{tr}(\mathbf{W}\mathbf{H}_{k:\infty}\mathbf{H}_{k:\infty}^{\mathbf{w}}) =: \tilde{T}_3$$

where $\mathbf{A}_k := \mathbf{S}_{k:\infty}\mathbf{H}_{k:\infty}\mathbf{S}_{k:\infty}^{\top}$. For $\tilde{T}_3$, following the same argument for $T_3$ in the proof of Lemma C.2, we have

$$\tilde{T}_3 \geq \sum_{i=1}^{d-k} \mu_i(\mathbf{I}_{d-k} - \mathbf{H}_{k:\infty}^{1/2}\mathbf{S}_{k:\infty}^{\top}\mathbf{A}_k^{-1}\mathbf{S}_{k:\infty}\mathbf{H}_{k:\infty}^{1/2}) \cdot \mu_{d+1-k-i}(\mathbf{H}_{k:\infty}\mathbf{H}_{k:\infty}^{\mathbf{w}})$$

$$\geq \sum_{i=k+M}^{d} \mu_i(\mathbf{H}\mathbf{H}^{\mathbf{w}}) \gtrsim \sum_{i=k+M}^{d} i^{a-b}\lambda_i,$$

for any $k \leq d$ where the last inequality uses Assumption 3. Setting $k = 0$ maximizes the lower bound and concludes the proof. $\square$

## C.4 Examples on matching bounds for Approx

In this section, we derive matching upper and lower bounds for Approx (defined in Eq. 4 and 7) in three concrete examples: power-law spectrum (Lemma C.4), power-law spectrum with source condition (Lemma C.5) and logarithmic power-law spectrum (Lemma D.5).

**Lemma C.4** (Bounds on Approx under the power-law spectrum). *Suppose Assumption 1C and 2 hold. Then with probability at least $1 - e^{-\Omega(M)}$ over the randomness of $\mathbf{S}$*

$$M^{1-a} \lesssim \mathbb{E}_{\mathbf{w}^*}\mathsf{Approx} \lesssim M^{1-a}.$$

*Here, the hidden constants only depend on the power-law degree $a$.*

*Proof of Lemma C.4.* For the upper bound, by Lemma C.1 and noting $\mathbb{E}\mathbf{w}_i^{*2} = 1$ for all $i$, we have with probability at least $1 - e^{-\Omega(M)}$

$$\mathbb{E}_{\mathbf{w}^*}\mathsf{Approx} \lesssim \sum_{k>k_1} \lambda_i + \left(\frac{\sum_{i>k_1}\lambda_i}{M} + \lambda_{k_1+1} + \sqrt{\frac{\sum_{i>k_1}\lambda_i^2}{M}}\right) \cdot k_1$$

$$\lesssim k_1^{1-a} + \left(\frac{k_1^{1-a}}{M} + k_1^{-a} + \sqrt{\frac{k_1^{1-2a}}{M}}\right)k_1$$

$$\lesssim \left(\frac{k_1}{M} + 1\right)k_1^{1-a}$$

for any given $k_1 \leq M/2$. Here the hidden constants depend on $a$. Therefore, letting $k_1 = M/2$ in the upper bound yields

$$\mathbb{E}_{\mathbf{w}^*}\mathsf{Approx} \lesssim M^{1-a}$$

with probability at least $1 - e^{-\Omega(M)}$.

For the lower bound, we have from Lemma C.2 that

$$\mathbb{E}_{\mathbf{w}^*}\mathsf{Approx} \gtrsim \sum_{i=M}^{\infty} i^{-a} \gtrsim M^{1-a}.$$

This completes the proof. $\square$

**Lemma C.5** (Bounds on Approx under the source condition). *Suppose Assumption 3 hold. Then with probability at least $1 - e^{-\Omega(M)}$ over the randomness of $\mathbf{S}$*

$$M^{1-b} \lesssim \mathbb{E}_{\mathbf{w}^*}\mathsf{Approx} \lesssim M^{1-b}.$$

*Here, the hidden constants only depend on the power-law degrees $a, b$.*

*Proof of Lemma C.5.* For the upper bound, by Lemma C.1 and noting $\mathbb{E}\mathbf{w}_i^{*2} \asymp i^{a-b}$ for all $i$, we have with probability at least $1 - e^{-\Omega(M)}$

$$\mathsf{Approx} \lesssim \sum_{k>k_1} \lambda_i i^{a-b} + \left( \frac{\sum_{i>k_1} \lambda_i}{M} + \lambda_{k_1+1} + \sqrt{\frac{\sum_{i>k_1} \lambda_i^2}{M}} \right) \cdot k_1^{1+a-b}$$

$$\lesssim k_1^{1-b} + \left( \frac{k_1^{1-a}}{M} + k_1^{-a} + \sqrt{\frac{k_1^{1-2a}}{M}} \right) k_1^{1+a-b}$$

$$\lesssim \left( \frac{k_1}{M} + 1 \right) k_1^{1-b}$$

for any given $k_1 \leq M/2$. Here the hidden constants depend on $a, b$. Moreover, choosing $k_1 = M/2$ in the upper bound gives

$$\mathbb{E}_{\mathbf{w}^*} \mathsf{Approx} \lesssim M^{1-b}$$

with probability at least $1 - e^{-\Omega(M)}$.

For the lower bound, we have from Lemma C.3 that

$$\mathbb{E}_{\mathbf{w}^*} \mathsf{Approx} \gtrsim \sum_{i=M}^{\infty} i^{-a} \cdot i^{a-b} \gtrsim M^{1-b}.$$

This completes the proof. $\qquad \square$

**Lemma C.6** (Bounds on Approx under the logarithmic power-law spectrum). *Suppose Assumption 4 hold. Then with probability at least $1 - e^{-\Omega(M)}$ over the randomness of* $\mathbf{S}$

$$\log^{1-a} M \lesssim \mathbb{E}_{\mathbf{w}^*} \mathsf{Approx} \lesssim \log^{1-a} M.$$

*Here, the hidden constants only depend on the power-law degree $a$.*

*Proof of Lemma C.6.* For the upper bound, by Lemma C.1 and noting $\mathbb{E}\mathbf{w}_i^{*2} = 1$ for all $i$, we have with probability at least $1 - e^{-\Omega(M)}$

$$\mathsf{Approx} \lesssim \sum_{k>k_1} \lambda_i + \left( \frac{\sum_{i>k_1} \lambda_i}{M} + \lambda_{k_1+1} + \sqrt{\frac{\sum_{i>k_1} \lambda_i^2}{M}} \right) k_1$$

$$\lesssim \log^{1-a} k_1 + \left( \frac{\log^{1-a} k_1}{M} + k_1^{-1} \log^{-a} k_1 + \sqrt{\frac{k_1^{1-2a}}{M}} \right) k_1$$

$$\lesssim \left( 1 + \frac{k_1}{M} + \frac{1}{\log k_1} + \frac{1}{\log k_1} \sqrt{\frac{k_1}{M}} \right) \log^{1-a} k_1$$

$$\lesssim \log^{1-a} k_1$$

for any given $k_1 \leq M/2$, where the third line uses $\sum_{i>k_1} \lambda_i^2 \lesssim 1/(k_1 \log^{2a} k_1)$. Choosing $k_1 = M/2$, we obtain

$$\mathbb{E}_{\mathbf{w}^*} \mathsf{Approx} \lesssim \log^{1-a} M$$

with probability at least $1 - e^{-\Omega(M)}$. Here the hidden constants depend on $a, b$.

For the lower bound, we have from Lemma C.2 that

$$\mathbb{E}_{\mathbf{w}^*} \mathsf{Approx} \gtrsim \sum_{i=M}^{\infty} \lambda_i \gtrsim \sum_{i=M}^{\infty} i^{-1} \log^{-a} i \gtrsim \log^{1-a} M.$$

Therefore, we have established matching upper and lower bounds for Approx. $\qquad \square$

# D Bias error

In this section, we derive upper and lower bounds for $\mathsf{Bias}(\mathbf{w}^*)$ defined in Eq. (5). Moreover, we show that the upper and lower bounds match up to constant factors in concrete examples.

## D.1 An upper bound

**Lemma D.1** (Upper bound on the bias term). *Suppose the initial stepsize $\gamma \leq \frac{1}{c\,\mathrm{tr}(\mathbf{SHS}^\top)}$ for some constant $c > 1$. Then for any $\mathbf{w}^* \in \mathbb{H}$ and $k \in [d]$ such that $r(\mathbf{H}) \geq k + M$, the bias term in (5) satisfies*

$$\mathsf{Bias}(\mathbf{w}^*) \lesssim \frac{1}{N_{\mathtt{eff}}\gamma}\|\mathbf{v}^*\|_2^2.$$

*Moreover, for any $k \leq M/3$ such that $r(\mathbf{H}) \geq k + M$, the bias term satisfies*

$$\mathsf{Bias}(\mathbf{w}^*) \lesssim \frac{\|\mathbf{w}_{0:k}^*\|_2^2}{N_{\mathtt{eff}}\gamma} \cdot \left[\frac{\mu_{M/2}(\mathbf{A}_k)}{\mu_M(\mathbf{A}_k)}\right]^2 + \|\mathbf{w}_{k:\infty}^*\|_{\mathbf{H}_{k:\infty}}^2$$

*with probability $1 - e^{-\Omega(M)}$, where $\mathbf{A}_k := \mathbf{S}_{k:\infty}\mathbf{H}_{k:\infty}\mathbf{S}_{k:\infty}^\top$, $\{\mu_i(\mathbf{A}_k)\}_{i=1}^M$ denote the eigenvalues of $\mathbf{A}_k$ in non-increasing order for some constant $c > 1$.*

*Proof of Lemma D.1.* Similar to the proof of Lemma C.1, we can without loss of generality assume the covariance matrix $\mathbf{H} = \mathrm{diag}\{\lambda_1, \lambda_2, \ldots, \lambda_d\}$ where $\lambda_i \geq \lambda_j$ for any $i \geq j$. Let $\mathbf{SH}^{1/2} = \tilde{\mathbf{U}}\left(\tilde{\boldsymbol{\Lambda}}^{1/2} \quad \mathbf{0}\right)\tilde{\mathbf{V}}^\top$ be the singular value decomposition of $\mathbf{SHS}^\top$, where $\tilde{\boldsymbol{\Lambda}} := \mathrm{diag}\{\tilde{\lambda}_1, \tilde{\lambda}_2, \ldots, \tilde{\lambda}_d\}$ is a diagonal matrix diagonal entries in non-increasing order. Define $\mathbf{A}_k := \mathbf{S}_{k:\infty}\mathbf{H}_{k:\infty}\mathbf{S}_{k:\infty}^\top$. Then it follows from similar arguments as in Lemma C.1 that $\mathbf{A}_k$ is invertible.

Since

$$\|\gamma_t\mathbf{SHS}^\top\|_2 = \gamma_t\tilde{\lambda}_1 \leq \gamma\tilde{\lambda}_1 \leq \frac{\tilde{\lambda}_1}{c\sum_{i=1}^M \tilde{\lambda}_i} \leq 1$$

for some constant $c > 1$ by the stepsize assumption, it follows that $\mathbf{I}_M - \gamma_t\mathbf{SHS}^\top \succ \mathbf{0}_M$ for all $t \in [N]$. Therefore, it can be verified that

$$\prod_{t=1}^N (\mathbf{I}_M - \gamma_t\mathbf{SHS}^\top)\mathbf{SHS}^\top \prod_{t=1}^N (\mathbf{I}_M - \gamma_t\mathbf{SHS}^\top) \preceq (\mathbf{I}_M - \gamma\mathbf{SHS}^\top)^{N_{\mathtt{eff}}}\mathbf{SHS}^\top(\mathbf{I}_M - \gamma\mathbf{SHS}^\top)^{N_{\mathtt{eff}}} =: \mathsf{M},$$

and by definition of $\mathsf{Bias}(\mathbf{w}^*)$ in Eq. (5), we have

$$\mathsf{Bias}(\mathbf{w}^*) \asymp \left\|\prod_{t=1}^N \left(\mathbf{I} - \gamma_t\mathbf{SHS}^\top\right)\mathbf{v}^*\right\|_{\mathbf{SHS}^\top}^2 \leq \left\|\left(\mathbf{I} - \gamma\mathbf{SHS}^\top\right)^{N_{\mathtt{eff}}}\mathbf{v}^*\right\|_{\mathbf{SHS}^\top}^2$$

$$= \langle \mathsf{M}, \mathbf{v}^{*\otimes 2}\rangle. \tag{18}$$

Note that the eigenvalues of $\mathsf{M}$ are $\{\tilde{\lambda}_i(1 - \gamma\tilde{\lambda}_i)^{2N_{\mathtt{eff}}}\}_{i=1}^M$. Since the function $f(x) = x(1 - \gamma x)^{2N_{\mathtt{eff}}}$ is maximized at $x_0 = 1/[(2N_{\mathtt{eff}} + 1)\gamma]$ for $x \in [0, 1/\gamma]$ with $f(x_0) \lesssim 1/(N_{\mathtt{eff}}\gamma)$, it follows that

$$\|\mathsf{M}\|_2 \leq c/(N_{\mathtt{eff}}\gamma) \tag{19}$$

for some constant $c > 0$. The first part of Lemma D.1 follows immediately.

Now we prove the second part of Lemma D.1. Recall that $\mathbf{v}^* = \left(\mathbf{SHS}^\top\right)^{-1}\mathbf{SHw}^*$. Substituting $\mathbf{SH} = (\mathbf{S}_{0:k}\mathbf{H}_{0:k} \quad \mathbf{S}_{k:\infty}\mathbf{H}_{k:\infty})$ into $\mathbf{v}^*$, we obtain

$$\langle \mathsf{M}, \mathbf{v}^{*\otimes 2}\rangle = \langle \mathsf{M}, ((\mathbf{SHS}^\top)^{-1}\mathbf{SHw}^*)^{\otimes 2}\rangle$$

$$= \mathbf{w}^{*\top}\mathbf{HS}^\top(\mathbf{SHS}^\top)^{-1}\mathsf{M}(\mathbf{SHS}^\top)^{-1}\mathbf{SHw}^*$$

$$\leq 2T_1 + 2T_2,$$

where

$$T_1 := (\mathbf{w}_{0:k}^*)^\top \mathbf{H}_{0:k} \mathbf{S}_{0:k}^\top (\mathbf{SHS}^\top)^{-1} \mathsf{M} (\mathbf{SHS}^\top)^{-1} \mathbf{S}_{0:k} \mathbf{H}_{0:k} \mathbf{w}_{0:k}^*, \tag{20}$$

$$T_2 := (\mathbf{w}_{k:\infty}^*)^\top \mathbf{H}_{k:\infty} \mathbf{S}_{k:\infty}^\top (\mathbf{SHS}^\top)^{-1} \mathsf{M} (\mathbf{SHS}^\top)^{-1} \mathbf{S}_{k:\infty} \mathbf{H}_{k:\infty} \mathbf{w}_{k:\infty}^*. \tag{21}$$

We claim the following results which we prove later. With probability $1 - e^{-\Omega(M)}$

$$T_1 \le \frac{c\|\mathbf{w}_{0:k}^*\|_2^2}{N_{\texttt{eff}}\gamma} \cdot \left[\frac{\mu_{M/2}(\mathbf{A}_k)}{\mu_M(\mathbf{A}_k)}\right]^2 \tag{22a}$$

for some constant $c > 0$.

$$T_2 \le \|\mathbf{w}_{k:\infty}^*\|_{\mathbf{H}_{k:\infty}}^2. \tag{22b}$$

Combining Eq. (22a), (22b) gives the second part of Lemma D.1.

**Proof of claim** (22a)    By definition of $T_1$, we have

$$T_1 \le \|\mathbf{H}_{0:k} \mathbf{S}_{0:k}^\top (\mathbf{SHS}^\top)^{-1} \mathsf{M} (\mathbf{SHS}^\top)^{-1} \mathbf{S}_{0:k} \mathbf{H}_{0:k}\|_2 \cdot \|\mathbf{w}_{0:k}^*\|_2^2.$$

Moreover,

$$\|\mathbf{H}_{0:k} \mathbf{S}_{0:k}^\top (\mathbf{SHS}^\top)^{-1} \mathsf{M} (\mathbf{SHS}^\top)^{-1} \mathbf{S}_{0:k} \mathbf{H}_{0:k}\|_2$$
$$\le \|\mathsf{M}\|_2 \cdot \|(\mathbf{SHS}^\top)^{-1} \mathbf{S}_{0:k} \mathbf{H}_{0:k}\|_2^2$$
$$\le \frac{c}{N_{\texttt{eff}}\gamma} \|(\mathbf{SHS}^\top)^{-1} \mathbf{S}_{0:k} \mathbf{H}_{0:k}\|_2^2$$

for some constant $c > 0$, where the last line uses Eq. (19).

It remains to show

$$\|(\mathbf{SHS}^\top)^{-1} \mathbf{S}_{0:k} \mathbf{H}_{0:k}\|_2 \le c \cdot \frac{\mu_{M/2}(\mathbf{A}_k)}{\mu_M(\mathbf{A}_k)} \tag{23}$$

for some constant $c > 0$ with probability $1 - e^{-\Omega(M)}$. Since $\mathbf{SHS}^\top = \mathbf{S}_{0:k} \mathbf{H}_{0:k} \mathbf{S}_{0:k}^\top + \mathbf{A}_k$, we have

$$(\mathbf{SHS}^\top)^{-1} \mathbf{S}_{0:k} \mathbf{H}_{0:k} = (\mathbf{A}_k^{-1} - \mathbf{A}_k^{-1} \mathbf{S}_{0:k} [\mathbf{H}_{0:k}^{-1} + \mathbf{S}_{0:k}^\top \mathbf{A}_k^{-1} \mathbf{S}_{0:k}]^{-1} \mathbf{S}_{0:k}^\top \mathbf{A}_k^{-1}) \mathbf{S}_{0:k} \mathbf{H}_{0:k}$$
$$= \mathbf{A}_k^{-1} \mathbf{S}_{0:k} \mathbf{H}_{0:k} - \mathbf{A}_k^{-1} \mathbf{S}_{0:k} [\mathbf{H}_{0:k}^{-1} + \mathbf{S}_{0:k}^\top \mathbf{A}_k^{-1} \mathbf{S}_{0:k}]^{-1} \mathbf{S}_{0:k}^\top \mathbf{A}_k^{-1} \mathbf{S}_{0:k} \mathbf{H}_{0:k}$$
$$= \mathbf{A}_k^{-1} \mathbf{S}_{0:k} [\mathbf{H}_{0:k}^{-1} + \mathbf{S}_{0:k}^\top \mathbf{A}_k^{-1} \mathbf{S}_{0:k}]^{-1} \mathbf{H}_{0:k}^{-1} \mathbf{H}_{0:k}$$
$$= \mathbf{A}_k^{-1} \mathbf{S}_{0:k} [\mathbf{H}_{0:k}^{-1} + \mathbf{S}_{0:k}^\top \mathbf{A}_k^{-1} \mathbf{S}_{0:k}]^{-1}, \tag{24}$$

where the second line uses Woodbury's identity. Since

$$\mathbf{H}_{0:k}^{-1} + \mathbf{S}_{0:k}^\top \mathbf{A}_k^{-1} \mathbf{S}_{0:k} \succeq \mathbf{S}_{0:k}^\top \mathbf{A}_k^{-1} \mathbf{S}_{0:k},$$

it follows that

$$\|[\mathbf{H}_{0:k}^{-1} + \mathbf{S}_{0:k}^\top \mathbf{A}_k^{-1} \mathbf{S}_{0:k}]^{-1}\|_2 \le \|[\mathbf{S}_{0:k}^\top \mathbf{A}_k^{-1} \mathbf{S}_{0:k}]^{-1}\|_2.$$

Therefore, with probability at least $1 - e^{-\Omega(M)}$

$$\|\mathbf{A}_k^{-1} \mathbf{S}_{0:k} [\mathbf{H}_{0:k}^{-1} + \mathbf{S}_{0:k}^\top \mathbf{A}_k^{-1} \mathbf{S}_{0:k}]^{-1}\|_2 \le \|\mathbf{A}_k^{-1}\|_2 \cdot \|\mathbf{S}_{0:k}\|_2 \cdot \|[\mathbf{H}_{0:k}^{-1} + \mathbf{S}_{0:k}^\top \mathbf{A}_k^{-1} \mathbf{S}_{0:k}]^{-1}\|_2$$
$$\le \|\mathbf{A}_k^{-1}\|_2 \cdot \|\mathbf{S}_{0:k}\|_2 \cdot \|[\mathbf{S}_{0:k}^\top \mathbf{A}_k^{-1} \mathbf{S}_{0:k}]^{-1}\|_2$$
$$\le \frac{\|\mathbf{A}_k^{-1}\|_2 \cdot \|\mathbf{S}_{0:k}\|_2}{\mu_{\min}(\mathbf{S}_{0:k}^\top \mathbf{A}_k^{-1} \mathbf{S}_{0:k})} \lesssim \frac{\|\mathbf{A}_k^{-1}\|_2}{\mu_{\min}(\mathbf{S}_{0:k}^\top \mathbf{A}_k^{-1} \mathbf{S}_{0:k})}$$

where the last inequality follows from the fact that $\|\mathbf{S}_{0:k}\|_2 = \sqrt{\|\mathbf{S}_{0:k}^\top \mathbf{S}_{0:k}\|_2} \le c$ for some constant $c > 0$ when $k \le M/2$ with probability at least $1 - e^{-\Omega(M)}$. Since $\mathbf{S}_{0:k}$ is independent of $\mathbf{A}_k$ and the distribution of $\mathbf{S}_{0:k}$ is rotationally invariant, we may write $\mathbf{S}_{0:k}^\top \mathbf{A}_k^{-1} \mathbf{S}_{0:k} = \sum_{i=1}^M \frac{1}{\tilde{\lambda}_{M-i}} \tilde{\mathbf{s}}_i \tilde{\mathbf{s}}_i^\top,$

where $\tilde{\mathbf{s}}_i \overset{\text{iid}}{\sim} \mathcal{N}(0, \mathbf{I}_k/M)$ and $(\hat{\lambda}_i)_{i=1}^M$ are eigenvalues of $\mathbf{A}_k$ in non-increasing order. Therefore, for $k \leq M/3$

$$\mathbf{S}_{0:k}^\top \mathbf{A}_k^{-1} \mathbf{S}_{0:k} = \sum_{i=1}^M \frac{1}{\hat{\lambda}_{M-i}} \tilde{\mathbf{s}}_i \tilde{\mathbf{s}}_i^\top \succeq \sum_{i=1}^{M/2} \frac{1}{\hat{\lambda}_{M-i}} \tilde{\mathbf{s}}_i \tilde{\mathbf{s}}_i^\top \succeq \frac{1}{\hat{\lambda}_{M/2}} \sum_{i=1}^{M/2} \tilde{\mathbf{s}}_i \tilde{\mathbf{s}}_i^\top \succeq \frac{c\mathbf{I}_k}{\hat{\lambda}_{M/2}} \qquad (25)$$

for some constant $c > 0$ with probability at least $1 - e^{-\Omega(M)}$, where in the last line we again use the concentration properties of gaussian covariance matrices (see e.g., Theorem 6.1 in [41]). As a direct consequence, we have

$$\|\mathbf{A}_k^{-1} \mathbf{S}_{0:k} [\mathbf{H}_{0:k}^{-1} + \mathbf{S}_{0:k}^\top \mathbf{A}_k^{-1} \mathbf{S}_{0:k}]^{-1}\|_2 \leq c \cdot \frac{\mu_{M/2}(\mathbf{A}_k)}{\mu_M(\mathbf{A}_k)}$$

with probability at least $1 - e^{-\Omega(M)}$ for some constant $c > 0$. This concludes the proof.

**Proof of claim** (22b)    By definition of $T_2$ in Eq. (21), we have

$$\begin{aligned}
T_2 &= \mathbf{w}_{k:\infty}^{*\top} \mathbf{H}_{k:\infty} \mathbf{S}_{k:\infty}^\top (\mathbf{SHS}^\top)^{-1/2} (\mathbf{I}_M - \gamma \mathbf{SHS}^\top)^{2N_{\text{eff}}} (\mathbf{SHS}^\top)^{-1/2} \mathbf{S}_{k:\infty} \mathbf{H}_{k:\infty} \mathbf{w}_{k:\infty}^* \\
&\leq \mathbf{w}_{k:\infty}^{*\top} \mathbf{H}_{k:\infty} \mathbf{S}_{k:\infty}^\top (\mathbf{SHS}^\top)^{-1} \mathbf{S}_{k:\infty} \mathbf{H}_{k:\infty} \mathbf{w}_{k:\infty}^* \\
&\leq \|\mathbf{H}_{k:\infty}^{1/2} \mathbf{S}_{k:\infty}^\top (\mathbf{SHS}^\top)^{-1} \mathbf{S}_{k:\infty} \mathbf{H}_{k:\infty}^{1/2}\| \cdot \|\mathbf{w}_{k:\infty}^*\|_{\mathbf{H}_{k:\infty}}^2 \\
&\leq \|\mathbf{w}_{k:\infty}^*\|_{\mathbf{H}_{k:\infty}}^2,
\end{aligned}$$

where the last line follows from

$$\begin{aligned}
\|\mathbf{H}_{k:\infty}^{1/2} \mathbf{S}_{k:\infty}^\top (\mathbf{SHS}^\top)^{-1} \mathbf{S}_{k:\infty} \mathbf{H}_{k:\infty}^{1/2}\|_2 &= \|\mathbf{H}_{k:\infty}^{1/2} \mathbf{S}_{k:\infty}^\top (\mathbf{S}_{0:k} \mathbf{H}_{0:k} \mathbf{S}_{0:k}^\top + \mathbf{S}_{k:\infty} \mathbf{H}_{k:\infty} \mathbf{S}_{k:\infty}^\top)^{-1} \mathbf{S}_{k:\infty} \mathbf{H}_{k:\infty}^{1/2}\|_2 \\
&\leq \|\mathbf{H}_{k:\infty}^{1/2} \mathbf{S}_{k:\infty}^\top \mathbf{A}_k^{-1} \mathbf{S}_{k:\infty} \mathbf{H}_{k:\infty}^{1/2}\|_2 \leq 1.
\end{aligned}$$

$\square$

### D.2   A lower bound

**Lemma D.2** (Lower bound on the bias term). *Suppose $\mathbf{w}^*$ follows some prior distribution and the initial stepsize $\gamma \leq \frac{1}{c\,\mathrm{tr}(\mathbf{SHS}^\top)}$ for some constant $c > 2$. Let $\mathbf{H}^{\mathbf{w}} := \mathbb{E}\mathbf{w}^*\mathbf{w}^{*\top}$. Then the bias term in Eq. (5) satisfies*

$$\mathbb{E}_{\mathbf{w}^*} \mathsf{Bias}(\mathbf{w}^*) \gtrsim \sum_{i:\tilde{\lambda}_i < 1/(\gamma N_{\text{eff}})} \frac{\mu_i(\mathbf{SHH}^{\mathbf{w}}\mathbf{HS}^\top)}{\mu_i(\mathbf{SHS}^\top)}$$

*almost surely, where $\mathsf{M}_{\mathsf{N}} := \mathbf{SHS}^\top (\mathbf{I} - 2\gamma \mathbf{SHS}^\top)^{2N_{\text{eff}}}$.*

*Proof of Lemma D.2.* Adopt the notations in the proof of Lemma D.1. By definition of the bias term, we have

$$\begin{aligned}
\mathsf{Bias}(\mathbf{w}^*) &\eqsim \left\| \prod_{t=1}^N (\mathbf{I} - \gamma_t \mathbf{SHS}^\top) \mathbf{v}^* \right\|_{\mathbf{SHS}^\top}^2 \\
&= \left\langle \mathbf{SHS}^\top \prod_{t=1}^N (\mathbf{I} - \gamma_t \mathbf{SHS}^\top)^{2N_{\text{eff}}}, \mathbf{v}^{*\otimes 2} \right\rangle \\
&\geq \left\langle \mathbf{SHS}^\top (\mathbf{I} - \sum_{t=1}^N \gamma_t \mathbf{SHS}^\top)^{2N_{\text{eff}}}, \mathbf{v}^{*\otimes 2} \right\rangle \\
&\geq \left\langle \mathbf{SHS}^\top (\mathbf{I} - 2\gamma \mathbf{SHS}^\top)^{2N_{\text{eff}}}, \mathbf{v}^{*\otimes 2} \right\rangle =: \langle \mathsf{M}_{\mathsf{N}}, \mathbf{v}^{*\otimes 2} \rangle, \qquad (26)
\end{aligned}$$

where the third line uses $\mathbf{I}_M - 2\gamma_t \mathbf{SHS}^\top \succ \mathbf{0}_M$ for all $t \in [N]$ established in the proof of Lemma D.1, $\sum_{i=1}^N \gamma_i \leq 2\gamma N_{\text{eff}}$, and the fact that $(1-w)(1-v) \geq 1 - w - v$ for $w, v > 0$. Substituting the definition of $\mathbf{v}^*$ in Eq. (5) into the expression, we obtain

$$\mathbb{E}_{\mathbf{w}^*} \mathsf{Bias}(\mathbf{w}^*) \gtrsim \mathbb{E}_{\mathbf{w}^*} \langle \mathsf{M}_{\mathsf{N}}, \mathbf{v}^{*\otimes 2} \rangle = \mathbb{E}_{\mathbf{w}^*} \langle \mathsf{M}_{\mathsf{N}}, ((\mathbf{SHS}^\top)^{-1} \mathbf{SHw}^*)^{\otimes 2} \rangle$$

$$= \text{tr}(\mathbf{H}\mathbf{S}^\top(\mathbf{S}\mathbf{H}\mathbf{S}^\top)^{-1}\mathsf{M}_\mathsf{N}(\mathbf{S}\mathbf{H}\mathbf{S}^\top)^{-1}\mathbf{S}\mathbf{H}\mathbf{H}^\mathbf{w})$$

$$= \text{tr}((\mathbf{S}\mathbf{H}\mathbf{S}^\top)^{-1}\mathsf{M}_\mathsf{N}(\mathbf{S}\mathbf{H}\mathbf{S}^\top)^{-1}\mathbf{S}\mathbf{H}\mathbf{H}^\mathbf{w}\mathbf{H}\mathbf{S}^\top)$$

$$\geq \sum_{i=1}^{M} \mu_{M-i+1}((\mathbf{S}\mathbf{H}\mathbf{S}^\top)^{-1}\mathsf{M}_\mathsf{N}(\mathbf{S}\mathbf{H}\mathbf{S}^\top)^{-1}) \cdot \mu_i(\mathbf{S}\mathbf{H}\mathbf{H}^\mathbf{w}\mathbf{H}\mathbf{S}^\top),$$

where the last line uses Von Neumann's trace inequality. Continuing the calculation, we have

$$\mathbb{E}_{\mathbf{w}^*}\text{Bias}(\mathbf{w}^*) \gtrsim \sum_{i=1}^{M} \frac{\mu_i(\mathbf{S}\mathbf{H}\mathbf{H}^\mathbf{w}\mathbf{H}\mathbf{S}^\top)}{\mu_i((\mathbf{S}\mathbf{H}\mathbf{S}^\top)^2\mathsf{M}_\mathsf{N}^{-1})}$$

$$= \sum_{i=1}^{M} \frac{\mu_i(\mathbf{S}\mathbf{H}\mathbf{H}^\mathbf{w}\mathbf{H}\mathbf{S}^\top)}{\mu_i\left((\mathbf{S}\mathbf{H}\mathbf{S}^\top)(\mathbf{I} - 2\gamma\mathbf{S}\mathbf{H}\mathbf{S}^\top)^{-2N_{\text{eff}}}\right)}$$

$$\gtrsim \sum_{i:\tilde{\lambda}_i < 1/(\gamma N_{\text{eff}})} \frac{\mu_i(\mathbf{S}\mathbf{H}\mathbf{H}^\mathbf{w}\mathbf{H}\mathbf{S}^\top)}{\mu_i(\mathbf{S}\mathbf{H}\mathbf{S}^\top)},$$

where the first inequality uses $\mu_{M+i-1}(A) = \mu_i^{-1}(A^{-1})$ for any positive definite matrix $A \in \mathbb{R}^{M \times M}$, and the second line follows from the definition of $\mathsf{M}_\mathsf{N}$ and the fact that $(1 - \lambda\gamma N_{\text{eff}})^{-2N_{\text{eff}}} \lesssim 1$ when $\lambda < 1/(\gamma N_{\text{eff}})$. $\qquad\square$

### D.3  Examples on matching bounds for $\text{Bias}(\mathbf{w}^*)$

In this section, we derive matching upper and lower bounds for $\text{Bias}(\mathbf{w}^*)$ in (5) in three scenarios: power-law spectrum (Lemma D.3), power-law spectrum with source condition (Lemma D.4) and logarithmic power-law spectrum (Lemma D.5). Recall that we define $\text{Bias} := \mathbb{E}_{\mathbf{w}^*}\text{Bias}(\mathbf{w}^*)$.

**Lemma D.3** (Bounds on Bias under the power-law spectrum). *Suppose Assumption 1C and 2 hold and the initial stepsize $\gamma \leq \frac{1}{c\,\text{tr}(\mathbf{S}\mathbf{H}\mathbf{S}^\top)}$ for some constant $c > 2$. Then with probability at least $1 - e^{-\Omega(M)}$ over the randomness of $\mathbf{S}$*

$$\mathbb{E}_{\mathbf{w}^*}\text{Bias}(\mathbf{w}^*) \lesssim \max\left\{(N_{\text{eff}}\gamma)^{1/a-1}, \ M^{1-a}\right\},$$

*and*

$$\mathbb{E}_{\mathbf{w}^*}\text{Bias}(\mathbf{w}^*) \gtrsim (N_{\text{eff}}\gamma)^{1/a-1}$$

*when $(N_{\text{eff}}\gamma)^{1/a} \leq M/c$ for some constant $c > 0$. Here, all the (hidden) constants depend only on the power-law degree $a$.*

*Proof of Lemma D.3.* For the upper bound, using Lemma G.5, D.1 and the assumption that $\mathbb{E}\mathbf{w}_i^{*2} = 1$ for all $i > 0$, with probability at least $1 - e^{-\Omega(M)}$, we have

$$\mathbb{E}_{\mathbf{w}^*}\text{Bias}(\mathbf{w}^*) \lesssim \mathbb{E}_{\mathbf{w}^*}\left[\frac{\|\mathbf{w}_{0:k_2}^*\|_2^2}{N_{\text{eff}}\gamma} + \|\mathbf{w}_{k_2:\infty}^*\|_{\mathbf{H}_{k_2:\infty}}^2\right]$$

$$\lesssim \frac{k_2}{N_{\text{eff}}\gamma} + \sum_{k>k_2} \lambda_i$$

$$\approx \frac{k_2}{N_{\text{eff}}\gamma} + k_2^{1-a}$$

$$\lesssim \max\left\{(N_{\text{eff}}\gamma)^{1/a-1}, \ M^{1-a}\right\},$$

where in the last inequality, we choose $k_2 = \lceil M/3 \rceil \wedge (N_{\text{eff}}\gamma)^{1/a}$ to minimize the upper bound.

When $(N_{\text{eff}}\gamma)^{1/a} \leq M/3$, combining Lemma D.2 and G.4 gives the lower bound

$$\mathbb{E}_{\mathbf{w}^*}\text{Bias}(\mathbf{w}^*) \gtrsim \sum_{i:\tilde{\lambda}_i < 1/(N_{\text{eff}}\gamma)} \frac{\mu_i(\mathbf{S}\mathbf{H}\mathbf{H}^\mathbf{w}\mathbf{H}\mathbf{S}^\top)}{\mu_i(\mathbf{S}\mathbf{H}\mathbf{S}^\top)} = \sum_{i:\tilde{\lambda}_i < 1/(N_{\text{eff}}\gamma)} \frac{\mu_i(\mathbf{S}\mathbf{H}^2\mathbf{S}^\top)}{\mu_i(\mathbf{S}\mathbf{H}\mathbf{S}^\top)},$$

$$\gtrsim \sum_{\tilde{\lambda}_i < 1/(N_{\text{eff}}\gamma), i \le M} \frac{i^{-2a}}{i^{-a}} = \sum_{\lambda_i < 1/(N_{\text{eff}}\gamma), i \le M} i^{-a} \gtrsim (N_{\text{eff}}\gamma)^{1/a-1}$$

with probability at least $1 - e^{-\Omega(M)}$. Here, the hidden constants depend only on $a$. $\qquad\square$

**Lemma D.4** (Bounds on Bias under the source condition)**.** *Suppose Assumption 3 hold and the initial stepsize* $\gamma \le \frac{1}{c\,\text{tr}(\mathbf{SHS}^\top)}$ *for some constant* $c > 2$*. Then with probability at least* $1 - e^{-\Omega(M)}$ *over the randomness of* $\mathbf{S}$

$$\mathbb{E}_{\mathbf{w}^*}\text{Bias}(\mathbf{w}^*) \lesssim \max\left\{(N_{\text{eff}}\gamma)^{(1-b)/a}, \, M^{1-b}\right\},$$

*and*

$$\mathbb{E}_{\mathbf{w}^*}\text{Bias}(\mathbf{w}^*) \gtrsim (N_{\text{eff}}\gamma)^{(1-b)/a}$$

*when* $(N_{\text{eff}}\gamma)^{1/a} \le M/c$ *for some constant* $c > 0$*. In both results, the hidden constants depend only on* $a, b$*.*

*Proof of Lemma D.4.* For the upper bound, using Lemma G.5, D.1 and the assumption that (w.l.o.g.) $\mathbb{E}\mathbf{w}_i^{*2} \approx i^{a-b}$ for all $i > 0$, with probability at least $1 - e^{-\Omega(M)}$, we have

$$\mathbb{E}_{\mathbf{w}^*}\text{Bias}(\mathbf{w}^*) \lesssim \mathbb{E}_{\mathbf{w}^*}\left[\frac{\|\mathbf{w}_{0:k_2}^*\|_2^2}{N_{\text{eff}}\gamma} + \|\mathbf{w}_{k_2:\infty}^*\|_{\mathbf{H}_{k_2:\infty}}^2\right]$$

$$\lesssim \frac{k_2^{1+a-b}}{N_{\text{eff}}\gamma} + \sum_{k>k_2} \lambda_i \cdot i^{a-b}$$

$$\lesssim \frac{k_2^{1+a-b}}{N_{\text{eff}}\gamma} + k_2^{1-b}$$

$$\lesssim \max\left\{(N_{\text{eff}}\gamma)^{(1-b)/a}, \, M^{1-b}\right\}$$

when $b < a + 1$, where in the last inequality, we choose $k_2 = \lceil M/3\rceil \wedge (N_{\text{eff}}\gamma)^{1/a}$ to minimize the upper bound.

When $(N_{\text{eff}}\gamma)^{1/a} \le M/c$ for some large constant $c > 0$, combining Lemma D.2 and G.4 yields the lower bound

$$\mathbb{E}_{\mathbf{w}^*}\text{Bias}(\mathbf{w}^*) \gtrsim \sum_{i:\tilde{\lambda}_i < 1/(N_{\text{eff}}\gamma)} \frac{\mu_i(\mathbf{SHH}^{\mathbf{w}}\mathbf{HS}^\top)}{\mu_i(\mathbf{SHS}^\top)} \approx \sum_{i:\tilde{\lambda}_i < 1/(N_{\text{eff}}\gamma)} \frac{\mu_i(\mathbf{SH}^{(a+b)/a}\mathbf{S}^\top)}{\mu_i(\mathbf{SHS}^\top)},$$

$$\gtrsim \sum_{\tilde{\lambda}_i < 1/(N_{\text{eff}}\gamma), i \le M} \frac{i^{-(a+b)}}{i^{-a}} = \sum_{\lambda_i < 1/(N_{\text{eff}}\gamma), i \le M} i^{-b} \gtrsim (N_{\text{eff}}\gamma)^{(1-b)/a}$$

with probability at least $1 - e^{-\Omega(M)}$. Here, the hidden constants depend only on $a, b$.

**Upper bound when** $b \ge a + 1$**.** Following the previous derivations, when $b = a + 1$, we have with probability at least $1 - e^{-\Omega(M)}$

$$\mathbb{E}_{\mathbf{w}^*}\text{Bias}(\mathbf{w}^*) \lesssim \mathbb{E}_{\mathbf{w}^*}\left[\frac{\|\mathbf{w}_{0:k_2}^*\|_2^2}{N_{\text{eff}}\gamma} + \|\mathbf{w}_{k_2:\infty}^*\|_{\mathbf{H}_{k_2:\infty}}^2\right]$$

$$\lesssim \frac{\log k_2}{N_{\text{eff}}\gamma} + k_2^{1-b}$$

$$\lesssim \frac{\log(N_{\text{eff}}\gamma)}{N_{\text{eff}}\gamma} + M^{1-b}$$

where the last line follows by setting $k_2 = \lceil M/3\rceil \wedge (N_{\text{eff}}\gamma)^{1/a}$. When $b > a + 1$, we have with probability at least $1 - e^{-\Omega(M)}$

$$\mathbb{E}_{\mathbf{w}^*}\text{Bias}(\mathbf{w}^*) \lesssim \mathbb{E}_{\mathbf{w}^*}\left[\frac{\|\mathbf{w}_{0:k_2}^*\|_2^2}{N_{\text{eff}}\gamma} + \|\mathbf{w}_{k_2:\infty}^*\|_{\mathbf{H}_{k_2:\infty}}^2\right]$$

$$\lesssim \frac{1}{N_{\text{eff}}\gamma} + k_2^{1-b}$$

$$\lesssim \frac{1}{N_{\text{eff}}\gamma} + M^{1-b},$$

where the last follows by chooing $k_2 = M/3$ to minimize the upper bound.

Note that there exist non-constant gaps between the upper and lower bounds on the bias term in the simple regime where $b \geq a + 1$. We leave a more precise analysis of the bias term for future work.

$\square$

**Lemma D.5** (Bounds on Bias under the logarithmic power-law spectrum). *Suppose Assumption 4 hold and the initial stepsize $\gamma \leq \frac{1}{c\,\text{tr}(\mathbf{SHS}^\top)}$ for some constant $c > 2$. Let $\bar{\mathsf{k}} := \inf\{k : k \log^a k \geq N_{\text{eff}}\gamma\}$. Then with probability at least $1 - e^{-\Omega(M)}$ over the randomness of $\mathbf{S}$*

$$\mathbb{E}_{\mathbf{w}^*}\text{Bias}(\mathbf{w}^*) \lesssim \max\left\{\log^{1-a}(N_{\text{eff}}\gamma), \log^{1-a} M\right\},$$

*and*

$$\mathbb{E}_{\mathbf{w}^*}\text{Bias}(\mathbf{w}^*) \gtrsim \log^{1-a}(N_{\text{eff}}\gamma)$$

*when $(N_{\text{eff}}\gamma) \leq M^c$ for some sufficiently small constant $c > 0$. Here, all constants depend only on the power-law degree $a$.*

*Proof of Lemma D.5.* For the upper bound, using Lemma G.7, D.1 and the assumption that $\mathbb{E}\mathbf{w}_i^{*2} = 1$ for all $i > 0$, with probability at least $1 - e^{-\Omega(M)}$, we have

$$\mathbb{E}_{\mathbf{w}^*}\text{Bias}(\mathbf{w}^*) \lesssim \mathbb{E}_{\mathbf{w}^*}\left[\frac{\|\mathbf{w}_{0:k_2}^*\|_2^2}{N_{\text{eff}}\gamma} + \|\mathbf{w}_{k_2:\infty}^*\|_{\mathbf{H}_{k_2:\infty}}^2\right]$$

$$\lesssim \frac{k_2}{N_{\text{eff}}\gamma} + \sum_{k > k_2} \lambda_i$$

$$\approx \frac{k_2}{N_{\text{eff}}\gamma} + \log^{1-a} k_2$$

$$\lesssim \max\left\{\log^{1-a}(N_{\text{eff}}\gamma), \log^{1-a} M\right\},$$

where in the last inequality, we choose $k_2 = \lceil M/3 \rceil \wedge \left[(N_{\text{eff}}\gamma)/\log^a(N_{\text{eff}}\gamma)\right]$ to minimize the upper bound.

Recall $\mathsf{k}^* \approx M/\log M$ (for example we may define $\mathsf{k}^* = \inf\{k : k \log k \geq M\}$) in Lemma G.6. Combining Lemma D.2 and G.6 gives the lower bound

$$\mathbb{E}_{\mathbf{w}^*}\text{Bias}(\mathbf{w}^*) \gtrsim \sum_{i:\tilde{\lambda}_i < 1/(N_{\text{eff}}\gamma)} \frac{\mu_i(\mathbf{SHH}^{\mathbf{w}}\mathbf{HS}^\top)}{\mu_i(\mathbf{SHS}^\top)} \approx \sum_{i:\tilde{\lambda}_i < 1/(N_{\text{eff}}\gamma)} \frac{\mu_i(\mathbf{SH}^2\mathbf{S}^\top)}{\mu_i(\mathbf{SHS}^\top)},$$

$$\gtrsim \sum_{\tilde{\lambda}_i < 1/(N_{\text{eff}}\gamma), i \leq \mathsf{k}^*} \frac{i^{-2}\log^{-2a} i}{i^{-1}\log^{-a} i} = \sum_{\lambda_i < 1/(N_{\text{eff}}\gamma), i \leq \mathsf{k}^*} i^{-1}\log^{-a} i$$

$$\gtrsim \sum_{i=N_{\text{eff}}\gamma}^{\mathsf{k}^*} i^{-1}\log^{-a} i$$

$$\gtrsim \log^{1-a}(N_{\text{eff}}\gamma) - \log^{1-a}(\mathsf{k}^*)$$

$$\gtrsim \log^{1-a}(N_{\text{eff}}\gamma) - c_1 \log^{1-a}(M)$$

with probability at least $1 - e^{-\Omega(M)}$ for some constant $c_1 > 0$. Here, the (hidden) constants depend only on $a$. Therefore, when $(N_{\text{eff}}\gamma)^{1/a} \leq M^c$ for some sufficiently small constant $c > 0$, we have

$$\mathbb{E}_{\mathbf{w}^*}\text{Bias}(\mathbf{w}^*) \gtrsim \log^{1-a}(N_{\text{eff}}\gamma) - c_1 \log^{1-a}(M) \gtrsim \log^{1-a}(N_{\text{eff}}\gamma).$$

with probability at least $1 - e^{-\Omega(M)}$.

$\square$

# E   Variance error

In this section, we present matching upper and lower bounds on the variance term Var defined in (6) under the power-law or logarithmic power-law spectrum.

**Lemma E.1** (Matching bounds on Var under power-law spectrum). *Under Assumption 2,* Var *defined in Eq. (6) satisfies*

$$\text{Var} \approx \frac{\min\{M,\ (N_{\text{eff}}\gamma)^{1/a}\}}{N_{\text{eff}}}$$

*with probability at least* $1 - e^{-\Omega(M)}$ *over the randomness of* **S**. *Here, the hidden constants only depend on* $a$.

*Proof of Lemma E.1.*  By the definition of Var in Eq. (6) and Lemma G.4, we have

$$\begin{aligned}
\text{Var} &= \frac{\#\{\tilde{\lambda}_j \geq 1/(N_{\text{eff}}\gamma)\} + (N_{\text{eff}}\gamma)^2 \sum_{\tilde{\lambda}_j < 1/(N_{\text{eff}}\gamma)} \tilde{\lambda}_j^2}{N_{\text{eff}}} \\
&\approx \frac{\min\left\{M,\ (N_{\text{eff}}\gamma)^{1/a} + (N_{\text{eff}}\gamma)^2 \cdot (N_{\text{eff}}\gamma)^{(1-2a)/a}\right\}}{N_{\text{eff}}} \\
&\approx \frac{\min\{M,\ (N_{\text{eff}}\gamma)^{1/a}\}}{N_{\text{eff}}}
\end{aligned}$$

with probability at least $1 - e^{-\Omega(M)}$ over the randomness of **S**. Here the hidden constants may depend on $a$. $\quad\square$

**Lemma E.2** (Matching bounds on Var under logarithmic power-law spectrum). *Under Assumption 4,* Var *defined in Eq. (6) satisfies*

$$\text{Var} \approx \frac{\min\{M,\ \bar{k}\}}{N_{\text{eff}}} \approx \frac{\min\{M,\ (N_{\text{eff}}\gamma)/\log^a(N_{\text{eff}}\gamma)\}}{N_{\text{eff}}}$$

*with probability at least* $1 - e^{-\Omega(M)}$ *over the randomness of* **S**, *where* $\bar{k} := \inf\{k :\ k\log^a k \geq (N_{\text{eff}}\gamma)\}$ *and* $\approx$ *hides constants that only depend on* $a$.

*Proof of Lemma E.2.*  Define $k^* = \inf\{k :\ k\log k \geq M\}$ and let $\tilde{D} := \#\{\tilde{\lambda}_j \geq 1/(N_{\text{eff}}\gamma)\} + (N_{\text{eff}}\gamma)^2 \sum_{\tilde{\lambda}_j < 1/(N_{\text{eff}}\gamma)} \tilde{\lambda}_j^2$. By the definition of Var in Eq. (6) and Lemma G.6, we have

$$\begin{aligned}
\text{Var} &= \frac{\#\{\tilde{\lambda}_j \geq 1/(N_{\text{eff}}\gamma)\} + (N_{\text{eff}}\gamma)^2 \sum_{\tilde{\lambda}_j < 1/(N_{\text{eff}}\gamma)} \tilde{\lambda}_j^2}{N_{\text{eff}}} \\
&= \frac{\tilde{D}}{N_{\text{eff}}} \approx \frac{\min\{M, \bar{k}\}}{N_{\text{eff}}}
\end{aligned}$$

with probability at least $1 - e^{-\Omega(M)}$ over the randomness of **S**, where the second line follows from

$$\tilde{D} \gtrsim \#\{\tilde{\lambda}_j \geq 1/(N_{\text{eff}}\gamma)\} \approx \frac{N_{\text{eff}}\gamma}{\log^a(N_{\text{eff}}\gamma)}, \qquad \text{and}$$

$$\tilde{D} \lesssim \frac{N_{\text{eff}}\gamma}{\log^a(N_{\text{eff}}\gamma)} + \frac{(N_{\text{eff}}\gamma)^2}{\log^{2a}(N_{\text{eff}}\gamma)} \cdot \sum_{j:\tilde{\lambda}_j < 1/(N_{\text{eff}}\gamma)} \frac{1}{j^2} \lesssim \frac{N_{\text{eff}}\gamma}{\log^a(N_{\text{eff}}\gamma)}$$

when $\bar{k} \lesssim M$. $\quad\square$

# F   Expected risk of the average of (SGD) iterates

In this section, we study the expected risk of the average of (SGD) iterates. Namely, we consider a fixed stepsize (SGD) procedure where $\gamma_t = \gamma$ and define $\bar{\mathbf{v}}_N := \sum_{i=0}^{N-1} \mathbf{v}_i/N$. Our goal is to derive matching upper and lower bounds $\mathcal{R}(\bar{\mathbf{v}}_N)$ in terms of the sample size $N$ and model size $M$.

Compared with the last iterate of (SGD) with geometrically decaying stepsizes, we show that the average of (SGD) iterates with a fixed stepsize achieves a better risk, in the sense that the effective sample size $N_{\texttt{eff}}$ is replaced by $N$ in the bounds (c.f. Theorem 4.1). This may give improvement up to logarithmic factors.

We start with invoking the following result in [48].

**Theorem F.1** (A variant of Theorem 2.1 and 2.2 in [48]). *Suppose Assumption 1 hold. Consider an $M$-dimensional sketched predictor trained by fixed stepsize (SGD) with $N$ samples. Let $\bar{\mathbf{v}}_N := \sum_{i=0}^{N-1} \mathbf{v}_i/N$ be the average of the iterations of SGD. Assume $\mathbf{v}_0 = \mathbf{0}$ and $\sigma^2 \gtrsim 1$. Conditional on $\mathbf{S}$ and suppose the stepsize $\gamma < 1/(c\,\mathrm{tr}(\mathbf{SHS}^\top))$ for some constant $c > 0$, then there exist* Approx, Bias, Var *such that*

$$\mathbb{E}\mathcal{R}_M(\bar{\mathbf{v}}_N) - \sigma^2 \approx \mathbb{E}_{\mathbf{w}^*}\mathsf{Approx} + \mathsf{Bias} + \sigma^2\mathsf{Var},$$

*where the expectation of $\mathcal{R}_M$ is over $\mathbf{w}^*$ and $(\mathbf{x}_i, y_i)_{i=1}^N$ and*

$$\mathsf{Approx} := \mathbb{E}\xi^2$$
$$= \left\| \left( \mathbf{I} - \mathbf{H}^{\frac{1}{2}}\mathbf{S}^\top \left(\mathbf{SHS}^\top\right)^{-1}\mathbf{SH}^{\frac{1}{2}} \right) \mathbf{H}^{\frac{1}{2}}\mathbf{w}^* \right\|^2,$$

$$\mathbb{E}_{\mathbf{w}^*}(T_1 + T_3) \lesssim \mathsf{Bias} \lesssim \mathbb{E}_{\mathbf{w}^*}(T_2 + T_4),$$
$$\mathsf{Var} \approx \frac{D_{\mathrm{eff},N}}{N},$$

*and*

$$T_1 := \frac{1}{\gamma^2 N^2} \mathrm{tr}\left( \left(\mathbf{I} - (\mathbf{I} - \gamma\mathbf{SHS}^\top)^{N/4}\right)^2 (\mathbf{SHS}^\top)^{-1}\mathbf{B}_0 \right), \tag{27a}$$

$$T_2 := \frac{1}{\gamma^2 N^2} \mathrm{tr}\left( \left(\mathbf{I} - (\mathbf{I} - \gamma\mathbf{SHS}^\top)^N\right)^2 (\mathbf{SHS}^\top)^{-1}\mathbf{B}_0 \right), \tag{27b}$$

$$T_3 := \frac{1}{\gamma N^2} \mathrm{tr}\left( \left(\mathbf{I} - (\mathbf{I} - \gamma\mathbf{SHS}^\top)^{N/4}\right)\mathbf{B}_0 \right) \cdot \mathrm{tr}\left( \left(\mathbf{I} - (\mathbf{I} - \gamma\mathbf{SHS}^\top)^{N/4}\right)^2 \right), \tag{27c}$$

$$T_4 := \frac{1}{\gamma N} \mathrm{tr}\left( \mathbf{B}_0 - (\mathbf{I} - \gamma\mathbf{SHS}^\top)^N \mathbf{B}_0 (\mathbf{I} - \gamma\mathbf{SHS}^\top)^N \right) \cdot \frac{D_{\mathrm{eff},N}}{N}, \tag{27d}$$

$$\mathbf{B}_0 := \mathbf{v}^*\mathbf{v}^{*\top}, \tag{27e}$$

$$D_{\mathrm{eff},N} := \#\{\tilde{\lambda}_j \geq 1/(N\gamma)\} + (N\gamma)^2 \sum_{\tilde{\lambda}_j < 1/(N\gamma)} \tilde{\lambda}_j^2, \tag{27f}$$

*where $(\tilde{\lambda}_j)_{j=1}^M$ are eigenvalue of $\mathbf{SHS}^\top$.*

See Section F.2.1 for the proof.

For $T_i (i = 1, 2, 3, 4)$, we also have the following upper (and lower) bounds.

**Lemma F.2** (Lower bound on $T_1$). *Under the assumptions and notations in Theorem F.1, we have*

$$\mathbb{E}_{\mathbf{w}^*}T_1 \gtrsim \sum_{i:\tilde{\lambda}_i < 1/(\gamma N)} \frac{\mu_i(\mathbf{SH}^2\mathbf{S}^\top)}{\mu_i(\mathbf{SHS}^\top)}$$

*almost surely, where $(\tilde{\lambda}_i)_{i=1}^N$ are eigenvalues of $\mathbf{SHS}^\top$ in non-increasing order.*

See the proof in Section F.2.2.

**Lemma F.3** (Upper bound on $T_2$). *Under the assumptions and notations in Theorem F.1, for any $k \leq M/3$ such that $r(\mathbf{H}) \geq k + M$, we have with probability at least $1 - e^{-\Omega(M)}$ that*

$$T_2 \lesssim \frac{1}{N\gamma}\left[\frac{\mu_{M/2}(\mathbf{A}_k)}{\mu_M(\mathbf{A}_k)}\right]^2 \cdot \|\mathbf{w}_{0:k}^*\|^2 + \|\mathbf{w}_{k:\infty}^*\|_{\mathbf{H}_{k:\infty}}^2,$$

*where $\mathbf{A}_k := \mathbf{S}_{k:\infty}\mathbf{H}_{k:\infty}\mathbf{S}_{k:\infty}^\top$.*

See the proof in Section F.2.3.

**Lemma F.4** (Lower bound on $T_3$). *Under the assumptions and notations in Theorem F.1, we have*

$$\mathbb{E}_{\mathbf{w}^*} T_3 \gtrsim \frac{D_{\mathrm{eff},N}}{N} \cdot \sum_{i:\tilde{\lambda}_i < 1/(\gamma N)} \frac{\mu_i(\mathbf{SH}^2\mathbf{S}^\top)}{\mu_i(\mathbf{SHS}^\top)}$$

*almost surely, where $(\tilde{\lambda}_i)_{i=1}^M$ are eigenvalues of $\mathbf{SHS}^\top$ in non-increasing order.*

See the proof in Section F.2.4.

**Lemma F.5** (Upper bound on $T_4$). *Under the assumptions and notations in Theorem F.1 and assume $r(\mathbf{H}) \geq M$, we have*

$$T_4 \lesssim \|\mathbf{w}^*\|_{\mathbf{H}}^2 \cdot \frac{D_{\mathrm{eff},N}}{N}$$

*almost surely, where $\mathbf{A}_k := \mathbf{S}_{k:\infty}\mathbf{H}_{k:\infty}\mathbf{S}_{k:\infty}^\top$.*

See the proof in Section F.2.5.

With these results at hand, we are ready to derive upper and lower bounds for the risk of the average of (SGD) iterates.

## F.1   Matching bounds for the average of (SGD) iterates under power-law spectrum

In this section, we derive upper and lower bounds for the expected risk under the power-law spectrum. Our main result (Theorem F.6) follows directly from Theorem F.1 and the bounds on $T_i (i = 1, 2, 3, 4)$ in Lemmas F.2 to F.5.

**Theorem F.6** (Scaling law for average iterates of SGD). *Suppose Assumption 1 and 2 hold and $\sigma^2 \lesssim 1$. Then there exists some $a$-dependent constant $c > 0$ such that when $\gamma \leq c$, with probability at least $1 - e^{-\Omega(M)}$ over the randomness of the sketch matrix $\mathbf{S}$, we have*

$$\mathbb{E}\mathcal{R}_M(\bar{\mathbf{v}}_N) = \sigma^2 + \Theta\big(M^{1-a}\big) + \Theta\big((N\gamma)^{1/a-1}\big),$$

*where the expectation is over the randomness of $\mathbf{w}^*$ and $(\mathbf{x}_i, y_i)_{i=1}^N$, and $\Theta(\cdot)$ hides constants that may depend on $a$.*

See the proof in Section F.2.6.

Compared with Theorem 4.1, Theorem F.6 suggests that the average of (SGD) achieves a smaller risk in the sketched linear model—the $(N_{\mathtt{eff}}\gamma)^{1/a}$ is replaced by $(N\gamma)^{1/a}$ in the bound for the bias term. This is intuitive since the sum of stepsizes $\sum_t \gamma_t \asymp N_{\mathtt{eff}}\gamma$ for the geometrically decaying stepsize scheduler while $\sum_t \gamma_t \asymp N\gamma$ for the fixed stepsize scheduler.

We also verify the observations in Theorem F.6 via simulations. We adopt the same model and setup as in Section 5 but use the average of iterates of fixed stepsize (SGD) (denoted by $\bar{\mathbf{v}}_N$) as the predictor. From Figure 3 and 4 we see that the expected risk $\mathbb{E}\mathcal{R}(\bar{\mathbf{v}}_N)$ also scales following a power-law relation in both sample size $N$ and model size $M$. Moreover, the fitted exponents match our theoretical predictions in Theorem F.6.

## F.2   Proofs

### F.2.1   Proof of Theorem F.1

Similar to the proof of Theorem A.4, we have the decomposition

$$\mathcal{R}(\bar{\mathbf{v}}_N) = \sigma^2 + \mathsf{Approx} + \|\bar{\mathbf{v}}_N - \mathbf{v}^*\|_{\mathbf{SHS}^\top}^2.$$

Note that $(\mathbf{v}_t)_{t=1}^N$ can also be viewed as the SGD iterates on the model $y = \langle \mathbf{Sx}, \mathbf{v}^* \rangle + \xi + \epsilon$, where the noise satisfies

$$\mathbb{E}(\xi + \epsilon)^2 = \mathcal{R}(\mathbf{v}^*) = \mathbb{E}\xi^2 + \sigma^2.$$

Therefore, the upper and lower bounds on Bias, Var follow directly from the proof of Theorem 2.1, 2.2 and related lemmas (Lemma B.6, B.11, C.3, C.5) in [48].

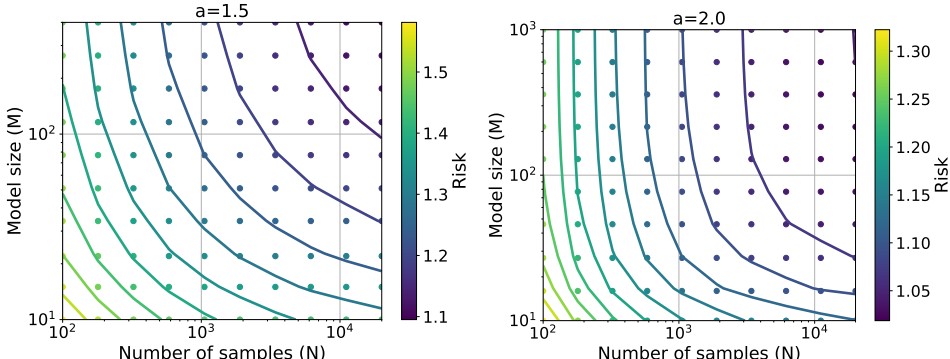

Figure 3: The expected risk (Risk) of the average of iterates of (SGD) versus the sample size $N$ and the model size $M$ for different power-law degrees $a$. The expected risk is computed by averaging over 1000 independent samples of $(\mathbf{w}^*, \mathbf{S})$. We fit the expected risk using the formula Risk $\sim \sigma^2 + c_1/M^{a_1} + c_2/N^{a_2}$ via minimizing the Huber loss as in [21]. Parameters: $\sigma = 1, \gamma = 0.1$. Left: For $a = 1.5, d = 20000$, the fitted exponents are $(a_1, a_2) = (0.59, 0.33) \approx (0.5, 0.33)$. Right: For $a = 2, d = 2000$, the fitted exponents are $(a_1, a_2) = (1.09, 0.49) \approx (1.0, 0.5)$. Note that the values of $(a_1, a_2)$ are close to our theoretical predictions $(a - 1, 1 - 1/a)$ in both cases.

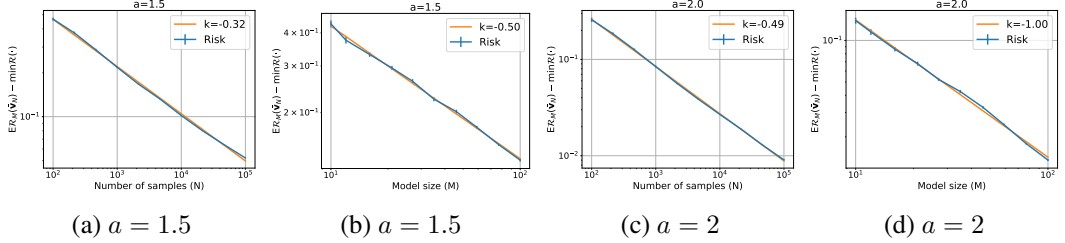

(a) $a = 1.5$      (b) $a = 1.5$      (c) $a = 2$      (d) $a = 2$

Figure 4: The expected risk of the average of iterates of (SGD) minus the irreducible risk versus the effective sample size and model size. Parameters $\sigma = 1, \gamma = 0.1$. (a), (b): $a = 1.5, d = 10000$; (c), (d): $a = 2, d = 1000$. The errobars denotes the $\pm 1$ standard deviation of estimating the expected risk using 100 independent samples of $(\mathbf{w}^*, \mathbf{S})$. We use linear functions to fit the expected risk under log-log scale and report the slope of the fitted lines (denoted by $k$).

### F.2.2   Proof of Lemma F.2

Let $f_1(A) := (\mathbf{I} - (\mathbf{I} - \gamma A)^{N/4})^2 A^{-1}/\gamma^2/N^2$ for any positive definite matrix $A \in \mathbb{R}^{M \times M}$. Since $\gamma \leq 1/(c\operatorname{tr}(\mathbf{SHS}^\top))$, we have $f_1(\mathbf{SHS}^\top) \succeq \mathbf{0}$. By definition of $T_1$ and recalling $\mathbf{v}^* = (\mathbf{SHS}^\top)^{-1}\mathbf{SHw}^*$, we have with probability at least $1 - e^{-\Omega(M)}$ that

$$\mathbb{E}_{\mathbf{w}^*} T_1 = \mathbb{E}_{\mathbf{w}^*}[\mathbf{w}^{*\top}\mathbf{HS}^\top(\mathbf{SHS}^\top)^{-1}f_1(\mathbf{SHS}^\top)(\mathbf{SHS}^\top)^{-1}\mathbf{SHw}^*]$$
$$= \operatorname{tr}\left([(\mathbf{SHS}^\top)^{-1}f_1(\mathbf{SHS}^\top)(\mathbf{SHS}^\top)^{-1}](\mathbf{SH}^2\mathbf{S}^\top)\right).$$

Following the proof of Lemma D.2 (by Von Neumann's trace inequality), we have

$$\mathbb{E}_{\mathbf{w}^*} T_1 \geq \sum_{i=1}^{M} \frac{\mu_i(\mathbf{SH}^2\mathbf{S}^\top)}{\mu_i\left((\mathbf{SHS}^\top)^2 f_1(\mathbf{SHS}^\top)^{-1}\right)}$$
$$\geq \sum_{i:\tilde{\lambda}_i < 1/(\gamma N)} \frac{\mu_i(\mathbf{SH}^2\mathbf{S}^\top)}{\mu_i\left((\mathbf{SHS}^\top)^2 f_1(\mathbf{SHS}^\top)^{-1}\right)}$$
$$\gtrsim \sum_{i:\tilde{\lambda}_i < 1/(\gamma N)} \frac{\mu_i(\mathbf{SH}^2\mathbf{S}^\top)}{\mu_i(\mathbf{SHS}^\top)},$$

where the third inequality is due to

$$\lambda/f_1(\lambda) \lesssim \frac{\lambda^2\gamma^2 N^2}{(1-(1-\gamma\lambda)^{N/4})^2} \lesssim \frac{N^2}{(\sum_{i=0}^{N/4-1}(1-\gamma\lambda)^i)^2} \lesssim \frac{1}{(1-\gamma\lambda)^{2N}} \lesssim 1$$

when $\lambda < 1/(N\gamma)$.

### F.2.3   Proof of Lemma F.3

By definition of $T_2$, the fact that $1-x^N = (1-x)\sum_{i=0}^{N-1}x^i$, and recalling $\mathbf{v}^* = (\mathbf{SHS}^\top)^{-1}\mathbf{SHw}^*$, we have

$$T_2 = \mathbf{w}^{*\top}\mathbf{HS}^\top f_2(\mathbf{SHS}^\top)\mathbf{SHw}^*,$$
$$\leq 2[\underbrace{\mathbf{w}_{0:k}^{*\top}\mathbf{H}_{0:k}\mathbf{S}_{0:k}^\top f_2(\mathbf{SHS}^\top)\mathbf{S}_{0:k}\mathbf{H}_{0:k}\mathbf{w}_{0:k}^*}_{T_{21}} + \underbrace{\mathbf{w}_{k:\infty}^{*\top}\mathbf{H}_{k:\infty}\mathbf{S}_{k:\infty}^\top f_2(\mathbf{SHS}^\top)\mathbf{S}_{k:\infty}\mathbf{H}_{k:\infty}\mathbf{w}_{k:\infty}^*}_{T_{22}}],$$

where $f_2(A) := [\sum_{i=0}^{N-1}(\mathbf{I}-\gamma A)^i]^2/A/N^2$ for any symmetric matrix $A \in \mathbb{R}^{M\times M}$. Moreover, we have

$$T_{21} = \mathbf{w}_{0:k}^{*\top}\mathbf{H}_{0:k}\mathbf{S}_{0:k}^\top f_2(\mathbf{SHS}^\top)\mathbf{S}_{0:k}\mathbf{H}_{0:k}\mathbf{w}_{0:k}^*$$
$$\leq \|f_2(\mathbf{SHS}^\top)(\mathbf{SHS}^\top)^2\| \cdot \|(\mathbf{SHS}^\top)^{-1}\mathbf{S}_{0:k}\mathbf{H}_{0:k}\mathbf{w}_{0:k}^*\|^2.$$

Using the assumption on the stepsize that $\gamma \leq 1/(c\operatorname{tr}(\mathbf{SHS}^\top))$, we have

$$\|f_2(\mathbf{SHS}^\top)(\mathbf{SHS}^\top)^2\| \leq \max_{\lambda\in[0,1/\gamma]}\frac{1}{N^2}\Big[\sum_{i=0}^{N-1}(1-\gamma\lambda)^i\Big]^2\lambda$$
$$= \max_{\lambda\in[0,1/\gamma]}\frac{1}{N^2\gamma}\Big[\sum_{i=0}^{N-1}(1-\gamma\lambda)^i\Big]\cdot(1-(1-\gamma\lambda)^N)$$
$$\leq \frac{1}{N^2\gamma}\cdot N\cdot 1 = \frac{1}{N\gamma}. \qquad (28)$$

Combining Eq. (28) with Eq. (23) in the proof of Lemma D.1 (note that we assume $k \leq M/3$), we obtain

$$T_{21} \leq c\frac{1}{N\gamma}\Big[\frac{\mu_{M/2}(\mathbf{A}_k)}{\mu_M(\mathbf{A}_k)}\Big]^2\cdot\|\mathbf{w}_{0:k}^*\|^2$$

for some constant $c > 0$ with probability at least $1 - e^{-\Omega(M)}$. For $T_{22}$, we have

$$T_{22} = \mathbf{w}_{k:\infty}^{*\top}\mathbf{H}_{k:\infty}\mathbf{S}_{k:\infty}^\top f_2(\mathbf{SHS}^\top)\mathbf{S}_{k:\infty}\mathbf{H}_{k:\infty}\mathbf{w}_{k:\infty}^*$$
$$\leq \|f_2(\mathbf{SHS}^\top)\mathbf{SHS}^\top\|\cdot\|(\mathbf{SHS}^\top)^{-1/2}(\mathbf{S}_{k:\infty}\mathbf{H}_{k:\infty}^{1/2})\mathbf{H}_{k:\infty}^{1/2}\mathbf{w}_{k:\infty}^*\|^2$$
$$\leq \|f_2(\mathbf{SHS}^\top)\mathbf{SHS}^\top\|\cdot\|(\mathbf{SHS}^\top)^{-1/2}\mathbf{S}_{k:\infty}\mathbf{H}_{k:\infty}^{1/2}\|^2\cdot\|\mathbf{w}_{k:\infty}^*\|_{\mathbf{H}_{k:\infty}}^2.$$

Since $\|f_2(\mathbf{SHS}^\top)\mathbf{SHS}^\top\| = \|[\sum_{i=0}^{N-1}(\mathbf{I}-\gamma\mathbf{SHS}^\top)^i]^2/N^2\| \leq 1$ by the assumption $\gamma \leq 1/(c\operatorname{tr}(\mathbf{SHS}^\top))$, and

$$\|(\mathbf{SHS}^\top)^{-1/2}\mathbf{S}_{k:\infty}\mathbf{H}_{k:\infty}^{1/2}\|^2 = \|\mathbf{H}_{k:\infty}^{1/2}\mathbf{S}_{k:\infty}^\top(\mathbf{SHS}^\top)^{-1}\mathbf{S}_{k:\infty}\mathbf{H}_{k:\infty}^{1/2}\|$$
$$= \|\mathbf{H}_{k:\infty}^{1/2}\mathbf{S}_{k:\infty}^\top(\mathbf{S}_{0:k}\mathbf{H}_{0:k}\mathbf{S}_{0:k}^\top + \mathbf{S}_{k:\infty}\mathbf{H}_{k:\infty}\mathbf{S}_{k:\infty}^\top)^{-1}\mathbf{S}_{k:\infty}\mathbf{H}_{k:\infty}^{1/2}\|$$
$$\leq \|\mathbf{H}_{k:\infty}^{1/2}\mathbf{S}_{k:\infty}^\top(\mathbf{S}_{k:\infty}\mathbf{H}_{k:\infty}\mathbf{S}_{k:\infty}^\top)^{-1}\mathbf{S}_{k:\infty}\mathbf{H}_{k:\infty}^{1/2}\| = 1,$$

it follows that $T_{22} \leq \|\mathbf{w}_{k:\infty}^*\|_{\mathbf{H}_{k:\infty}}^2$. Combining the bounds on $T_{21}, T_{22}$ completes the proof.

### F.2.4   Proof of Lemma F.4

Let $f_3(A) := (\mathbf{I} - (\mathbf{I}-\gamma A)^{N/4})/\gamma/N^2$ for any positive definite matrix $A \in \mathbb{R}^{M\times M}$. Following the same arguments as in the proof of Lemma F.2, we have $f_3(\mathbf{SHS}^\top) \succeq \mathbf{0}$ and

$$\mathbb{E}_{\mathbf{w}^*}\Big[\frac{1}{\gamma N^2}\operatorname{tr}\Big(\Big(\mathbf{I}-(\mathbf{I}-\gamma\mathbf{SHS}^\top)^{N/4}\Big)\mathbf{B}_0\Big)\Big]$$

$$= \mathbb{E}_{\mathbf{w}^*}[\mathbf{w}^{*\top}\mathbf{HS}^\top(\mathbf{SHS}^\top)^{-1}f_3(\mathbf{SHS}^\top)(\mathbf{SHS}^\top)^{-1}\mathbf{SHw}^*]$$
$$= \operatorname{tr}((\mathbf{SHS}^\top)^{-1}f_3(\mathbf{SHS}^\top)(\mathbf{SHS}^\top)^{-1}\mathbf{SH}^2\mathbf{S}^\top).$$

Moreover,

$$\mathbb{E}_{\mathbf{w}^*}T_1 \geq \sum_{i=1}^{M} \frac{\mu_i(\mathbf{SH}^2\mathbf{S}^\top)}{\mu_i\Big((\mathbf{SHS}^\top)^2 f_3(\mathbf{SHS}^\top)^{-1}\Big)}$$

$$\geq \sum_{i:\tilde{\lambda}_i < 1/(\gamma N)} \frac{\mu_i(\mathbf{SH}^2\mathbf{S}^\top)}{\mu_i\Big((\mathbf{SHS}^\top)^2 f_3(\mathbf{SHS}^\top)^{-1}\Big)}$$

$$\gtrsim \frac{1}{N} \sum_{i:\tilde{\lambda}_i < 1/(\gamma N)} \frac{\mu_i(\mathbf{SH}^2\mathbf{S}^\top)}{\mu_i(\mathbf{SHS}^\top)}, \qquad (29)$$

where the third inequality is due to

$$\lambda/f_3(\lambda) \lesssim \frac{\lambda\gamma N^2}{1-(1-\gamma\lambda)^{N/4}} \lesssim \frac{N^2}{\sum_{i=0}^{N/4-1}(1-\gamma\lambda)^i} \lesssim \frac{N}{(1-\gamma\lambda)^N} \lesssim N$$

when $\lambda < 1/(N\gamma)$. Note that

$$1-(1-\gamma\tilde{\lambda}_i)^{\frac{N}{4}} \geq \begin{cases} 1-(1-\frac{1}{N})^{\frac{N}{4}} \geq 1-e^{-\frac{1}{4}} \geq \frac{1}{5}, & \tilde{\lambda}_i \geq \frac{1}{\gamma N}, \\ \frac{N}{4}\cdot\gamma\tilde{\lambda}_i - \frac{N(N-4)}{32}\cdot\gamma^2\tilde{\lambda}_i^2 \geq \frac{N}{5}\cdot\gamma\tilde{\lambda}_i, & \tilde{\lambda}_i < \frac{1}{\gamma N}, \end{cases} \geq \frac{1}{5}\min\{N\gamma\tilde{\lambda}_i, 1\}. \qquad (30)$$

We thus have

$$\operatorname{tr}\left(\left(\mathbf{I}-(\mathbf{I}-\gamma\mathbf{SHS}^\top)^{N/4}\right)^2\right) = \sum_{i=1}^{M}[1-(1-\gamma\tilde{\lambda}_i)^{\frac{N}{4}}]^2 \gtrsim \sum_{i=1}^{M}\min\{(N\gamma\tilde{\lambda}_i)^2, 1\}$$

$$= \#\{\tilde{\lambda}_i \geq \frac{1}{N\gamma}\} + N^2\gamma^2 \sum_{\tilde{\lambda}_i < 1/(N\gamma)} \tilde{\lambda}_i^2 = D_{\text{eff},N}. \qquad (31)$$

Combining Eq. (31) and (29) completes the proof.

### F.2.5 Proof of Lemma F.5

Substituting $\mathbf{v}^* = (\mathbf{SHS}^\top)^{-1}\mathbf{SHw}^*$ in the expression of $T_4$ and noting $\mathbf{v}_0 = \mathbf{0}$, we have

$$T_4 = \frac{1}{\gamma N}\operatorname{tr}\left(\mathbf{B}_0 - (\mathbf{I}-\gamma\mathbf{SHS}^\top)^N\mathbf{B}_0(\mathbf{I}-\gamma\mathbf{SHS}^\top)^N\right) \cdot \frac{D_{\text{eff},N}}{N}$$

$$= \frac{1}{\gamma N}\operatorname{tr}\left(\mathbf{w}^{*\top}\mathbf{HS}^\top(\mathbf{SHS}^\top)^{-1}[\mathbf{I}-(\mathbf{I}-\gamma\mathbf{SHS}^\top)^{2N}](\mathbf{SHS}^\top)^{-1}\mathbf{SHw}^*\right) \cdot \frac{D_{\text{eff},N}}{N}$$

$$=: \operatorname{tr}\left(\mathbf{w}^{*\top}\mathbf{HS}^\top f_4(\mathbf{SHS}^\top)\mathbf{SHw}^*\right) \cdot \frac{D_{\text{eff},N}}{N}, \qquad (32)$$

where $f_4(A) := A^{-1}\big[\mathbf{I}-(\mathbf{I}-\gamma A)^{2N}\big]A^{-1}/(N\gamma)$ for any symmetric matrix $A \in \mathbb{R}^{M\times M}$. Moreover,

$$\operatorname{tr}\left(\mathbf{w}^{*\top}\mathbf{HS}^\top f_4(\mathbf{SHS}^\top)\mathbf{SHw}^*\right)$$
$$\leq \|f_4(\mathbf{SHS}^\top)\mathbf{SHS}^\top\| \cdot \|(\mathbf{SHS}^\top)^{-1/2}\mathbf{SH}^{1/2}\|^2 \cdot \|\mathbf{w}_*\|_{\mathbf{H}}^2$$
$$\leq \|f_4(\mathbf{SHS}^\top)\mathbf{SHS}^\top\| \cdot \|\mathbf{w}_*\|_{\mathbf{H}}^2.$$

Since

$$\|f_4(\mathbf{SHS}^\top)\mathbf{SHS}^\top\| = \frac{1}{N}\|\sum_{i=0}^{2N-1}(\mathbf{I}-\gamma\mathbf{SHS}^\top)^i\| \leq 2$$

by our assumption on the stepsize, it follows that

$$\operatorname{tr}\left(\mathbf{w}^{*\top}\mathbf{HS}^\top f_4(\mathbf{SHS}^\top)\mathbf{SHw}^*\right) \lesssim \|\mathbf{w}_*\|_{\mathbf{H}}^2. \qquad (33)$$

Combining Eq. (32) and (33) we find

$$T_4 \lesssim \|\mathbf{w}_*\|_{\mathbf{H}}^2 \cdot \frac{D_{\text{eff},N}}{N}.$$

### F.2.6 Proof of Theorem F.6

First, by Lemma G.4 we have $1/\operatorname{tr}(\mathbf{SHS}^\top) \gtrsim c_1$ for some $a$-dependent $c_1 > 0$ with probability at least $1 - e^{-\Omega(M)}$. Therefore we may choose $c$ sufficiently small so that $\gamma \leq c$ implies $\gamma \lesssim 1/\operatorname{tr}(\mathbf{SHS}^\top)$ with probability at least $1 - e^{-\Omega(M)}$. Now, suppose we have $\gamma \lesssim 1/\operatorname{tr}(\mathbf{SHS}^\top)$. Following the notations in Theorem F.1, we claim the following bounds on Approx, Bias, Var:

$$\mathbb{E}\mathsf{Approx} \asymp M^{1-a} \tag{34a}$$

$$\mathsf{Var} \asymp \min\left\{ M, \ (N\gamma)^{1/a} \right\}/N. \tag{34b}$$

$$\mathsf{Bias} \lesssim \max\left\{ M^{1-a}, \ (N\gamma)^{1/a-1} \right\}, \tag{34c}$$

$$\mathsf{Bias} \gtrsim (N\gamma)^{1/a-1} \text{ when } (N\gamma)^{1/a} \leq M/c \text{ for some constant } c > 0, \tag{34d}$$

with probability at least $1 - e^{-\Omega(M)}$. Putting the bounds together yields Theorem F.6.

**Proof of claim (34a)** Note that our definition of Approx in Thereom F.1 is the same as that in Eq. (4) (and 7). Therefore the claim follows immediately from Lemma C.4.

**Proof of claim (34b)** This follows from the proof of Lemma E.1 with $N_{\texttt{eff}}$ replaced by $N$.

**Proof of claim (34c)** By Theorem F.1, Lemma F.3 and F.5, we have

$$\mathsf{Bias} \lesssim \mathbb{E}_{\mathbf{w}^*} \frac{\|\mathbf{w}^*_{0:k_2}\|_2^2}{N\gamma} \cdot \left[ \frac{\mu_{M/2}(\mathbf{S}_{k_2:\infty}\mathbf{H}_{k_2:\infty}\mathbf{S}^\top_{k_2:\infty})}{\mu_M(\mathbf{S}_{k_2:\infty}\mathbf{H}_{k_2:\infty}\mathbf{S}^\top_{k_2:\infty})} \right]^2 + \mathbb{E}_{\mathbf{w}^*}\|\mathbf{w}^*_{k_2:\infty}\|^2_{\mathbf{H}_{k_2:\infty}} + \sigma^2 \frac{D_{\mathrm{eff},N}}{N},$$

$$\lesssim \frac{k_2}{N\gamma} \left[ \frac{\mu_{M/2}(\mathbf{S}_{k_2:\infty}\mathbf{H}_{k_2:\infty}\mathbf{S}^\top_{k_2:\infty})}{\mu_M(\mathbf{S}_{k_2:\infty}\mathbf{H}_{k_2:\infty}\mathbf{S}^\top_{k_2:\infty})} \right]^2 + k_2^{1-a} + \frac{D_{\mathrm{eff},N}}{N}$$

with probability at least $1 - e^{-\Omega(M)}$ for any $k_2 \leq M/3$. Choosing $k_2 = \min\{M/3, (N\gamma)^{1/a}\}$ and using Lemma G.4 and claim (34b), we obtain

$$\mathsf{Bias} \lesssim \max\left\{ M^{1-a}, \ (N\gamma)^{1/a-1} \right\} + \frac{\min\left\{ M, \ (N\gamma)^{1/a} \right\}}{N} \lesssim \max\left\{ M^{1-a}, \ (N\gamma)^{1/a-1} \right\} + (N\gamma)^{1/a-1}$$

$$\lesssim \max\left\{ M^{1-a}, \ (N\gamma)^{1/a-1} \right\}$$

with probability at least $1 - e^{-\Omega(M)}$.

**Proof of claim (34d)** By Theorem F.1 and Lemma F.2, we have

$$\mathbb{E}_{\mathbf{w}^*}\mathsf{Bias} \gtrsim \sum_{i:\tilde{\lambda}_i < 1/(\gamma N)} \frac{\mu_i(\mathbf{SH}^2\mathbf{S}^\top)}{\mu_i(\mathbf{SHS}^\top)}.$$

When $(N\gamma)^{1/a} \leq M/c$ for some large constant $c > 0$, we have from Lemma G.4 that

$$\mathbb{E}_{\mathbf{w}^*}\mathsf{Bias} \gtrsim \sum_{i:\tilde{\lambda}_i < 1/(\gamma N)} \frac{i^{-2a}}{i^{-a}} = \sum_{i:\tilde{\lambda}_i < 1/(\gamma N)} i^{-a} \gtrsim [(N\gamma)^{1/a}]^{1-a} = (N\gamma)^{1/a-1}$$

with probability at least $1 - e^{-\Omega(M)}$.

## G  Concentration lemmas

### G.1  General concentration results

**Lemma G.1.** *Suppose that $\mathbf{S} \in \mathbb{R}^{M \times d}$ is such that* [3]

$$\mathbf{S}_{ij} \sim \mathcal{N}(0, 1/M).$$

---

[3] We allow $d = \infty$.

*Let $(\lambda_i)_{i\geq 1}$ be the eigenvalues of $\mathbf{H}$ in non-increasing order. Let $(\tilde{\lambda}_i)_{i=1}^M$ be the eigenvalues of $\mathbf{SHS}^\top$ in non-increasing order. Then there exists a constant $c > 1$ such that for every $M \geq 0$ and every $0 \leq k \leq M$, with probability $\geq 1 - e^{-\Omega(M)}$, we have*

$$\text{for every } j \leq M, \quad \left|\tilde{\lambda}_j - \left(\lambda_j + \frac{\sum_{i>k}\lambda_i}{M}\right)\right| \leq c \cdot \left(\sqrt{\frac{k}{M}} \cdot \lambda_j + \lambda_{k+1} + \sqrt{\frac{\sum_{i>k}\lambda_i^2}{M}}\right).$$

*As a direct consequence, for $k \leq M/c^2$, we have*

$$\text{for every } j \leq M, \quad \left|\tilde{\lambda}_j - \left(\lambda_j + \frac{\sum_{i>k}\lambda_i}{M}\right)\right| \leq \frac{1}{2}\cdot\left(\lambda_j + \frac{\sum_{i>k}\lambda_i}{M}\right) + c_1 \cdot \lambda_{k+1},$$

*where $c_1 = c + 2c^2$.*

*Proof of Lemma G.1.* We have the following decomposition motivated by Swartworth and Woodruff [38] (see their Section 3.4, Proof of Theorem 1).

$$\mathbf{SHS}^\top = \mathbf{S}_{0:k}\mathbf{H}_{0:k}\mathbf{S}_{0:k}^\top + \mathbf{S}_{k:\infty}\mathbf{H}_{k:\infty}\mathbf{S}_{k:\infty}^\top$$
$$= \mathbf{S}_{0:k}\mathbf{H}_{0:k}\mathbf{S}_{0:k}^\top + \frac{\sum_{i>k}\lambda_i}{M}\cdot\mathbf{I}_M + \mathbf{S}_{k:\infty}\mathbf{H}_{k:\infty}\mathbf{S}_{k:\infty}^\top - \frac{\sum_{i>k}\lambda_i}{M}\cdot\mathbf{I}_M.$$

We remark that this decomposition idea has been implicitly used in Bartlett et al. [5] to control the eigenvalues of a Gram matrix. In fact, we will use techniques from Bartlett et al. [5] to obtain a sharper bound than that presented in Swartworth and Woodruff [38].

For the upper bound, we have

$$\mu_j(\mathbf{SHS}^\top) \leq \mu_j\left(\mathbf{S}_{0:k}\mathbf{H}_{0:k}\mathbf{S}_{0:k}^\top + \frac{\sum_{i>k}\lambda_i}{M}\cdot\mathbf{I}_M\right) + \left\|\mathbf{S}_{k:\infty}\mathbf{H}_{k:\infty}\mathbf{S}_{k:\infty}^\top - \frac{\sum_{i>k}\lambda_i}{M}\cdot\mathbf{I}_M\right\|_2$$
$$= \mu_j\left(\mathbf{S}_{0:k}\mathbf{H}_{0:k}\mathbf{S}_{0:k}^\top\right) + \frac{\sum_{i>k}\lambda_i}{M}\cdot\mathbf{I}_M + \left\|\mathbf{S}_{k:\infty}\mathbf{H}_{k:\infty}\mathbf{S}_{k:\infty}^\top - \frac{\sum_{i>k}\lambda_i}{M}\cdot\mathbf{I}_M\right\|_2$$
$$\leq \mu_j\left(\mathbf{S}_{0:k}\mathbf{H}_{0:k}\mathbf{S}_{0:k}^\top\right) + \frac{\sum_{i>k}\lambda_i}{M}\cdot\mathbf{I}_M + c_1 \cdot \left(\lambda_{k+1} + \sqrt{\frac{\sum_{i>k}\lambda_i^2}{M}}\right),$$

where the last inequality is by Lemma G.2. For $j \leq k$, using Lemma G.3, we have

$$\mu_j\left(\mathbf{S}_{0:k}\mathbf{H}_{0:k}\mathbf{S}_{0:k}^\top\right) \leq \lambda_j + c_2 \cdot \sqrt{\frac{k}{M}} \cdot \lambda_j.$$

For $k < j \leq M$, we have

$$\mu_j\left(\mathbf{S}_{0:k}\mathbf{H}_{0:k}\mathbf{S}_{0:k}^\top\right) = 0 \leq \lambda_j + c_2 \cdot \sqrt{\frac{k}{M}} \cdot \lambda_j.$$

Putting these together, we have the following for every $j = 1,\ldots,M$:

$$\mu_j(\mathbf{SHS}^\top) \leq \mu_j\left(\mathbf{S}_{0:k}\mathbf{H}_{0:k}\mathbf{S}_{0:k}^\top\right) + \frac{\sum_{i>k}\lambda_i}{M}\cdot\mathbf{I}_M + c_1 \cdot \left(\lambda_{k+1} + \sqrt{\frac{\sum_{i>k}\lambda_i^2}{M}}\right)$$
$$\leq \lambda_j + \frac{\sum_{i>k}\lambda_i}{M}\cdot\mathbf{I}_M + c \cdot \left(\sqrt{\frac{k}{M}}\cdot\lambda_j + \lambda_{k+1} + \sqrt{\frac{\sum_{i>k}\lambda_i^2}{M}}\right).$$

Similarly, we can show the lower bound. By the decomposition, we have

$$\mu_j(\mathbf{SHS}^\top) \geq \mu_j\left(\mathbf{S}_{0:k}\mathbf{H}_{0:k}\mathbf{S}_{0:k}^\top + \frac{\sum_{i>k}\lambda_i}{M}\cdot\mathbf{I}_M\right) - \left\|\mathbf{S}_{k:\infty}\mathbf{H}_{k:\infty}\mathbf{S}_{k:\infty}^\top - \frac{\sum_{i>k}\lambda_i}{M}\cdot\mathbf{I}_M\right\|$$
$$= \mu_j\left(\mathbf{S}_{0:k}\mathbf{H}_{0:k}\mathbf{S}_{0:k}^\top\right) + \frac{\sum_{i>k}\lambda_i}{M}\cdot\mathbf{I}_M - \left\|\mathbf{S}_{k:\infty}\mathbf{H}_{k:\infty}\mathbf{S}_{k:\infty}^\top - \frac{\sum_{i>k}\lambda_i}{M}\cdot\mathbf{I}_M\right\|$$

$$\geq \mu_j\left(\mathbf{S}_{0:k}\mathbf{H}_{0:k}\mathbf{S}_{0:k}^\top\right) + \frac{\sum_{i>k}\lambda_i}{M} \cdot \mathbf{I}_M - c_1 \cdot \left(\lambda_{k+1} + \sqrt{\frac{\sum_{i>k}\lambda_i^2}{M}}\right),$$

where the last inequality is by Lemma G.2. For $j \leq k$, using Lemma G.3, we have

$$\mu_j\left(\mathbf{S}_{0:k}\mathbf{H}_{0:k}\mathbf{S}_{0:k}^\top\right) \geq \lambda_j - c_2 \cdot \sqrt{\frac{k}{M}} \cdot \lambda_j.$$

For $k < j \leq M$, we have

$$\mu_j\left(\mathbf{S}_{0:k}\mathbf{H}_{0:k}\mathbf{S}_{0:k}^\top\right) = 0 \geq \lambda_j - \lambda_{k+1} - c_2 \cdot \sqrt{\frac{k}{M}} \cdot \lambda_j,$$

where the last inequality is due to $\lambda_j \leq \lambda_k$ for $j \geq k$. Putting these together, we have

$$\mu_j\left(\mathbf{S}\mathbf{H}\mathbf{S}^\top\right) \geq \mu_j\left(\mathbf{S}_{0:k}\mathbf{H}_{0:k}\mathbf{S}_{0:k}^\top\right) + \frac{\sum_{i>k}\lambda_i}{M} \cdot \mathbf{I}_M - c_1 \cdot \left(\lambda_{k+1} + \sqrt{\frac{\sum_{i>k}\lambda_i^2}{M}}\right)$$

$$\geq \lambda_j + \frac{\sum_{i>k}\lambda_i}{M} \cdot \mathbf{I}_M - c \cdot \left(\sqrt{\frac{k}{M}} \cdot \lambda_j + \lambda_{k+1} + \sqrt{\frac{\sum_{i>k}\lambda_i^2}{M}}\right).$$

So far, we have proved the first claim. To show the second claim, we simply apply

$$c \cdot \sqrt{\frac{k}{M}} \leq \frac{1}{2} \quad \text{for } k \leq M/c^2,$$

and

$$c \cdot \sqrt{\frac{\sum_{i>k}\lambda_i^2}{M}} \leq c \cdot \sqrt{\frac{\sum_{i>k}\lambda_i}{M} \cdot \lambda_{k+1}}$$

$$\leq \frac{1}{2} \cdot \frac{\sum_{i>k}\lambda_i}{M} + 2c^2 \cdot \lambda_{k+1},$$

in the first claim. $\qquad\square$

**Lemma G.2** (Tail concentration, Lemma 26 in Bartlett et al. [5])**.** *For any $k \geq 1$, with probability at least $1 - e^{-\Omega(M)}$, we have*

$$\left\|\mathbf{S}_{k:\infty}\mathbf{H}_{k:\infty}\mathbf{S}_{k:\infty}^\top - \frac{\sum_{i>k}\lambda_i}{M} \cdot \mathbf{I}_M\right\|_2 \lesssim \lambda_{k+1} + \sqrt{\frac{\sum_{i>k}\lambda_i^2}{M}}.$$

*Moreover, the minimum eigenvalue of $\mathbf{S}_{k:\infty}\mathbf{H}_{k:\infty}\mathbf{S}_{k:\infty}^\top$ satisfies*

$$\mu_{\min}\left(\mathbf{S}_{k:\infty}\mathbf{H}_{k:\infty}\mathbf{S}_{k:\infty}^\top\right) \gtrsim \lambda_{k+2M}$$

*with probability at least $1 - e^{-\Omega(M)}$.*

*Proof of Lemma G.2.* The first part of Lemma G.2 is a version of Lemma 26 in [5] (see their proof). We provide proof here for completeness.

We write $\mathbf{S} \in \mathbb{R}^{M \times p}$ as

$$\mathbf{S} = (\mathbf{s}_1 \quad \dots \quad \mathbf{s}_p), \quad \mathbf{s}_i \sim \mathcal{N}\left(0, \frac{1}{M} \cdot \mathbf{I}_M\right), \quad i \geq 1.$$

Since Gaussian distribution is rotational invariance, without loss of generality, we may assume

$$\mathbf{H} = \operatorname{diag}\{\lambda_1, \dots, \lambda_p\}.$$

Then we have

$$\mathbf{S}_{k:\infty}\mathbf{H}_{k:\infty}\mathbf{S}_{k:\infty}^\top = \sum_{i>k}\lambda_i\mathbf{s}_i\mathbf{s}_i^\top.$$

Fixing a unit vector $\mathbf{v} \in \mathbb{R}^M$, then

$$\mathbf{v}^\top \mathbf{S}_{k:\infty} \mathbf{H}_{k:\infty} \mathbf{S}_{k:\infty}^\top \mathbf{v} = \sum_{i>k} \lambda_i \left( \mathbf{s}_i^\top \mathbf{v} \right)^2,$$

where each $\mathbf{s}_i^\top \mathbf{v}$ is $(1/M)$-subGaussian. By Bernstein's inequality, we have, with probability $\geq 1 - \delta$,

$$\left| \sum_{i>k} \lambda_i \left( \mathbf{s}_i^\top \mathbf{v} \right)^2 - \frac{\sum_{i>k} \lambda_i}{M} \right| \lesssim \frac{1}{M} \cdot \left( \lambda_{k+1} \cdot \log \frac{1}{\delta} + \sqrt{\sum_{i>k} \lambda_i^2 \cdot \log \frac{1}{\delta}} \right).$$

By a union bound and net argument on $\mathcal{S}^{M-1}$, we have, with probability $\geq 1 - \delta$, for every unit vector $\mathbf{v} \in \mathbb{R}^M$,

$$\left| \sum_{i>k} \lambda_i \left( \mathbf{s}_i^\top \mathbf{v} \right)^2 - \frac{\sum_{i>k} \lambda_i}{M} \right| \lesssim \frac{1}{M} \cdot \left( \lambda_{k+1} \cdot \left( M + \log \frac{1}{\delta} \right) + \sqrt{\sum_{i>k} \lambda_i^2 \cdot \left( M + \log \frac{1}{\delta} \right)} \right).$$

So with probability at least $1 - e^{-\Omega(M)}$, we have

$$\left\| \mathbf{S}_{k:\infty} \mathbf{H}_{k:\infty} \mathbf{S}_{k:\infty}^\top - \frac{\sum_{i>k} \lambda_i}{M} \cdot \mathbf{I}_M \right\|_2 \lesssim \frac{1}{M} \cdot \left( \lambda_{k+1} \cdot M + \sqrt{\sum_{i>k} \lambda_i^2 \cdot M} \right)$$

$$\approx \lambda_{k+1} + \sqrt{\frac{\sum_{i>k} \lambda_i^2}{M}},$$

which completes the proof of the first part of Lemma G.2.

To prove the second part of Lemma G.2, it suffices to note that

$$\mathbf{S}_{k:\infty} \mathbf{H}_{k:\infty} \mathbf{S}_{k:\infty}^\top \succeq \sum_{i=k+1}^{2M+k} \lambda_i \mathbf{s}_i \mathbf{s}_i^\top \succeq \lambda_{2M+k} \cdot \sum_{i=k+1}^{2M+k} \mathbf{s}_i \mathbf{s}_i^\top \succeq c\lambda_{2M+k} \cdot \mathbf{I}_M$$

for some constant $c > 1$ with probability at least $1 - e^{-\Omega(M)}$, where the last line follows from concentration properties of Gaussian covariance matrices (see e.g., Thereom 6.1 [41]). $\qquad \square$

**Lemma G.3** (Head concentration). *With probability at least $1 - e^{-\Omega(M)}$, we have*

$$\text{for every } j \leq k, \quad |\mu_j(\mathbf{S}_{0:k} \mathbf{H}_{0:k} \mathbf{S}_{0:k}^\top) - \lambda_j| \lesssim \sqrt{\frac{k}{M}} \cdot \lambda_j.$$

*Proof of Lemma G.3.* Note that the spectrum of $\mathbf{S}_{0:k} \mathbf{H}_{0:k} \mathbf{S}_{0:k}^\top$ is indentical to the spectrum of $\mathbf{H}_{0:k}^{1/2} \mathbf{S}_{0:k}^\top \mathbf{S}_{0:k} \mathbf{H}_{0:k}^{1/2}$. We will bound the latter. We start with bounding the spectrum of $\mathbf{S}_{0:k} \mathbf{S}_{0:k}^\top$. To this end, we write $\mathbf{S}_{0:k}^\top \in \mathbb{R}^{k \times M}$ as

$$\mathbf{S}_{0:k}^\top = (\mathbf{s}_1 \quad \cdots \quad \mathbf{s}_M), \quad \mathbf{s}_i \sim \mathcal{N}\left( 0, \frac{1}{M} \cdot \mathbf{I}_k \right), \quad i = 1, \ldots, M.$$

Then repeating the argument in Lemma G.2, we have, with probability $\geq 1 - \delta$, for every unit vector $\mathbf{v} \in \mathbb{R}^k$,

$$\left| \mathbf{v}^\top \mathbf{S}_{0:k}^\top \mathbf{S}_{0:k} \mathbf{v} - 1 \right| = \left| \sum_{i=1}^M \left( \mathbf{s}_i^\top \mathbf{v} \right)^2 - 1 \right|$$

$$\lesssim \frac{1}{M} \cdot \left( 1 \cdot \left( k + \log \frac{1}{\delta} \right) + \sqrt{M \cdot \left( k + \log \frac{1}{\delta} \right)} \right)$$

$$\lesssim \sqrt{\frac{k + \log(1/\delta)}{M}}.$$

So we have, with probability $\geq 1 - e^{-\Omega(M)}$,

$$\left\| \mathbf{S}_{0:k}^\top \mathbf{S}_{0:k} - \mathbf{I}_k \right\|_2 \lesssim \sqrt{\frac{k}{M}}.$$

This implies that

$$\mu_j\big(\mathbf{H}_{0:k}^{1/2}\mathbf{S}_{0:k}^\top\mathbf{S}_{0:k}\mathbf{H}_{0:k}^{1/2}\big) \leq \mu_j\big(\mathbf{H}_{0:k}^{1/2}\mathbf{H}_{0:k}^{1/2}\big) + c_1 \cdot \sqrt{\frac{k}{M}} \cdot \mu_j\big(\mathbf{H}_{0:k}^{1/2}\mathbf{H}_{0:k}^{1/2}\big)$$

$$= \lambda_j + c_1 \cdot \sqrt{\frac{k}{M}} \cdot \lambda_j,$$

and that

$$\mu_j\big(\mathbf{H}_{0:k}^{1/2}\mathbf{S}_{0:k}^\top\mathbf{S}_{0:k}\mathbf{H}_{0:k}^{1/2}\big) \geq \mu_j\big(\mathbf{H}_{0:k}^{1/2}\mathbf{H}_{0:k}^{1/2}\big) - c_1 \cdot \sqrt{\frac{k}{M}} \cdot \mu_j\big(\mathbf{H}_{0:k}^{1/2}\mathbf{H}_{0:k}^{1/2}\big)$$

$$= \lambda_j - c_1 \cdot \sqrt{\frac{k}{M}} \cdot \lambda_j.$$

We have completed the proof. □

### G.2 Concentration results under power-law spectrum

**Lemma G.4** (Eigenvalues of $\mathbf{SHS}^\top$ under power-law spectrum). *Suppose Assumption 2 hold. There exist $a$-dependent constants $c_2 > c_1 > 0$ such that*

$$c_1 j^{-a} \leq \mu_j(\mathbf{SHS}^\top) \leq c_2 j^{-a}$$

*with probability at least $1 - e^{-\Omega(M)}$.*

*Proof of Lemma G.4.* Let $(\tilde{\lambda}_i)_{i=1}^M$ denote the eigenvalues of $\mathbf{SHS}^\top$ in an non-increasing order. Using Lemma G.1 with $k = M/c$ for some sufficiently large constant $c$ and noting that $\sum_{i>k} i^{-a} \approx k^{1-a}$, we have

$$\frac{1}{2} \cdot (j^{-a} + \tilde{c}_1 M^{-a}) - \tilde{c}_2 \cdot M^{-a} \leq \tilde{\lambda}_j \leq \frac{3}{2} \cdot (j^{-a} + \tilde{c}_1 M^{-a}) + \tilde{c}_2 \cdot M^{-a}$$

for every $j \in [M]$ for some constants $\tilde{c}_i, i \in [2]$ with probability at least $1 - e^{-\Omega(M)}$. Therefore, for all $j \leq M/\tilde{c}$ for some sufficiently large constant $\tilde{c} > 1$, we have

$$\tilde{\lambda}_j \in [\tilde{c}_3 j^{-a}, \tilde{c}_4 j^{-a}]$$

with probability at least $1 - e^{-\Omega(M)}$ for some constants $\tilde{c}_3, \tilde{c}_4 > 0$. For $j \in [M/\tilde{c}, M]$, by monotonicity of the eigenvalues, we have

$$\tilde{\lambda}_j \leq \tilde{\lambda}_{\lfloor M/\tilde{c}\rfloor} \leq \tilde{c}_4\Big(\Big\lfloor\frac{M}{\tilde{c}}\Big\rfloor\Big)^{-a} \leq \tilde{c}_5 M^{-a} \leq \tilde{c}_5 j^{-a}$$

for some sufficiently large constant $\tilde{c}_5 > \tilde{c}_4$ with probability at least $1 - e^{-\Omega(M)}$. Moreover, using Lemma G.2 with $k = 0$, we obtain

$$\tilde{\lambda}_j \geq \tilde{\lambda}_M \geq \mu_{\min}(\mathbf{S}_{k:\infty}\mathbf{H}_{k:\infty}\mathbf{S}_{k:\infty}^\top) \geq \tilde{c}_6\tilde{\lambda}_{2M} \geq \tilde{c}_7(M/\tilde{c})^{-a} \geq \tilde{c}_8 j^{-a}$$

with probability at least $1 - e^{-\Omega(M)}$ for some constants $\tilde{c}_6, \tilde{c}_7, \tilde{c}_8 > 0$. Combining the bounds for $j \leq M/\tilde{c}$ and $j \in [M/\tilde{c}, M]$ completes the proof. □

**Lemma G.5** (Ratio of eigenvalues of $\mathbf{S}_{k:\infty}\mathbf{H}_{k:\infty}\mathbf{S}_{k:\infty}^\top$ under power-law spectrum). *Suppose Assumption 2 hold. There exists some $a$-dependent constant $c > 0$ such that for any $k \geq 1$, the ratio between the $M/2$-th and $M$-th eigenvalues*

$$\frac{\mu_{M/2}(\mathbf{S}_{k:\infty}\mathbf{H}_{k:\infty}\mathbf{S}_{k:\infty}^\top)}{\mu_M(\mathbf{S}_{k:\infty}\mathbf{H}_{k:\infty}\mathbf{S}_{k:\infty}^\top)} \leq c$$

*with probability at least $1 - e^{-\Omega(M)}$.*

*Proof of Lemma G.5.* We prove the lemma under two scenarios where $k$ is relatively small (or large) compared with $M$.

Let $c > 0$ be some sufficiently large constant. Applying Lemma G.1 with $\mathbf{H}_{k:\infty}$ replacing $\mathbf{H}$, for $k_0 = M/c$, we have

$$\mu_{M/2}(\mathbf{S}_{k:\infty}\mathbf{H}_{k:\infty}\mathbf{S}_{k:\infty}^\top) \leq \frac{3}{2}\cdot\left(\lambda_{M/2+k} + \frac{\sum_{i>k_0}\lambda_{i+k}}{M}\right) + c_1\cdot\lambda_{k_0+1+k},$$

$$\lesssim \left(\frac{M}{2}+k\right)^{-a} + \frac{(k_0+k)^{1-a}}{M} + (k_0+1+k)^{-a}$$

$$\lesssim (k\vee M)^{-a} + (k\vee M)^{-a}\left(1\vee\frac{k}{M}\right) + (k\vee M)^{-a}$$

$$\lesssim (k\vee M)^{-a}\left(1\vee\frac{k}{M}\right) \tag{35}$$

with probability at least $1 - e^{-\Omega(M)}$ for some constant $c_1 > 0$.

**Case 1:** $k \lesssim M$    From Lemma G.2, we have

$$\mu_{\min}(\mathbf{S}_{k:\infty}\mathbf{H}_{k:\infty}\mathbf{S}_{k:\infty}^\top) \gtrsim \lambda_{k+2M} \gtrsim (k\vee M)^{-a}.$$

with probability at least $1 - e^{-\Omega(M)}$. Therefore

$$\frac{\mu_{M/2}(\mathbf{S}_{k:\infty}\mathbf{H}_{k:\infty}\mathbf{S}_{k:\infty}^\top)}{\mu_M(\mathbf{S}_{k:\infty}\mathbf{H}_{k:\infty}\mathbf{S}_{k:\infty}^\top)} \lesssim 1$$

with probability at least $1 - e^{-\Omega(M)}$ when $k/M \lesssim 1$.

**Case 2:** $k \gtrsim M$    On the other hand, when $k$ is relatively large, using Lemma G.1 with $\mathbf{H}_{k:\infty}$ replacing $\mathbf{H}$ again, we obtain

$$\mu_M(\mathbf{S}_{k:\infty}\mathbf{H}_{k:\infty}\mathbf{S}_{k:\infty}^\top) \geq \frac{1}{2}\cdot\left(\lambda_{M+k} + \frac{\sum_{i>k_0}\lambda_{i+k}}{M}\right) - c_1\cdot\lambda_{k_0+1+k},$$

$$\geq c_2\left[(M+k)^{-a} + \frac{(k_0+k)^{1-a}}{M}\right] - c_3\cdot(k_0+1+k)^{-a}$$

with probability at least $1 - e^{-\Omega(M)}$, where $c_1, c_2, c_3 > 0$ are some universal constants. Choosing $k_0 = M/c^2$ for some sufficiently large constant $c > 0$, we further obtain

$$\mu_M(\mathbf{S}_{k:\infty}\mathbf{H}_{k:\infty}\mathbf{S}_{k:\infty}^\top) \geq c_4(M+k)^{-a}\left[1+\frac{k}{M}\right] - c_5(M+k)^{-a}$$

$$\geq c_6(M\vee k)^{-a}\left[1\vee\frac{k}{M}\right] - c_7(M\vee k)^{-a} \tag{36}$$

with probability at least $1 - e^{-\Omega(M)}$, where $(c_i)_{i=4}^7$ are $a$-dependent constants. Since

$$c_6(M\vee k)^{-a}\left[1\vee\frac{k}{M}\right] - c_7(M\vee k)^{-a} \geq \frac{c_6}{2}(k\vee M)^{-a}\left(1\vee\frac{k}{M}\right)$$

when $k$ is large, i.e., $k/M > \tilde{c}$ for some sufficiently large $a$-dependent constant $\tilde{c} > 0$ that may depend on $(c_i)_{i=1}^7$, we have from Eq. (35) and (36) that

$$\frac{\mu_{M/2}(\mathbf{S}_{k:\infty}\mathbf{H}_{k:\infty}\mathbf{S}_{k:\infty}^\top)}{\mu_M(\mathbf{S}_{k:\infty}\mathbf{H}_{k:\infty}\mathbf{S}_{k:\infty}^\top)} \lesssim \frac{(k\vee M)^{-a}\left(1\vee\frac{k}{M}\right)}{(k\vee M)^{-a}\left(1\vee\frac{k}{M}\right)} \lesssim 1$$

with probability at least $1 - e^{-\Omega(M)}$. $\square$

### G.3 Concentration results under logarithmic power-law spectrum

**Lemma G.6** (Proof of Theorem 6 in [5]). *Suppose Assumption 4 hold. Then there exist some $a$-dependent constants $c, \tilde{c} > 0$ such that, with probability at least $1 - e^{-\Omega(M)}$*

$$\mu_j(\mathbf{S}\mathbf{H}\mathbf{S}^\top) \in \begin{cases} [c\cdot j^{-1}\log^{-a}(j+1), \tilde{c}\cdot j^{-1}\log^{-a}(j+1)] & j \leq \mathsf{k}^*, \\ [c\cdot M^{-1}\log^{1-a}(M), \tilde{c}\cdot M^{-1}\log^{1-a}(M)] & \mathsf{k}^* < j \leq M, \end{cases}$$

*where* $\mathsf{k}^* \asymp M/\log(M)$. *Also, there exists some $a$-dependent constants $c_1, c_2 > 0$ such that*

$$\frac{c_1}{j \log^{2a}(j+1)} \le \mu_j(\mathbf{SH}^2\mathbf{S}^\top) \le \frac{c_2}{j \log^{2a}(j+1)}$$

*with probability at least $1 - e^{-\Omega(M)}$.*

*Proof of Lemma G.6.* The proof is adapted from the proof of Theorem 6 in [5]. We include it here for completeness.

**First part of Lemma G.6.** In Lemma G.1, for some constant $c > 1$, choose

$$\mathsf{k}^* := \min\left\{ k \ge 0 : \sum_{i>k} \lambda_i \ge c \cdot M \cdot \lambda_{k+1} \right\}.$$

Then with probability $\ge 1 - e^{-\Omega(M)}$, we have:

$$\text{for every } 1 \le j \le M, \quad \frac{1}{c_1} \cdot \left( \lambda_j + \frac{\sum_{i>\mathsf{k}^*} \lambda_i}{M} \right) \le \tilde{\lambda}_j \le c_1 \cdot \left( \lambda_j + \frac{\sum_{i>\mathsf{k}^*} \lambda_i}{M} \right),$$

where $c_1 > 1$ is a constant.

When $\lambda_j \asymp j^{-1} \log^{-a}(j+1)$, we have

$$\mathsf{k}^* \asymp M/\log(M),$$

and

$$\sum_{i>\mathsf{k}^*} \lambda_i \asymp \log^{1-a}(\mathsf{k}^*) \asymp \log^{1-a}(M).$$

Therefore, we have

$$\tilde{\lambda}_j \asymp \lambda_j + \frac{\sum_{i>\mathsf{k}^*} \lambda_i}{M}$$

$$\asymp \begin{cases} j^{-1} \log^{-a}(j+1) & j \le \mathsf{k}^*, \\ M^{-1} \log^{1-a}(M) & \mathsf{k}^* < j \le M, \end{cases}$$

where $\mathsf{k}^* \asymp M/\log(M)$.

**Second part of Lemma G.6.** Let $\bar{\lambda}_i$ denote the $i$-th eigenvalue of $\mathbf{SH}^2\mathbf{S}^\top$ for $i \in [M]$. Using Lemma G.1 with $k = M/c$ for some sufficiently large constant $c_0$ and noting that $\sum_{i>k} \lambda_i^2 \asymp \sum_{i>k} i^{-2} \log^{-2a}(i+1) \lesssim k^{-1} \log^{-2a} k$, we have

$$\frac{1}{2} \cdot j^{-2} \log^{-2a}(j+1) - \tilde{c}_2 \cdot M^{-2} \log^{-2a} M$$

$$\le \bar{\lambda}_j \le \frac{3}{2} \cdot (j^{-2} \log^{-2a}(j+1) + \tilde{c}_1 M^{-2} \log^{-2a} M) + \tilde{c}_2 \cdot M^{-2} \log^{-2a} M$$

for every $j \in [M]$ for some constants $\tilde{c}_i, i \in [2]$ with probability at least $1 - e^{-\Omega(M)}$. Therefore, for all $j \le M/\tilde{c}$ for some sufficiently large constant $\tilde{c} > 1$, we have

$$\bar{\lambda}_j \in [\tilde{c}_3 \cdot j^{-2} \log^{-2a}(j+1), \tilde{c}_4 \cdot j^{-2} \log^{-2a}(j+1)]$$

with probability at least $1 - e^{-\Omega(M)}$ for some constants $\tilde{c}_3, \tilde{c}_4 > 0$. For $j \in [M/\tilde{c}, M]$, by monotonicity of the eigenvalues, we have

$$\bar{\lambda}_j \le \bar{\lambda}_{\lfloor M/\tilde{c} \rfloor} \le \tilde{c}_4 \left( \left\lfloor \frac{M}{\tilde{c}} \right\rfloor \right)^{-2} \log^{-2a} \left( \left\lfloor \frac{M}{\tilde{c}} \right\rfloor \right) \le \tilde{c}_5 M^{-2} \log^{-2a} M \le \tilde{c}_6 \cdot j^{-2} \log^{-2a}(j+1)$$

for some constants $c_5, c_6 > 0$ with probability at least $1 - e^{-\Omega(M)}$. Moreover, using Lemma G.2 with $k = 0$, we obtain

$$\bar{\lambda}_j \ge \bar{\lambda}_M \ge \mu_{\min}(\mathbf{S}_{k:\infty}\mathbf{H}^2_{k:\infty}\mathbf{S}^\top_{k:\infty}) \ge \tilde{c}_7 \bar{\lambda}_{2M} \ge \tilde{c}_8 \cdot j^{-2} \log^{-2a}(j+1)$$

with probability at least $1 - e^{-\Omega(M)}$ for some constants $\tilde{c}_7, \tilde{c}_8 > 0$ when $j \in [M/\tilde{c}, M]$. Combining the bounds for $j \le M/\tilde{c}$ and $j \in [M/\tilde{c}, M]$ completes the proof. $\square$

**Lemma G.7** (Ratio of eigenvalues of $\mathbf{S}_{k:\infty}\mathbf{H}_{k:\infty}\mathbf{S}_{k:\infty}^{\top}$ under logarithmic power-law spectrum).
*Suppose Assumption 4 hold. There exists some $a$-dependent constant $c > 0$ such that for any $k \geq 1$, the ratio between the $M/2$-th and $M$-th eigenvalues*

$$\frac{\mu_{M/2}(\mathbf{S}_{k:\infty}\mathbf{H}_{k:\infty}\mathbf{S}_{k:\infty}^{\top})}{\mu_{M}(\mathbf{S}_{k:\infty}\mathbf{H}_{k:\infty}\mathbf{S}_{k:\infty}^{\top})} \leq c$$

*with probability at least $1 - e^{-\Omega(M)}$.*

*Proof of Lemma G.7.* Similar to the proof of Lemma G.5, we prove the lemma under two scenarios where $k$ is relatively small (or large) compared with $M$.

Let $c > 0$ be some sufficiently large constant. Applying Lemma G.1 with $\mathbf{H}_{k:\infty}$ replacing $\mathbf{H}$, for $k_0 = M/c$, we have

$$\mu_{M/2}(\mathbf{S}_{k:\infty}\mathbf{H}_{k:\infty}\mathbf{S}_{k:\infty}^{\top}) \leq \frac{3}{2} \cdot \left(\lambda_{M/2+k} + \frac{\sum_{i>k_0}\lambda_{i+k}}{M}\right) + c_1 \cdot \lambda_{k_0+1+k},$$

$$\lesssim \left(\frac{M}{2} + k\right)^{-1}\log^{-a}\left(\frac{M}{2} + k\right) + \frac{\log^{1-a}(k_0 + k)}{M} + \frac{\log^{-a}(k_0 + 1 + k)}{k_0 + 1 + k}$$

$$\lesssim \frac{\log^{-a}(M + k)}{(M + k)} + \frac{\log^{1-a}(M + k)}{M} \lesssim \frac{\log^{1-a}(M + k)}{M} \qquad (37)$$

with probability at least $1 - e^{-\Omega(M)}$ for some constant $c_1 > 0$.

**Case 1: $k \lesssim M$.** Applying Lemma G.1 with $\mathbf{H}_{k:\infty}$ replacing $\mathbf{H}$, for $k_0 = M/c$, we have

$$\mu_{M}(\mathbf{S}_{k:\infty}\mathbf{H}_{k:\infty}\mathbf{S}_{k:\infty}^{\top}) \gtrsim \frac{1}{2} \cdot \left(\lambda_{M+k} + \frac{\sum_{i>k_0}\lambda_{i+k}}{M}\right) - c_1 \cdot \lambda_{k_0+1+k},$$

$$\gtrsim (M + k)^{-1}\log^{-a}(M + k) + \frac{\log^{1-a}(k_0 + k)}{M} - c\frac{\log^{-a}(k_0 + 1 + k)}{k_0 + 1 + k}$$

$$\gtrsim \frac{\log^{-a}(M + k)}{(M + k)} + \frac{\log^{1-a}(M + k)}{M} - \tilde{c}\frac{\log^{-a}(M)}{M}$$

$$\gtrsim \frac{\log^{1-a}M}{M}$$

with probability at least $1 - e^{-\Omega(M)}$. Therefore,

$$\frac{\mu_{M/2}(\mathbf{S}_{k:\infty}\mathbf{H}_{k:\infty}\mathbf{S}_{k:\infty}^{\top})}{\mu_{M}(\mathbf{S}_{k:\infty}\mathbf{H}_{k:\infty}\mathbf{S}_{k:\infty}^{\top})} \lesssim \left[\frac{\log^{1-a}(M + k)}{M}\right] / \left[\frac{\log^{1-a}M}{M}\right] \lesssim 1$$

with probability at least $1 - e^{-\Omega(M)}$ when $k/M \lesssim 1$.

**Case 2: $k \gtrsim M$.** On the other hand, when $k$ is relatively large, using Lemma G.1 with $\mathbf{H}_{k:\infty}$ replacing $\mathbf{H}$ and $k_0 = M/c$ again, we obtain

$$\mu_{M}(\mathbf{S}_{k:\infty}\mathbf{H}_{k:\infty}\mathbf{S}_{k:\infty}^{\top}) \geq \frac{1}{2} \cdot \left(\lambda_{M+k} + \frac{\sum_{i>k_0}\lambda_{i+k}}{M}\right) - c_1 \cdot \lambda_{k_0+1+k},$$

$$\gtrsim (M + k)^{-1}\log^{-a}(M + k) + \frac{\log^{1-a}(k_0 + k)}{M} - c_2\frac{\log^{-a}(k_0 + 1 + k)}{k_0 + 1 + k}$$

$$\gtrsim k^{-1}\log^{-a}(k) + \frac{\log^{1-a}(M + k)}{M} - c_3\frac{\log^{-a}(M + k)}{M + k}$$

$$\gtrsim \frac{\log^{1-a}(M + k)}{M}$$

with probability at least $1 - e^{-\Omega(M)}$, where $c_1, c_2, c_3 > 0$ are some $a$-dependent constants. Therefore,

$$\frac{\mu_{M/2}(\mathbf{S}_{k:\infty}\mathbf{H}_{k:\infty}\mathbf{S}_{k:\infty}^{\top})}{\mu_{M}(\mathbf{S}_{k:\infty}\mathbf{H}_{k:\infty}\mathbf{S}_{k:\infty}^{\top})} \lesssim \left[\frac{\log^{1-a}(M + k)}{M}\right] / \left[\frac{\log^{1-a}(M + k)}{M}\right] \lesssim 1$$

with probability at least $1 - e^{-\Omega(M)}$ when $k/M \gtrsim 1$. $\qquad \square$

