# OpenReview forum: "Scaling Laws in Linear Regression: Compute, Parameters, and Data"
_NeurIPS.cc/2024/Conference — NeurIPS 2024 poster_

### Official Review · Reviewer_8LLD · 2024-07-08

**Soundness:** 4
**Presentation:** 4
**Contribution:** 3
**Rating:** 8
**Confidence:** 2

**Summary:**

Large models often empirically satisfy the neural scaling law i.e. the error decreases when the number of data and the model size increase, however, this contradicts with the widely accepted belief in learning theory that the variance error (one of the decomposed errors) should increase with the model size. Authors investigate this in the infinite-dimensional linear regression setting and show that the variance error is dominated by other errors because of the implicit regularization of SGD. Authors also support their claim with numerical simulations.

**Strengths:**

- The authors reconcile empirically observed scaling law and the classical statistical learning theory, and offers a novel viewpoint to understand and explain their original "conflictions", which is novel and should be interesting for both learning theory and LLM community.
- The authors offer a general framework in section 6 to incorporate the general spectrum case of data covariance matrix, and it gives a novel technical innovations to sharply control the effect of data sketch.
- The paper is very well written and it is easy to understand

**Weaknesses:**

- The assumptions of the paper might be too strong with linear regression setting and Gaussian design assumption, is it possible to extend to the kernel setting and with relaxed assumption on features [1]?
- While the current analysis is focusing on one-pass SGD, which is not often used in LLM training, can the result be extended to Adam?
- The result is not very surprising

[1] Barzilai et al.,  Generalization in Kernel Regression Under Realistic Assumptions

**Questions:**

- I am confused about the sketch matrix setting? Is the fixed sketch matrix analogous to the neural net with size M and why the teacher-student model in previous work can be viewed as sketched linear regression model
- I am also confused about the Hypercontractivity assumption in Assumption 5, what's its intuition and why it is required in the proof?
- What's the purpose for setting geometric decaying stepsize, what happens if it is fixed or it is polynomial decaying?
- Is the regularization effect of SGD determined by the choice of stepsize, where the optimal choice controls the strength of the implicit bias of SGD?

**Limitations:**

See weakness

---

> ### Author Rebuttal · Authors · 2024-08-06
>
> Thank you for supporting our paper! We address your comments as follows.
>
> > Q1. The assumptions of the paper might be too strong with linear regression setting and Gaussian design assumption, is it possible to extend to the kernel setting and with relaxed assumption on features [1]?
>
> A1. The Gaussian design assumption enables us to derive matching lower bounds for linear regression under data sketch. Note that our upper bound does not require Gaussian design. We will clarify this in the revision. Extending our results to kernel settings following relaxed assumptions in [1] is an interesting future direction. However, we think the obtained guarantee might be weaker. For instance, Theorem 2 in [1] only provides constant probability guarantees while our results, albeit using stronger assumptions, hold with high probability.
>
> ---
>
> > Q2. While the current analysis is focusing on one-pass SGD, which is not often used in LLM training, can the result be extended to Adam?
>
> A2. Our SGD analysis cannot be directly extended to Adam because the structure of the randomness becomes convoluted in Adam. Nonetheless we think this is an important question and we will comment on this in the revision.
>
> ---
>
> > Q3. The result is not very surprising.
>
> A3. We respectfully disagree. Typical risk bounds only involve upper bounds, but an upper bound is insufficient to characterize scaling laws, since an upper bound can be loose and the implied scaling law might be just an artifact. Our results, by establishing matching upper and lower excess risk bounds, rigorously characterizes scaling laws. Results of such kind are quite rare to the best of our knowledge. Thus we believe our results are significant.
>
> ---
>
> > Q4. I am confused about the sketch matrix setting? Is the fixed sketch matrix analogous to the neural net with size M and why the teacher-student model in previous work can be viewed as sketched linear regression model
>
> A4. You are correct that our sketched data model is an analog to a two-layer network of size $M$ with fixed inner layer and linear activation. Although prior works formulated their problem as a teacher-student model, they (taking [Bordelon et al., 2024] as an example) assumed that the teacher model is an infinite dimensional linear model and the student model is another linear model using features projected by a random matrix $A$ (see their equations 1 and 3). Therefore their teacher-student model is effectively a sketched data model.
>
> ---
>
> > Q5. I am also confused about the Hypercontractivity assumption in Assumption 5, what's its intuition and why it is required in the proof?
>
> A5. The hypercontractivity condition requires the fourth moment tensor to be controlled by the second moment tensor. This is known to be weaker than subGaussainity while still allowing a sharp SGD analysis [Zou et al., 2023]. The intuition is that the expected risk of the SGD output involves at most the fourth moment information of the data, therefore some fourth moment condition is necessary to derive an expected risk bound.
>
> ---
>
> > Q6. What's the purpose for setting geometric decaying stepsize, what happens if it is fixed or it is polynomial decaying?
>
> A6. The geometrically decaying stepsize scheduler is known for the last iterate of SGD to achieve a nearly (up to logarithmic factors) minimax optimal excess risk [Ge et al., 2019] in linear regression. Therefore we focus on the geometrically decaying stepsize scheduler. The last iterate of SGD would suffer from a constant variance error if used with a constant stepsize. Moreover, a polynomially decaying stepsize scheduler leads to a worse risk bound compared to a geometrically decaying stepsize scheduler as shown in [Wu et al., 2022a].
>
> ---
>
> > Q7. Is the regularization effect of SGD determined by the choice of stepsize, where the optimal choice controls the strength of the implicit bias of SGD?
>
> A7. You are correct that the regularization of SGD is determined by the choice of stepsize. The optimal stepsize can be solved from the instance-wise sharp risk bounds. However, this requires certain knowledge of the task parameter and the spectrum of the data covariance. Additionally, $1/(\gamma n)$ in the SGD risk bound is comparable to $\lambda$ in the known ridge regression risk bound. This is explained in depth in [Zou et al., 2021].
>
> **References**
>
> - Blake Bordelon, Alexander Atanasov, and Cengiz Pehlevan. A dynamical model of neural scaling laws. arXiv preprint arXiv:2402.01092, 2024.
>
> - Rong Ge, Sham M Kakade, Rahul Kidambi, and Praneeth Netrapalli. The step decay schedule: A near optimal, geometrically decaying learning rate procedure for least squares. Advances in neural information processing systems, 32, 2019.
>
> - Jingfeng Wu, Difan Zou, Vladimir Braverman, Quanquan Gu, and Sham M. Kakade. Last iterate risk bounds of sgd with decaying stepsize for overparameterized linear regression. The 39th International Conference on Machine Learning, 2022a.
>
> - Difan Zou, Jingfeng Wu, Vladimir Braverman, Quanquan Gu, Dean P Foster, and Sham Kakade. The benefits of implicit regularization from sgd in least squares problems. Advances in Neural Information Processing Systems, 34:5456–5468, 2021.
>
> - Difan Zou, Jingfeng Wu, Vladimir Braverman, Quanquan Gu, and Sham M Kakade. Benign overfitting of constant-stepsize sgd for linear regression. Journal of Machine Learning Research, 24(326):1–58, 2023.

---

> > ### Comment · Reviewer_8LLD · 2024-08-09
> >
> > Thanks for your response and I will keep my positive score

---

### Official Review · Reviewer_9pNW · 2024-07-09

**Soundness:** 3
**Presentation:** 3
**Contribution:** 3
**Rating:** 5
**Confidence:** 3

**Summary:**

This work examines neural scaling laws, in the simplified setting of linear regression trained by one-pass SGD. In particular, it attempts to explain the apparent mismatch between the statistical theory on one hand, which predicts that variance error increases with model size, and neural scaling laws on the other, which state that predictive performance should increase with model size. By considering sketched covariates of an infinite dimensional linear regression model with Gaussian prior under mild spectral assumptions, scaling laws are derived while keeping track of a variance term that increases with model complexity and corrects for a mismatch between the best-in-class predictor and the best possible predictor. However, this term is of higher order than the usual approximation and bias terms appearing in scaling laws, accounting for the empirical accuracy of scaling laws in practice. Numerical examples are provided for synthetic data.

**Strengths:**

This is a good paper that addresses an important theoretical inconsistency, and provides progress towards more general results.
The paper is generally well written and communicated.

**Weaknesses:**

It is not clear when the variance term is first given on page 2 that it is of higher order than the other terms. The reader has to find the more precise presentation in Theorem 4.1 in order to verify this for themselves. Clarification of this when the term is first introduced would be helpful and improve the paper.
While the paper's contribution is primarily theoretical, the only examples given are synthetic. Validation of the theory on a large benchmark dataset, even if modified to make it appropriate to the assumptions, would strengthen the paper significantly.

**Questions:**

Can you please address the weaknesses outlined above?
How do you think this work can this be generalized to other settings of interest?
Under which settings should the variance term be accounted for?

**Limitations:**

Some discussion of the assumptions of their work and their limitations would be helpful. Broader societal impact is mostly not applicable.

---

> ### Author Rebuttal · Authors · 2024-08-06
>
> We thank the reviewer for their helpful feedback and comments.
>
> > Q1. It is not clear when the variance term is first given on page 2 that it is of higher order than the other terms. The reader has to find the more precise presentation in Theorem 4.1 in order to verify this for themselves. Clarification of this when the term is first introduced would be helpful and improve the paper.
>
> A1. We note that the display on page 2 already suggests variance error is of higher order. Notice that $\gamma = O(1)$, so we have
>
>
> $$ \mathrm{min} \\{ M, (N \gamma)^{1/a} \\} / N
> \le\mathrm{min} \\{ M, (N \gamma)^{1/a} \\} / (N\gamma)
> \le \\begin{cases}
> \frac{M}{N \gamma} \le M^{1-a} & M <  (N \gamma)^{1/a}, \\\\
> (N \gamma)^{1/a-1}  & M \ge  (N \gamma)^{1/a}.
> \\end{cases}
> $$
>
>
> This implies that the variance error is of higher order compared to the sum of the other terms. We will make this more clear in the revision.
>
> ---
>
> > Q2. While the paper's contribution is primarily theoretical, the only examples given are synthetic. Validation of the theory on a large benchmark dataset, even if modified to make it appropriate to the assumptions, would strengthen the paper significantly.
>
> A2. The power-law relations predicted by our theory are aligned with empirical findings in many other works [e.g., Hoffmann et al., 2022]. However, the hyper-parameter $a,b$ in our work are oracle properties of the data distribution, and it is a challenging problem to estimate them in practice.  Nevertheless, we believe this is an interesting problem to  study.
>
> ---
>
> > Q3. How do you think this work can this be generalized to other settings of interest? Under which settings should the variance term be accounted for?
>
> A3. We believe the intuition carried from our theory, where the scaling laws are due to the disappearance of the variance error due to the implicit regularization, can be generalized to other settings. For instance, the practical LLM training often makes only one pass of the data, so less noise is fitted which implicitly regularizes the variance error. We believe this is an important reason for the observed scaling law.
>
> ---
>
> **References**
>
> - Jordan Hoffmann, Sebastian Borgeaud, Arthur Mensch, Elena Buchatskaya, Trevor Cai, Eliza Rutherford, Diego de Las Casas, Lisa Anne Hendricks, Johannes Welbl, Aidan Clark, et al. Training compute-optimal large language models. arXiv preprint arXiv:2203.15556, 2022.

---

> > ### Comment · Reviewer_9pNW · 2024-08-11
> > **Response**
> >
> > Thanks for your response and clarification. I will maintain my score.

---

### Official Review · Reviewer_iyfq · 2024-07-10

**Soundness:** 4
**Presentation:** 4
**Contribution:** 1
**Rating:** 6
**Confidence:** 5

**Summary:**

This paper sheds new light on the bounds of linear regression by framing them in terms of scaling laws.

**Strengths:**

- The set-up is very clear, the story is well-narrated and the model, including some sketch matrix to play the role of the model-size, is well-thought.

- The authors try to be exhaustive with the assumptions: by including in a second time, (i) the assumptions that make everything work (Assumptions A and B) and that allow to go beyond Gaussian data ; (ii) source condition, (iii) some other spectrum decay.

**Weaknesses:**

It is difficult to talk about a weakness: the paper is self-contained, the results are solid and the context very well explained.

Perhaps the authors should put emphasis on the fact that there is **absolutely no technical novelty and that the contribution relies in re-framing known bounds under the perspective of scaling laws**.

**Questions:**

No question

---

> ### Author Rebuttal · Authors · 2024-08-06
>
> Thanks for the feedback. Below is our response to your question.
>
> > Q1. Perhaps the authors should put emphasis on the fact that there is absolutely no technical novelty and that the contribution relies in re-framing known bounds under the perspective of scaling laws.
>
> A1. We respectfully disagree with the claim that our results are re-framing of known bounds. As discussed in lines 129-133, compared to the existing SGD analysis, we consider an additional data sketch, which leads to a new approximation error. Moreover, a direct application of existing SGD analysis only leads to sketch-dependent bias and variance errors, whereas we establish matching upper and lower bounds on these errors that also control the effects of data sketch. To achieve these, we develop several new concentration results (such as Lemma 6.2) and make novel uses of Von-Neuman’s inequality (see for example the proof of Lemma C.2). Therefore, we believe we have made novel technical contributions.

---

> > ### Comment · Reviewer_iyfq · 2024-08-10
> > **After Rebuttal**
> >
> > Let me be clearer, obviously, nothing is completely black and white, and there is obviously a continuum between a simple reformulation and new contributions. What I mean is that I totally share the feeling of **Reviewer v1rx** that we do not understand what are the *real* novelties of the approach --beyond the story telling.
> >
> > Hence, if Lemma 6.2 or Lemma C.2 are really new achievements, I feel that they could be promoted. I sincerely feel it is not the case, but it is only a question of perception and I am happy to be wrong.
> >
> > Furthermore, convergence rates on SGD and concentration rates on sketching or random features has been widely studied, under the present form, it seems hard to really understand **where** the authors are.
> >
> > Yet, I maintain my score as I have nothing against nice re-formulation.

---

> > > ### Author Response · Authors · 2024-08-13
> > >
> > > Thanks for the comment. Below we provide a roadmap of our proof to address some of your concerns.
> > >
> > > We agree that convergence rates on SGD and concentration rates on sketching or random features have been widely studied in many contexts. However, obtaining the main results (Theorem 4.1-4.3) in our work requires both application of the existing results and non-trivial developments based on the existing bounds.
> > >
> > > More specifically, our Theorem 6.1 is mainly built upon existing results in [Wu et al., 2022a,b]. However, in order to obtain Theorem 4.1-4.3 from Theorem 6.1, we first develop novel upper/lower bounds (Theorem 6.3, 6.4) of the approximation, bias and variance terms. These bounds decompose $\mathsf{Approx}$, $\mathsf{ Bias}$ and $\mathsf{Var}$ into the head  and tail components (i.e. $\\|w^*_{1:k}\\|\_2\^2,\\|w^*_{k:\infty}\\|\^2\_{H_{k:\infty}}$)  and involve the ratio of eigenvalues of the sketched covariance matrix. The bounds are carefully built so that the upper and lower bounds match up to constant factors in our examples. Theorem 4.1-4.3 are then obtained based on instantiation of Theorem 6.3 and 6.4 under the power-law (or log-power-law ) spectrum.
> > >
> > > The role of Lemma 6.2 and C.2: they are intermediate results developed in our proof. Lemma C.2 proves the first part of Theorem 6.4; Lemma 6.2 characterizes the eigenvalues of the sketched covariance matrix and enables us to obtain Theorem 4.1 and 4.2 from Theorem 6.3 and 6.4.  While we appreciate the discussion and understand the perspective shared, we respectfully believe that some technical contributions have been made (e.g., novel uses of Von-Neuman’s inequality) in proving these lemmas and other results.

---

### Official Review · Reviewer_v1rx · 2024-07-15

**Soundness:** 3
**Presentation:** 3
**Contribution:** 2
**Rating:** 5
**Confidence:** 3

**Summary:**

Motivated by the recent neural scaling law literature, this work investigates the generalization error rate of a sketched linear regression model trained on Gaussian covariates with power-law spectrum under one-pass SGD.

The main result are lower and upper bounds with matching rates, providing a detailed characterization of the different contributions to the generalization error.

**Strengths:**

The paper is well-written and easy to parse. The topic is of interest to the NeurIPS community, and the contribution is timely.

**Weaknesses:**

In my reading, the two main weaknesses are:
1. The lack of a wider context. While it makes precise connections to other recent works on the "*"theory of neural scaling laws"*" such as [Bahri et al., 2021, Maloney et al., 2022, Bordelon et al., 2024], it ignores that the study of generalization error rates in least-squares / ridge regression under source and capacity conditions is a classical topic with an extensive literature. For instance:

     - [1,2,3] have studied both single and multiple pass SGD for least squares regression on Hilbert spaces under source and capacity conditions. This corresponds to the kernel limit $M\gg N$ in this work. Borrowing the notation from [3] for concreteness, after a quick comparison with the identification of the capacity $\alpha = a$ and source $r=\frac{b-1}{2a}$, I believe the "hard" and "easy" rates from Theorems 4.1 and 4.2 can be retrieved from the one-pass rates in [3].
     - [4] extends the discussion of [1-3] to the random features approximation of kernels.
     - The rates from one-pass SGD on least squares can be identified with the rates of ridge regression with an appropriately chosen regularization, which has been exactly characterized in [7]. For example, the blue and orange regions in Figure 1 from [7] correspond to rates in Theorem 4.2 with $\ell = 1$.
     - A recent extension of the above to RFs case appeared in [6] (though note some of these rates were known from [7]).

     From the discussion above, it is not really clear what rates are new in this submission. I am willing to change my evaluation if this can be clarified with an in-depth comparison with these works.

2. The organization of the manuscript in Theorems with "settings of increasing generality" can come through as deceptive. Aren't Theorems 4.3, 4.2 and 4.1 corollaries of Theorems 6.3 and 6.4?


**References**

- [1] Yao, Y., Rosasco, L. & Caponnetto, A. On Early Stopping in Gradient Descent Learning. Constr Approx 26, 289–315 (2007). https://doi.org/10.1007/s00365-006-0663-2
- [2] Ying, Y., & Pontil, M. (2008). Online gradient descent learning algorithms. Foundations of Computational Mathematics, 8, 561-596.
- [3] Pillaud-Vivien, L., Rudi, A., & Bach, F. (2018). Statistical optimality of stochastic gradient descent on hard learning problems
through multiple passes. Advances in Neural Information Processing Systems, 31.
- [4] Luigi Carratino, Alessandro Rudi, Lorenzo Rosasco. Learning with SGD and Random Features.  Part of Advances in Neural Information Processing Systems 31 (NeurIPS 2018).
- [5] Hugo Cui, Bruno Loureiro, Florent Krzakala, Lenka Zdeborová. Generalization Error Rates in Kernel Regression: The Crossover from the Noiseless to Noisy Regime. Part of Advances in Neural Information Processing Systems 34 (NeurIPS 2021).
- [6] Leonardo Defilippis, Bruno Loureiro, Theodor Misiakiewicz. Dimension-free deterministic equivalents for random feature regression. arXiv:2405.15699.
- [7] Alessandro Rudi and Lorenzo Rosasco. Generalization properties of learning with random features. In Advances in Neural Information Processing Systems, pages 3215–3225, 2017.

**Questions:**

- L43-48:

> "*This difference must be reconciled, otherwise, the statistical learning theory and the empirical scaling law make conflic predictions: as the model size increases, the theoretical bound (2) predicts an increase of variance error that eventually causes an increase of the population risk, but the neural scaling law (1) predicts a decrease of the population risk. In other words, it remains unclear when to follow the prediction of the empirical scaling law (1) and when to follow that of the statistical learning bound (2).*"

Given our current understanding of random features regression, this claim is inflated. Several of the works mentioned above [4,6,7], as well as many other works in the random features literature have made the observation that increasing the "width" $M$ does not necessarily lead to an increase of variance / overfitting, thanks to the implicit regularization.


- L273-275:

> "*It is noteworthy that our simulations demonstrate stronger observations than the theoretical results in Theorem 4.1, which only establishes matching upper and lower bounds up to a constant factor.*"

What do you exactly mean in this sentence? What in Figure 2 is stronger than the exponents reported in Theorem 4.1?

**Limitations:**

The assumptions under which the Theorems hold are clearly stated, but I miss an explicit discussion of the limitations.

---

> ### Author Rebuttal · Authors · 2024-08-06
>
> Thank you for your feedback. We will cite and discuss the relationship between the pointed papers and our work in the revision. However, there are several potential misunderstandings about our results, which we would like to clarify below.
>
> > Q1. [1,2,3] have studied both single and multiple pass SGD for least squares regression on Hilbert spaces under source and capacity conditions. This corresponds to the kernel limit $M\gg N$ in this work.
>
> A1. There are three major differences that clearly separate our work for studying scaling laws from the prior works. First, as you mentioned, [1,2,3] focused on the kernel limit case where $M \gg N$, while our work treats both $M$ and $N$ as free variables — this is a key feature to allow studying scaling laws, which are functions of both $M$ and $N$. Second, we establish matching upper and lower bounds for SGD while [1,2,3] only focused on the upper bound. Without a matching lower bound, it is unclear whether the derived scaling law is the true behavior, or just a mathematical artifact of a loose upper bound. Finally, our work also establishes scaling laws under logarithmic power law spectrums (see Section 4.2), which is beyond the scope of capacity condition.
>
> ---
>
> > Q2. Borrowing the notation from [3] for concreteness, after a quick comparison with the identification of the capacity $\alpha=a$ and source $r=\frac{b-1}{2a}$, I believe the "hard" and "easy" rates from Theorems 4.1 and 4.2 can be retrieved from the one-pass rates in [3].
>
> A2. Our results cannot be recovered from [3]. As we discussed above, our Theorems 4.1 and 4.2 prove instance-wise matching upper and lower bounds as functions of both $M$ and $N$, while Theorem 1 in [3] only covers upper bounds as function of $N$.
>
> ---
>
> > Q3. [4] extends the discussion of [1-3] to the random features approximation of kernels.
>
> A3. We hope our A1 and A2 have clarified the relationship between our work and [1,2,3], hence also clarifies this question.
>
> ---
>
> > Q4. The rates from one-pass SGD on least squares can be identified with the rates of ridge regression with an appropriately chosen regularization, which has been exactly characterized in [7]. For example, the blue and orange regions in Figure 1 from [7] correspond to rates in Theorem 4.2 with $\ell=1$. A recent extension of the above to RFs case appeared in [6] (though note some of these rates were known from [7]).
>
>
> A4. We hope our A1 and A2 have clarified the relationship between our work and [1,2,3], hence also clarifies this question. Moreover, we would like to point out SGD is a more practical algorithm than ridge regression.
>
> ---
>
> > Q5. The organization of the manuscript in Theorems with "settings of increasing generality" can come through as deceptive. Aren't Theorems 4.3, 4.2 and 4.1 corollaries of Theorems 6.3 and 6.4?
>
> A5.  Theorems 4.3, 4.2 and 4.1 are derived based on Theorems 6.3 and 6.4, however,  we believe the derivations are non-trivial and Theorems 6.3 and 6.4 can be viewed as the building blocks of the proof of our main results. Therefore, we put Theorem 6.3 and 6.4 after Theorems 4.3, 4.2 and 4.1.
>
> ---
>
> > Q6. L43-48… Given our current understanding of random features regression, this claim is inflated. Several of the works mentioned above [4,6,7], as well as many other works in the random features literature have made the observation that increasing the "width"  does not necessarily lead to an increase of variance / overfitting, thanks to the implicit regularization.
>
> A6. This is a nice comment. We will rewrite this part to emphasize the differences between our results and prior results as discussed before.
>
> ---
>
> > Q7. L273-275… What do you exactly mean in this sentence? What in Figure 2 is stronger than the exponents reported in Theorem 4.1?
>
> A7. While our theorems provide upper and lower bounds of matching rates, the exact constant factors do not match. Our experiments suggest that the constant should also match, as indicated by the linear trend and small standard deviation in e.g. Figure 2,  which is stronger than our theoretical prediction.
>
> ---
>
> > Q8. The assumptions under which the Theorems hold are clearly stated, but I miss an explicit discussion of the limitations.
>
> A8. Thanks for the suggestion. Most of our assumptions are standard in literature, however, we will discuss the limitations of our assumptions (e.g., gaussian design, source condition) in the revision.

---

> > ### Comment · Reviewer_v1rx · 2024-08-09
> > **Clarification**
> >
> > I thank the authors for their rebuttal that addressed some of my questions, in particular Q5-Q8. However, I find that your reply to Q1-Q4 miss my point.
> >
> > Let me try to be clearer.
> >
> > - I don’t mean to say your results lack technical novelty. I understand [1,2,3] provide only upper bounds for kernel methods ($M\gg N$), while you also provide lower bounds with matching rates.
> >
> > - I also understand that [5,6,7] are for ridge, not SGD. My point is also not about which one is a more practical algorithm.
> >
> >  In Q1-Q4 I (tried) to convey two points:
> >
> > 1. First, how do your exact rates compare with the upper bounds of [1,2,3]? Do they match in the regime where they compare? Are they tighter?
> > 2. Second, how do your rates compare with the ridge rates from [5,6,7] for kernels and RF? We know that running SGD to convergence with decreasing learning rate recover the least-squares rates, which are not the optimally regularized ridge rates. I want understand  how close to optimal is early stopping in terms of implicit regularisation. See for example Section 4 of [6] for a summary of the rates in [5,7].
> >
> > It is not just about citing papers, but how your work fit within the classical source and capacity literature, and the conclusions this allow us to make about optimality of early stopping with SGD in the different scaling regimes.

---

> > > ### Author Response · Authors · 2024-08-10
> > > **Reply to Reviewer v1rx**
> > >
> > > Thanks Reviewer v1rx for the clarification. We will show below our rate matches the prior SGD and ridge rates in the comparable regimes. For simplicity, we will ignore logarithmic factors by replacing $N_{\mathrm{eff}}$ with $N$ in our Theorem 4.2 in the following discussions. Note that the logarithmic factor can be removed by considering averaged SGD instead of last iterate SGD.
> > >
> > > - Comparion with the prior SGD rate. We use [3] as an example. Aligning the source and capacity conditions, we have $\alpha = a$ and $r = (b-1)/(2a)$. Then the rate in their page 4 becomes $O(1/(N\gamma)^{(b-1)/a}  + (N\gamma)^{1/a}/N)$, which exactly matches ours in Theorem 4.2 when $M$ is large in the comparable regime.
> > >
> > > - Comparing with the prior ridge rate. We use [7] as an example. Aligning the source and capacity conditons, we have $\gamma=1/a$ and $r = (b-1)/(2a)$. Then their optimal rate in Theorem 2 becomes $O(N^{-(b-1)/b})$, which matches our Theorem 4.2 when $M$ is large and $\gamma$ is tuned to be $\gamma^* = \Theta(N^{a/b-1})$ in the comparable regime.
> > >
> > > Finally, we would like to point out that [3] requires $\mathrm{tr}(\Sigma^{1/\alpha})$ to be finite. However, we do not require such condition, that is, our bounds allow $\mathrm{tr}(H^{1/a}) = \infty$ in our notation.
> > > We will add these discussions in the revision.

---

> > > > ### Comment · Reviewer_v1rx · 2024-08-11
> > > >
> > > > Thank you for the comparison, this is interesting and in line with what I expected.
> > > >
> > > > I am increasing my core, and won't object acceptance conditioned that the authors include this discussion and put their work in perspective with the literature above.

---

### Official Review · Reviewer_zAqY · 2024-07-17

**Soundness:** 4
**Presentation:** 4
**Contribution:** 4
**Rating:** 8
**Confidence:** 4

**Summary:**

This paper proves that the scaling laws that occur in deep learning also occur when solving linear regression with SGD.

Namely, suppose:
* that covariates x are Gaussian with  a power-law spectrum
* the true labels are given by <w^*, x> for unknown w^*
* we run one-pass SGD to do linear regression on an M-dimensional sketch of the covariates x
* we have N data points

Then the risk is upper-bounded by an expression that depends only on the "bias" and "approximation" terms, with inverse polynomial dependence as observed in neural scaling laws. The "variance" term is of lower order (provided the appropriate large step sizes are used).

Extensions to when the data follows a logarithmic power spectrum are also explored.

**Strengths:**

The result is timely and I think will be of high interest to the community, since it sheds insight on why neural scaling laws might occur in practice.

The paper is well-presented, and the proofs are neatly written and easy to follow.

As opposed to previous works, this paper provides risk bounds for any network width M, and any number of data points N, instead of taking one of the parameters to infinity and deriving a scaling law in the other parameter.

**Weaknesses:**

Typo: "optimally tunned"

**Questions:**

1. How dependent are the results on a geometric step-size schedule? What if a constant schedule, or schedule decreasing as gamma_t = t^{-c} is used instead?

2. In line 232, it is stated that when $1 \leq b \leq a$ the tasks are "relatively hard", and in line 236 that when $a < b < a+1$ the tasks are "relatively easy". Is what is meant here that these tasks are "relatively harder/easier" than when a = b?

**Limitations:**

Yes

---

> ### Author Rebuttal · Authors · 2024-08-06
>
> Thank you for supporting our work! We will fix the typo. We address your concerns as follows.
>
> > Q1. How dependent are the results on a geometric step-size schedule? What if a constant schedule, or schedule decreasing as $\gamma_t = t^{-c}$ is used instead?
>
> A1. The geometrically decaying stepsize scheduler is known for the last iterate of SGD to achieve a nearly (up to logarithmic factors) minimax optimal excess risk [Ge et al., 2019] in linear regression. Therefore we focus on the geometrically decaying stepsize scheduler. The last iterate of SGD would suffer from a constant variance error if used with a constant stepsize. Moreover, a polynomially decaying stepsize scheduler leads to a worse risk bound compared to a geometrically decaying stepsize scheduler as shown in [Wu et al., 2022a]. We will clarify this in the revision.
>
> ---
>
> > Q2. In line 232, it is stated that when $1 \leq b \leq a$ the tasks are "relatively hard", and in line 236 that when $a < b < a+1$ the tasks are "relatively easy". Is what is meant here that these tasks are "relatively harder/easier" than when a = b?
>
> A2.  You are correct. We will clarify this in the revision.
>
> ---
> **References:**
>
> - Rong Ge, Sham M Kakade, Rahul Kidambi, and Praneeth Netrapalli. The step decay schedule: A near optimal, geometrically decaying learning rate procedure for least squares. Advances in neural information processing systems, 32, 2019.
>
> - Jingfeng Wu, Difan Zou, Vladimir Braverman, Quanquan Gu, and Sham M. Kakade. Last iterate risk bounds of sgd
> with decaying stepsize for overparameterized linear regression. The 39th International Conference on Machine
> Learning, 2022.

---

> > ### Comment · Reviewer_zAqY · 2024-08-13
> > **Response**
> >
> > Thank you, I will keep my score.

---

### Decision · Program_Chairs · 2024-09-25

**Decision:**

Accept (poster)

**Comment:**

**Summary** The paper contains an analysis of a random features model trained with SGD.  It assumes an underlying infinite-dimensional linear regression problem.  The decay of the variances of the infinite dimensional problem and the target are both powerlaw type, denoted by parameters $(a,b)$ respectively.

The data is is projected into finite dimensions by means of a finite-dimensional (isotropic Gaussian) sketching matrix having $M$ dimensions, and then this sketched data and a noisy target (with noise level $\sigma^2$) is released to the optimizer.

Sharp upper and lower bounds for the risk (less its irreducible part) are given over a range of parameters $(a,b,\sigma^2)$, as the number of samples $N$ (hence number of steps of one-pass SGD) tend to $\infty$.

The new theoretical results leverage existing general risk bounds for one-pass SGD.  Novel bounds for the sketched problem are deduced which are needed to get the sharp dependence on the number of random features $M$.

**Limitations.**
The risk bounds are not sharp for all parameters in the phase space.
* $\sigma^2 = 0$ is not covered, nor are the estimates uniform as we take $\sigma^2 \to 0$.  (To the authors: please make it clear in Theorems 4.2,4.3 if this is the case; this was inferred from how you write Theorem 4 and from the $\Theta$ bound on the variance, which otherwise would seem too large.)

    Hence we are constrained to an asymptotic regime in which we have a fixed irreducible loss level, say $42$.  We are then trying to get to $42 + 10^{-x}$ risk, and we get an accurate picture of how to achieve this risk as $x\to\infty$.

    It seems to me that $\sigma^2$ should be small as a function of $M$, to be relevant to the scaling laws of Hoffman et al or Kaplan et al.  (Note that both these papers increase $M$ by $1000$ without seeing something like $\sigma^2$)

* In the range $b \geq a+1$, the estimates are not sharp.  So, the theoretical results are not quite all in place for the regime they consider.


**Novelty** The reviewers have raised the concern of novelty of these results.  The opinion of the AC is that the risk bounds are new in the regimes they consider (in contrast to the concerns raised by the reviewers);  the key differentiator between the results of (13551) and existing work is that one must make estimates for the spectrum of the sketched covariance (c.f. Lemma 6.2).  Having done so, one has a finite-dimensional approximation of a kernel problem, which requires effective SGD risk bounds.

Now on the other hand, it is also true that the role of $M$ is very slight in this whole picture.  And one can take $M\to\infty$ first in this picture, the bounds are consistent with existing kernel regression bounds at $M=\infty$.  Hence it is also true that the random sketch is not doing very much, which may be the source of this confusion.  (Indeed Lemma 6.2 in some sense is the statement the random sketch is just a truncation of the infinite kernel problem, in a relatively strong sense).

So the novelty in this paper is indeed the sharp analysis of a particular random least squares problem, especially eigenvalue/eigenvector estimates.

**Requests for modifications.**

* The reviewers have pointed out literature on source/capacity conditions, especially the list of v1rx.  You have written in the openreview comments how some of your bounds compare to existing ones.  These comparisons to the existing kernel regression literature will greatly contextualize these results, and would improve the manuscript.

**Recommendation**
This paper addresses a recent topic of interest (scaling laws, esp. compute constrained scaling laws) by sharpening and extending existing bounds, in particular giving bounds for powerlaw random features problems which are not in the existing literature.  It should be accepted as a poster.